# Approximately Pareto-optimal Solutions for Bi-Objective $k$-Clustering Problems

**Anna Arutyunova**
Heinrich Heine University Düsseldorf
Düsseldorf, Germany
anna.arutyunova@hhu.de

**Jan Eube**
University of Bonn
Bonn, Germany
eube@cs.uni-bonn.de

**Heiko Röglin**
University of Bonn
Bonn, Germany
roeglin@cs.uni-bonn.de

**Melanie Schmidt**
Heinrich Heine University Düsseldorf
Düsseldorf, Germany
mschmidt@hhu.de

**Sarah Sturm**
University of Bonn
Bonn, Germany
sturm@cs.uni-bonn.de

**Julian Wargalla**
Heinrich Heine University Düsseldorf
Düsseldorf, Germany
julian.wargalla@hhu.de

## Abstract

As a major unsupervised learning method, clustering has received a lot of attention over multiple decades. The various clustering problems that have been studied intensively include, e.g., the $k$-means problem and the $k$-center problem. However, in applications, it is common that good clusterings should optimize multiple objectives (e.g., visualizing data on a map by clustering districts into areas that are both geographically compact but also homogeneous with respect to the data). We study combinations of different objectives, for example optimizing $k$-center and $k$-means simultaneously or optimizing $k$-center with respect to two different metrics. Usually these objectives are conflicting and cannot be optimized simultaneously, making it necessary to find trade-offs. We develop novel algorithms for approximating the set of Pareto-optimal solutions for various combinations of two objectives. Our algorithms achieve provable approximation guarantees and we demonstrate in several experiments that the approximate Pareto front contains good clusterings that cannot be found by considering one of the objectives separately.

## 1 Introduction

Clustering is a major unsupervised learning method that is used to find structure in data. It is often described as the process of dividing objects into groups called *clusters* such that objects in the same cluster are similar and objects in different clusters are dissimilar. There are several mathematical and algorithmic ways to describe a good clustering, but two objectives that correspond to that description for the case of metric clustering are the *$k$-diameter problem* and *Single Linkage* clustering.

In the $k$-diameter problem, we are given a point set $\mathcal{P}$ from a metric space and a number $k$ and want to find a partition of $\mathcal{P}$ into $k$ clusters that minimizes the maximum diameter of any cluster. This models the first problem: Finding clusters where the objects in the same cluster are similar. This problem can be 2-approximated by well-known algorithms [30, 37]. But are approximate or even optimal solutions for this problem automatically also good for the second half of the description,

38th Conference on Neural Information Processing Systems (NeurIPS 2024).

Table 1: State-of-the-art approximation factors

| objective | rad / diam | med | mean | msr |
|---|---|---|---|---|
| best guarantee / reference | 2 / [30, 37] | 2.67059 / [16] | $9 + \epsilon$ / [4] | $3 + \epsilon$ / [11] |

i.e., ensuring that points in different clusters are dissimilar? Certainly not, as the small example in Figure 1 shows: An optimal $k$-diameter clustering with $k = 2$ has radius $\Delta$, but $x_2$ and $x_3$ are very close together in this solution.

Single Linkage clustering is a linkage method that finds the clustering in which the minimum distance between any two clusters is maximized, i.e., it focuses on the second goal. We refer to the problem of maximizing the minimum inter-cluster distance as the *k-separation problem* in the following, i.e., Single Linkage is an optimal algorithm for the $k$-separation problem. In Figure 1 with $k = 2$, Single Linkage computes the clustering $\{\{x_1, x_2, x_3\}, \{x_4\}\}$ or the symmetric clustering.

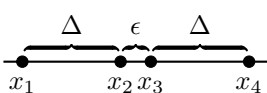

Figure 1: A toy example.

Unfortunately, there is an inherent tension between the $k$-diameter and $k$-separation problem, even if the points are on the Euclidean line $\mathbb{R}$. In Figure 1, we could observe that the optimal $k$-separation clustering is still a very good $k$-diameter clustering (assuming $\epsilon$ to be small).

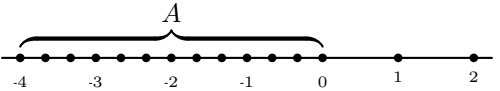

Figure 2: The set $\mathcal{X}_k$ for $k = 3$.

However, one can construct examples where the two cost functions cannot be reconciled:

**Example 1.** *For any fixed $k \in \mathbb{N}$ consider the set $\mathcal{X}_k = \{1, \ldots, k-1\} \cup A$ with $A = \{-x/k \mid x \in \{0, \ldots, k(k-1)^2\}\}$. Figure 2 illustrates this for $k = 3$. Now the only way to obtain a minimum inter-cluster distance of $1$ is to use $k - 1$ singleton clusters $\{1\}, \ldots, \{k-1\}$ and one cluster for $A$ (any other clustering needs to break $A$ and has a separation of $1/k$). Notice that this clustering has a maximum diameter of $(k-1)^2$ since that is the diameter of $A$. However, the best $k$-diameter clustering divides the points into intervals of equal length, achieving a diameter of $((k-1)^2 + (k-1))/k = k-1$, which is smaller by a factor of $k - 1$.*

Example 1 demonstrates that the natural goal of clustering is inherently a goal with two potentially conflicting objectives. Our formalization with $k$-diameter and $k$-separation is an example of a bi-objective clustering problem.

This is only one motivation for studying multi-objective (or bi-objective) clustering. Clustering with respect to multiple objectives simultaneously or based on different metrics arises naturally in applications (we cite a selection of works at the beginning of the related work section). Yet the theoretical study of multi-objective clustering problems is, so far, very limited. We consider the case of bi-objective clustering and study combinations of various clustering objectives with the possibility of optimizing each over its individual metric.

**Objectives** We restrict to the case where centers are chosen from the point set. This is common for $k$-median, but uncommon for $k$-means. We leave the extension where centers can be chosen from an ambient metric space to future work. So let $(\mathcal{P}, d)$ be a metric space. Given $C \subset \mathcal{P}$ with $|C| = k$ and an assignment $\sigma \colon \mathcal{P} \to C$ we define the following objective functions:

$k$**-center:** $\mathsf{rad}(C, \sigma, d) = \max_{p \in \mathcal{P}} d(p, \sigma(p))$
$k$**-diameter:** $\mathsf{diam}(C, \sigma, d) = \max_{p, q \in \mathcal{P} \colon \sigma(p) = \sigma(q)} d(p, q)$
$k$**-median:** $\mathsf{med}(C, \sigma, d) = \sum_{p \in \mathcal{P}} d(p, \sigma(p))$
$k$**-means:** $\mathsf{mean}(C, \sigma, d) = \sum_{p \in \mathcal{P}} d^2(p, \sigma(p))$
$k$**-min sum radii ($k$-msr):** $\mathsf{msr}(C, \sigma, d) = \sum_{c \in C} \max_{p \in \sigma^{-1}(c)} d(c, p)$
$k$**-separation:** $\mathsf{sep}(C, \sigma, d) = \min_{p, q \in \mathcal{P} \colon \sigma(p) \neq \sigma(q)} d(p, q)$

Notice that in contrast to standard definitions, here we need an assignment function because our solutions are trade-offs and may not necessarily assign points to closest centers (for $k$-diameter and $k$-separation we do not need the centers but only the partitioning that $\sigma$ induces; we use the same notation here for convenience and introduce additional notation in the respective technical sections). Each objective induces an optimization problem when we fix the number of clusters $k \in \mathbb{N}$ we want to

obtain[1]. The $k$-separation problem is a maximization problem, all other objectives are to be minimized, i.e., the aim is to compute a set $C$ with $k$ centers and an assignment function $\sigma$ with $\sigma^{-1}(c) \neq \emptyset$ for all $c \in C$ such that the respective objective function is as low as possible. All the minimization problems are NP-hard and also NP-hard to optimize to arbitrary precision [17, 31, 40, 44]. On the positive side, approximation algorithms with small constant factors are known for all the minimization problems mentioned, see Table 1 (all numbers are for the general metric case, including $k$-means). The $k$-separation maximization problem can be solved optimally by running the well-known Single Linkage algorithm until the given number $k$ of clusters is left.

**Use Cases**    We have two different use cases: 1) Simultaneously optimizing a well-known $k$-clustering minimization problem together with the conflicting $k$-separation maximization problem. 2) Optimizing two $k$-clustering minimization problems simultaneously. In both cases, we allow to choose different metrics for each objective. As an example for 2), imagine that we want to visualize data on a map by clustering districts into larger regions. Then regions should be homogeneous with respect to the data but at the same time also geographically coherent on the map, not stretching out too far. For this application we can use as the first metric the data-based metric and as the second metric the geographic distance between regions on the map. Then we can choose an objective, e.g., $k$-center, and find a good trade-off when optimizing $k$-center with respect to the two objectives. This idea is discussed in more detail in Section 3.2.

**Multi-objective Optimization and Pareto Sets**    Both use cases can benefit from the same tool box of multi-objective optimization. Standard approaches are optimizing a weighted sum of the objectives, or limiting one objective and optimizing the other (leading to a constrained clustering problem), or computing/approximating the *Pareto front*. We follow the latter approach and focus on algorithms to approximate the Pareto front of bi-objective clustering problems. Let us first define this concept formally: Given two metrics $d_1, d_2$ on $\mathcal{X}$ and two objectives $f_1, f_2 \in \{\mathsf{rad}, \mathsf{diam}, \mathsf{med}, \mathsf{mean}, \mathsf{msr}, \mathsf{sep}\}$, a solution $(C, \sigma)$ is *dominated* by a solution $(C', \sigma')$ if the following is true for both $i = 1, 2$: $f_i(C', \sigma', d_i) \leq f_i(C, \sigma, d_i)$ if $f_i$ is a minimization objective and $f_i(C', \sigma', d_i) \geq f_i(C, \sigma, d_i)$ if $f_i$ is a maximization objective, and in addition, for $i = 1$ or $i = 2$, the inequality is strict. A solution $(C, \sigma)$ is called *Pareto-optimal* if it is not dominated by any other solution. We denote by $\mathbb{P}$ the set of all Pareto optimal solutions and call $\mathbb{P}$ the *Pareto set* or *Pareto front*.

Notice that by definition the Pareto front contains a solution $(C_i, \sigma_i)$ optimizing objective $f_i$ for $i = 1, 2$. Since all aforementioned minimization objectives are NP-hard, we therefore cannot compute the Pareto fronts exactly in polynomial time unless $\mathsf{P} = \mathsf{NP}$. Instead we design algorithms for computing approximate Pareto fronts in the following sense: Given $\alpha = (\alpha_1, \alpha_2) \in \mathbb{R}^2$, a set $\mathbb{P}_\alpha$ is an $\alpha$-approximate Pareto set if the following is true: For every Pareto-optimal solution $(C, \sigma) \in \mathbb{P}$, there is a solution $(C', \sigma') \in \mathbb{P}_\alpha$ that is an $\alpha$-approximation to $(C, \sigma)$, i.e., $f_i(C', \sigma', d_1) \leq \alpha_i f_i(C, \sigma, d_i)$ for $i \in \{1, 2\}$ if $f_i$ is a minimization objective and $f_i(C', \sigma', d_1) \geq (1/\alpha_i) f_i(C, \sigma, d_i)$ for $i \in \{1, 2\}$ if $f_i$ is a maximization objective. Figure 3 shows two examples of approximate Pareto sets. In 3a) the trade-off is between $k$-separation and $k$-means (use case 1). This is discussed in Section 3.1. In 3b) both objectives are chosen as $k$-center, but with two different metrics that arise from a map design question discussed in Section 3.2 (use case 2). In both pictures the area dominated by a solution lies to the right and above the corresponding point.

While other approaches for multi-objective optimization (like optimizing a weighted sum of the objectives or solving a constrained clustering problem) produce a single trade-off of the objectives, an important advantage of the approximate Pareto front is that it consists of a diverse set of trade-offs. This is beneficial for data analysis because it offers more information and flexibility to the data analyst. In particular, an approximate Pareto front contains for any weighting of the objectives an approximately optimal solution and often, in addition to this, also other interesting solutions. We develop and analyze algorithms for computing approximate Pareto fronts. We also demonstrate experimentally the usefulness of approximate Pareto fronts and our algorithms in different applications.

---

[1]We treat $k$ as an exact bound (except for the somewhat different $\mathsf{msr}$ objective). For minimization problems, choosing $\leq k$ or $= k$ centers makes no difference since splitting clusters improves the cost. We prefer $= k$ centers because otherwise, combinations with $k$-separation would include a trivial Pareto-optimal solution that just constructs one cluster $P$ with infinite separation.

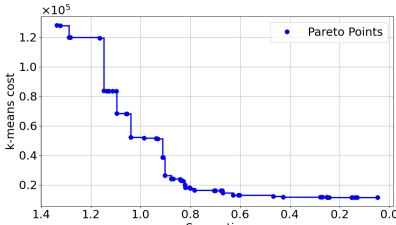
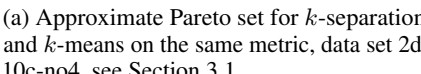
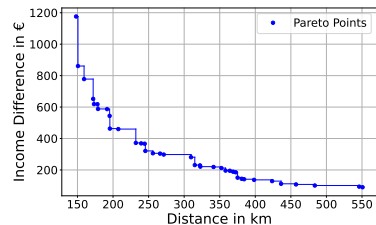

(a) Approximate Pareto set for $k$-separation and $k$-means on the same metric, data set 2d-10c-no4, see Section 3.1.

(b) Approximate Pareto set for $k$-center with two metrics as described in Section 3.2, with $k = 16$ clusters.

Figure 3: Examples of approximate Pareto sets copied from the later experimental sections.

**Related work on Pareto sets**  There is a large body of literature on multi-criteria optimization problems in different contexts (see, e.g., the book of Ehrgott [23]). A common approach is to compute or approximate the Pareto set because only Pareto-optimal solutions constitute reasonable trade-offs. If the Pareto set or its approximation is small, it can be presented to a human decision maker or used in some other post-processing step. For many multi-criteria optimization problems, algorithms for computing the Pareto set have been developed (e.g., for the multi-criteria shortest path problem [19] and the bi-criteria network flow problem [48]) The running time of these algorithms is usually polynomial in the output size, i.e., they are efficient if the Pareto set is small. A drawback of this is that Pareto sets can often be of exponential size in the worst case. To obtain more efficient algorithms and to reduce the size of the output, approximate Pareto sets have been studied extensively. Papadimitriou and Yannakakis showed that there always exist approximate Pareto sets of polynomial size and they showed that an $\alpha$-approximate Pareto set [49] for a problem can be computed efficiently if and only if a certain variant of the problem, called the gap problem, can be solved efficiently.

**Related Work on multi-objective clustering**  Heuristics for approximating Pareto fronts for clusterings are used in applications. Popular techniques include, e.g., genetic algorithms [24, 42], multi-objective particle swarm optimization [10, 14], evolutionary algorithms [36, 47], and combinations of $k$-means and fuzzy c-means [9]. All these heuristics have in common that they come without a guarantee on the approximation factor. We cannot review all the literature in this area and focus on what we perceive as most related.

The only work studying the same scenario is a paper by Alamdari and Shmoys [5] who discuss the combination of $k$-center and $k$-median and give, following the above definition, an algorithm that computes a $(4, 8)$-approximation of the Pareto front. Their algorithm uses an LP rounding approach based on a $k$-median LP with an additional side constraint that restricts the radius of clusters to a given number $L$. The algorithm uses an optimal (fractional) solution to this LP and rounds it to an integral solution, hereby increasing the radius to $4L$ and the $k$-median cost to $8$ times the LP cost. The general approach has some resemblance with our findings in Section 2.2 (and Supplementary Section B.3), but it is based on the LP rounding algorithm by Charikar et al. [12], while we adapt the primal dual algorithm by Jain and Vazirani [41]. We achieve an incomparable approximation factor of $(9, 6 + \epsilon)$. Alamdari and Shmoys also give an example that shows that the two objectives $k$-center and $k$-median cannot be optimized simultaneously in general, making a similar point as we do in Example 1 showing that at least one of the objectives can at best be $\Omega(\sqrt{n})$-approximated.

Davidson and Ravi [20] discuss the combination of $k$-means and must-link and cannot-link constraints. As we discuss in the introduction and in Lemma 3, enforcing a certain separability translates directly into must-link constraints. However, [20] is mostly concerned with NP-hardness results (which stem mostly from cannot-link constraints) and reformulating constraints in a computationally more feasible way. Our focus is on the computationally very efficient Single Linkage / must-link constraints and the computation of Pareto optimal solutions.

In Section 3.1 (and Section C.1), we evaluate if the usage of orthogonal objectives can improve the performance of clustering with respect to recovering a ground truth clustering. A similar approach is evaluated by Handl and Knowles in [35]. The first objective studied there is a $k$-means/$k$-median type objective called *deviation*. The second objective is based on nearest neighbors and evaluates the

fraction of points that satisfies that the point is in the same cluster as its nearest neighbor. This has a similar purpose as our use of separation in Section 3.1, and also leads to improved F-measures. The focus of that work is still different as we focus on provable approximations of the Pareto front.

For the evaluation in that same section, we use normalized mutual information scores following Fred and Jain [27]. In that paper, the idea is to combine multiple clusterings into one clustering in order to obtain more robust clusterings. The authors use a clustering ensemble of different solutions computed by $k$-means and then let their combination be guided based on the normalized mutual information score. The focus of that work is different since we aim to produce an informative ensemble while Fred and Jain aim at combining an ensemble into a single robust clustering.

## 2 Results

We provide algorithms to compute the approximate Pareto set for most combinations of two clustering objectives $f_1, f_2 \in \{\mathsf{rad}, \mathsf{diam}, \mathsf{med}, \mathsf{mean}, \mathsf{msr}, \mathsf{sep}\}$. To compute the approximate Pareto set we make use of the variety of approximation algorithms which optimize over one of the clustering objectives. This includes the 2-approximation for $k$-center/$k$-diameter by Hochbaum and Shmoys [37] which we adapt to compute an approximate Pareto set for the combination of $f_1 \in \{\mathsf{rad}, \mathsf{diam}, \mathsf{sep}\}$ and $f_2 \in \{\mathsf{rad}/\mathsf{diam}\}$. We further incorporate the radius/diameter in the $(6 + \epsilon)$-approximation for $k$-median (also yielding an $O(1)$-approximation for $k$-means) by Jain and Vazirani [41] to obtain an approximate Pareto set for the combination of $f_1 \in \{\mathsf{rad}, \mathsf{diam}\}$ and $f_2 \in \{\mathsf{med}, \mathsf{mean}\}$. We adapt the $(3 + \epsilon)$-approximation for $k$-min sum radii by Buchem et al. [11] to deal with clusters instead of points such that it can be combined with $k$-separation. For the combinations of $f_1 = \mathsf{sep}$ and $f_2 \in \{\mathsf{med}, \mathsf{mean}\}$ we use existing algorithms to compute a solution which optimizes $f_1$ and a solution which optimizes $f_2$ and combine them to a good solution for both objectives, similar to the nesting technique described by Lin et al. [46]. Lastly for the combinations of $f_1, f_2 \in \{\mathsf{med}, \mathsf{mean}\}$ we use the result on convex Pareto sets by Diakonikolas [22]. All of these adaptions are straightforward and we mainly follow the existing work. For the computation of the approximate Pareto set for $f_1, f_2 \in \{\mathsf{rad}, \mathsf{diam}\}$ we furthermore design a new algorithm with improved running time which differs significantly from the naive implementation of the Hochbaum and Shmoys [37] adaptation.

We complement these results by providing instances where it is not possible to find solutions which simultaneously approximate $k$-separation and any of the minimization problems to constant factors. We see this for $k$-separation and $k$-diameter in Ex. 1 above, it works analogously for $k$-center. For $k$-median we analyze a similar example in Obs. 11, $k$-means works analogously. For $k$-separation and $k$-MSR, the example can be found in Obs. 21. This highlights the advantage of Pareto fronts which allow for a trade-off between two opposing clustering objectives. In the following we explain our results and techniques in more detail.

### 2.1 Combining $k$-separation with various $k$-clustering minimization objectives

Single Linkage is known to have undesirable properties for practical applications, in particular chaining effects. Yet its underlying idea – creating separated clusters – makes a lot of sense intuitively. In this set of results, we investigate the combination of Single Linkage, i.e., the maximization of the $k$-separation objective, with algorithms that aim to construct good clusters with respect to various popular $k$-clustering criteria. The idea is that the clustering may benefit from a certain extent of cluster separation without the negative effect of using only Single Linkage to obtain the solution. In Section 3.1 we exemplify the benefit of this idea by a practical evaluation of the $k$-separation / $k$-means combination. There we find that ensuring a small amount of separability and then running $k$-means++ yields good results with respect to recovering a ground truth clustering in our experiments.

Table 2 gives an overview of our theoretical findings. An entry $(1, a)$ means that we show how to compute the approximate Pareto set $\mathbb{P}_{(1,a)}$. More precisely: Let $d_1$ and $d_2$ be two metrics on the same point set, and let $f_2 \in \{\mathsf{sep}, \mathsf{rad}, \mathsf{diam}, \mathsf{med}, \mathsf{mean}, \mathsf{msr}\}$ be the second objective. Now let $\alpha = (1, a)$ be the entry in the corresponding column of Table 2. Then this means that we can compute an $\alpha$-approximate Pareto set $\mathbb{P}_\alpha$ with respect to $\mathsf{sep}$ with metric $d_1$ and $f_2$ with metric $d_2$. The second line refers to the corresponding theorem. All the individual theorems and their proofs are in the supplementary material.

Table 2: Results for combining $k$-separation with various objectives. Here $\delta_1$ and $\delta_2$ refer to the best known approximation guarantee for $k$-median/$k$-means, currently $\delta_1 = 2.67059/\delta_2 = 9 + \epsilon$ [16, 4].

|  | $k$-separation | $k$-center | $k$-diameter | $k$-median | $k$-means | $k$-MSR |
|---|---|---|---|---|---|---|
| $k$-sep. | $(1,1)$ | $(1,2)$ | $(1,2)$ | $(1, 2 + \delta_1)$ | $(1, 4 + 4\sqrt{\delta_2} + \delta_2)$ | $(1, 3 + \epsilon)$ |
| proof | Thm. 5 | Thm. 9 | Thm. 7 | Thm. 12 | Thm. 17 | Thm. 24 |

**Techniques** The $k$-separation objective is friendly in the sense that we know exactly how optimal solutions look like. First observe that in any optimal solution, the objective value equals the distance of two points in different clusters. This means that there are only $\Theta(n^2)$ possible optimal values. And if we fix a value $\Delta$ to be the desired separation, then this means that all points $x, y$ with $d(x, y) < \Delta$ have to be in the same cluster because otherwise the objective is automatically below $\Delta$. And if we transitively merge all pairs $x, y$ with $d(x, y) < \Delta$, then we get a solution with objective value $\geq \Delta$. So it is well characterized how an optimal $k$-separation solution looks like (also see Lemma 3). To combine $k$-separation with other clustering objectives we first consider such a clustering with separation $\Delta$, if it has already $k$ clusters, there is nothing left to do. If it has more than $k$ clusters we decide which clusters to merge based on the second clustering objective.

This immediately leads to the $(1, 1)$-entry when combining two $k$-separation objectives (for different metrics): Iterate through all $\Theta(n^4)$ combinations of possible optimal objective values $(\Delta_1, \Delta_2)$. For each guess, merge $x, y$ if $d_1(x, y) < \Delta_1$ or $d_2(x, y) < \Delta_2$ (or both). While the running time can certainly be improved, this gives the full optimal Pareto front in polynomial time.

For $f_2 = \mathsf{rad}$ or $f_2 = \mathsf{diam}$, we also have that the optimum value of a solution is a pairwise distance between two points. Thus, there are also at most $O(n^2)$ possible values that $f_2$ can take. There is a well-known algorithm due to Hochbaum and Shmoys [37] that for any given radius $R$ (or diameter) can produce a solution with at most double that value or prove that no solution with value $R$ exists. We adapt this algorithm to work on the above clustering with Separation $\Delta$ as input. The algorithm is similar to the original work by Hochbaum and Shmoys. By iterating through all pairs of possible separation and possible radius/diameter values, we obtain the $(1, 2)$-approximate Pareto set. The details are discussed in Section A.2 of the supplementary material.

When combining sep with $f_2 = \mathsf{med}$ or $f_2 = \mathsf{mean}$ (Supplementary Section A.3), we still know that the separation value $\Delta$ is one of $O(n^2)$ candidates. We use the above clustering with separation $\Delta$ and decide which clusters to merge further based on an approximate clustering with respect to $f_2$. For this purpose we use the nesting technique described by Lin et al. [46] in the context of approximation algorithms for hierarchical clustering. Since we need an approximate clustering with respect to $f_2$, the quality of our Pareto set depends on the approximation ratio of the algorithm for $f_2$.

Finally, we combine sep with msr. This is the only place where we study msr as it is less common than the other clustering objectives. It is also the only objective where we deviate from the rule that clusters have to have $= k$ clusters and allow $< k$ clusters. We study it because it seems more aligned with $k$-separation than the other objectives. However, Obs. 21 reveals that also $k$-separation and $k$-MSR cannot be approximated simultaneously. In order to approximate the Pareto front, we adapt the very recent state-of-the-art algorithm for $k$-MSR approximation from [11] to our setting. That algorithm has an approximation ratio of $3 + \epsilon$ and indeed, we show how to compute $(1, 3 + \epsilon)$-approximate Pareto fronts. Buchem et al. [11] give an intricate primal-dual algorithm for the $k$-MSR problem. Our general idea follows the above themes: We iterate through all possible separation values $\Delta$, obtain a clustering with separation $\Delta$, and then compute an approximate $k$-MSR solution. However, for this to work we need to adapt the algorithm to cope with clusters instead of points. Similar to the adaption of the algorithm by Hochbaum and Shmoys the input now consists of clusters, and this needs to be taken into account when trying to cover them with balls. We discuss this in the supplementary material in Sec. A.4.

### 2.2 Combining $k$-center or $k$-diameter with a $k$-clustering minimization problem

Now we turn to combinations of popular $k$-clustering minimization problems. Table 3 lists combinations involving $k$-center or $k$-diameter. Again, let $d_1$ and $d_2$ be two metrics on the same point set. Then let $f_1 \in \{\mathsf{rad}, \mathsf{diam}\}$, let $f_2 \in \{\mathsf{rad}, \mathsf{diam}, \mathsf{med}, \mathsf{mean}\}$ be the second objective. Finally, let

Table 3: Results for combining rad and diam for two different metrics or with a sum-based objective. The combination of $k$-center/$k$-diameter with $k$-separation can be found in Table 2.

|  | $k$-center/$k$-diameter | $k$-median | $k$-means |
|---|---|---|---|
| $k$-center/$k$-diameter | $(2, 2)$ | $(9, 6 + \epsilon)$ / $(18, 6 + \epsilon)$ | $(9, 54 + \epsilon)$ / $(18, 54 + \epsilon)$ |
| proof | Cor. 37 | Thm. 41 / Cor. 44 | Thm. 42 / Cor. 44 |

$\alpha = (a, b)$ be the entry corresponding to $f_1$ and $f_2$ in Table 3. Then this means that we can compute an $\alpha$-approximate Pareto set $\mathbb{P}_\alpha$ with respect to rad or diam with metric $d_1$ and $f_2$ with metric $d_2$. The second row links to the proof in the supplementary material.

**Techniques.** The $k$-center objective has the already mentioned property that there are at most $O(n^2)$ possible objective values, one for each pairwise distance in the data set. This also directly limits the size of the resulting Pareto Set when combining two $k$-center or $k$-diameter objectives by $O(n^2)$. This is tight since we were able to provide an instance where the optimum Pareto set has indeed a size of $\Omega(n^2)$ (Supplementary Section B.2). However it can be proven that there always exists a $(2, 2)$-approximation of the Pareto front of size $O(n)$.

We adapt a well-known 2-approximation algorithm due to Hochbaum and Shmoys [37]. A straightforward adaption of this algorithm would iterate through $O(n^4)$ pairs $(r_1, r_2)$ of possible radii and check for each of them if a maximal independent set of size at most $k$ can be found in an appropriate threshold graph. While this would result in a running time of $O(n^6)$, we improve the running time by observing that only certain pairs $(r_1, r_2)$ need to be checked (in total $O(n^2)$ many) and by updating the threshold graph dynamically. While updating the threshold graph can have a time complexity of up to $\Omega(n^2)$ in certain iterations, we were able to use a suitable potential function to show that the amortized time needed to update the graph lies within $O(n)$ per iteration. Hereby we show that we can compute the approximate Pareto set $\mathbb{P}_{(2,2)}$ in time $O(n^3)$. The algorithm and its analysis also work for $k$-diameter, giving the same guarantees. For more details we refer to Section B.1 of the supplementary material.

To combine $k$-center with $k$-median and $k$-means (possibly with different metrics), we adapt the primal dual algorithm due to Jain and Vazirani [41]. Alamdari and Shmoys [5] adapt the approximation algorithm by Charikar et al. [12], which leads to a $(4, 8)$-approximate Pareto set. For recent algorithms [4, 15, 16], incorporating radius requirements does not seem easily possible.

The primal dual algorithm gives a $(6 + \epsilon)$-approximation guarantee for $k$-median and a 54-approximation guarantee for $k$-means (the paper only states $108$, but that is because the $k$-means problem is considered in $\mathbb{R}^d$ there, while we study it in the setting where centers are chosen from the input point set. As [4] also observe, the guarantee then improves to $54$.) The primal dual algorithm is based on an LP formulation where variables $x_{ij}$ indicate that point $j$ is assigned to center $i$. The adaptation starts by fixing a radius $R$ and removing all $x_{ij}$ for which $d(i, j) > R$. It then follows the proof of [41] to first obtain an algorithm for approximating facility location and then obtaining a $k$-median (or $k$-means) solution by combining two solutions: one with $< k$ clusters and high cost and one with $> k$ clusters and small cost. In the combination step we have to ensure that the radius stays bounded (see Lemma 47). We can ensure a 9-approximation with respect to the radius while keeping the original approximation factors for $k$-median/$k$-median from [41]. To extend the results to $k$-diameter, we use that the diameter is at most twice the radius. This standard observation transfers the results for rad with med and mean to diam.

## 2.3 Combinations of $k$-median and $k$-means

In the previous sections we have enumerated over a candidate set that contained all Pareto-optimal solutions and used this enumeration to compute approximately optimal solutions. This is no longer feasible here because the Pareto set is too large: Theorem 48 in the supplementary material shows that the combination of med and mean both with the same Euclidean metric can lead to Pareto sets of exponential size. Similar examples can be constructed for $f_1 = f_2$ and $f_1 \in \{\text{med}, \text{mean}\}$ if $d_1 \neq d_2$.

Table 4: Number of points $n$, dimension $d$, and number of desired clusters $k$ for all data sets in §3.1.

| **name** | Rice | Dry Bean | Wine | Optdigits | Iris | 2d-4c-no3 | 2d-10c-no3 | 2d-10c-no4 |
|---|---|---|---|---|---|---|---|---|
| **ref** | [1] | [2] | [3] | [6] | [26] | [34] | [32] | [34] |
| **n** | 3810 | 13611 | 178 | 5620 | 150 | 1123 | 3359 | 3291 |
| **d** | 7 | 16 | 13 | 64 | 4 | 2 | 2 | 2 |
| **k** | 2 | 7 | 3 | 10 | 3 | 4 | 10 | 10 |

A possible approach to compute an approximate Pareto set is due to Papadimitrou and Yannakakis [49]. They showed that there always exists an $\epsilon$-approximate Pareto set whose size is polynomial in the input size and $\frac{1}{\epsilon'}$ where $\epsilon = (1 + \epsilon', 1 + \epsilon')$. There is a polynomial algorithm to construct such an $\epsilon$-approximate Pareto set if and only if one is able to solve the following problem in polynomial time: Given the instance and $b_1, b_2 \in \mathbb{R}$, find a solution $s$ with $f_i(s) \leq b_i$ for $i \in \{1, 2\}$ or report that there is no solution $s$ with $(1 + \epsilon')f_i(s) \leq b_i$ for $i \in \{1, 2\}$. Unfortunately, for the combinations of the objectives med, mean it is unknown, whether this problem can be solved efficiently. We leave this as an open problem.

An alternative to approximate Pareto sets is to consider another subset of the Pareto set, namely the set of *supported* Pareto-optimal solutions, i.e., the set of solutions that are optimal for a weighted sum of the objectives. For two minimization objectives $f_1$ and $f_2$, a solution $(C, \sigma)$ is a supported Pareto-optimal solution if there exist weights $w_1, w_2 \geq 0$ such that $(C, \sigma)$ is a solution for the objective $w_1 f_1 + w_2 f_2$ with minimum value. For a solution $(C, \sigma)$ let $F(C, \sigma) = (f_1(C, \sigma), f_2(C, \sigma))$ and $F(\mathbb{P}) = \{F(C, \sigma) \mid (C, \sigma) \in \mathbb{P}\}$. The supported Pareto-optimal solutions are exactly the solutions that form the vertices of the convex hull of $F(\mathbb{P})$. Hence, we will call the set of supported Pareto-optimal solutions the *convex Pareto set* in the following. Diakonikolas [22] has also introduced the notion of approximate convex Pareto sets. For $\alpha = (\alpha_1, \alpha_2) \in \mathbb{R}^2$ we say that a set of solutions $\mathbb{CP}_\alpha$ is an approximate convex Pareto set if for any solution $(C, \sigma)$ there is a a convex combination of solutions in $\mathbb{CP}_\alpha$ that is an $\alpha$-approximation to $(C, \sigma)$. In this context we always assume $\alpha_1 = \alpha_2$ and therefore just write $\mathbb{CP}_{\alpha_1}$ instead of $\mathbb{CP}_{(\alpha_1, \alpha_2)}$ in the following. Diakonikolas [22] showed that an (F)PTAS for the convex Pareto exists if there is an (F)PTAS for optimizing weighted sums of the objectives. Based on this, we obtain the following theorem.

**Theorem 51.** *Given a finite set $\mathcal{P}$, metrics $d_1, d_2$ on $\mathcal{P}$ and two objectives $f_1, f_2 \in \{\text{med}, \text{mean}\}$. Let $\epsilon > 0$. Then we can compute an $(\delta + \epsilon)$-approximate convex Pareto set $\mathbb{CP}_{\delta+\epsilon}$ in time that is polynomial in the input size and $\frac{1}{\epsilon}$, where $\delta = 2.67059$ if $f_1 = \text{med}$ and $f_2 = \text{med}$, and $\delta = 9 + \epsilon$ if $f_1 = \text{med}$ and $f_2 = \text{mean}$ or $f_1 = \text{mean}$ and $f_2 = \text{mean}$. The size of this set $|\mathbb{CP}_{\delta+\epsilon}|$ is also polynomial in $\frac{1}{\epsilon}$ and $|P|$.*

## 3 Applications and Experimental Evaluation

We implemented algorithms for two objective combinations and tested them on different applications. The source code can be found at `https://github.com/algo-hhu/paretoClustering`.

### 3.1 Application: $k$-separation and $k$-means

We test the approach to combine $k$-separation with a conflicting objective. As that second objective, we choose $k$-means. Since the data sets used in our experiments are from $\mathbb{R}^d$, we do not use the algorithm described in Sec. A.3 for general metrics but a variant of it which is tailored to $\mathbb{R}^d$ and allows centers from $\mathbb{R}^d$ instead of only $P$. It is described below. The Euclidean metric is used for both sep and mean. We test this algorithm on data sets from $\mathbb{R}^d$ with available ground truth. We fix the number $k$ of clusters to be the desired number of clusters in the ground truth. All data sets were downloaded from freely available sources, see Table 4 for information on the data sets.

**Algorithm.** For every data set, we compute the approximate Pareto set for the desired number of clusters as follows. Recall that for a data set of size $n$ there are only $O(n^2)$ possible values for the separation. We compute these values in time $O(n^2 d)$ and sort them in increasing order in time $O(n^2 \log(n))$. Starting with separation 0, we increase the separation in every step to the next largest value. Suppose the separation is $\Delta$ in the current step, then we merge all points whose distance is at

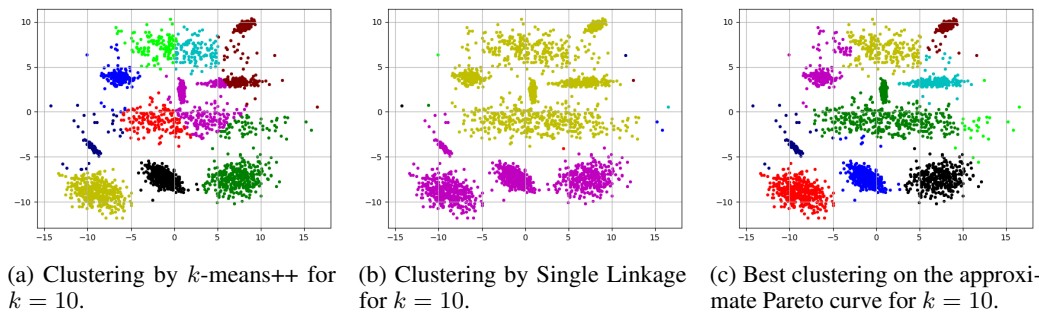

(a) Clustering by $k$-means++ for $k = 10$.

(b) Clustering by Single Linkage for $k = 10$.

(c) Best clustering on the approximate Pareto curve for $k = 10$.

Figure 4: Clusterings computed on data set `2d-10c-no4` by Handl and Knowles [33].

Table 5: NMI of the best solutions by single linkage and $k$-means++, and of the best solution $C^*$ in the Pareto set. Randomized algorithms were repeated 20 times and values are then averages.

|  | Rice | Dry Bean | Wine | Optdigits | Iris | 2d-4c-no3 | 2d-10c-no3 | 2d-10c-no4 |
|---|---|---|---|---|---|---|---|---|
| **SL** | 0.0007 | 0.1626 | 0.0615 | 0.1220 | 0.0000 | 0.4631 | 0.6763 | 0.7260 |
| **kM++** | 0.4693 | 0.5164 | 0.4265 | 0.7459 | 0.7405 | 0.8715 | 0.9065 | 0.9052 |
| **C\*** | 0.4728 | 0.5231 | 0.4400 | 0.7627 | 0.8578 | 0.9267 | 0.9491 | 0.9779 |

most $\Delta$. This can be done efficiently via a Union Find data structure. Since the resulting clustering may have more than $k$ clusters, we have to reduce the number of clusters to $k$. For data sets in $\mathbb{R}^d$ and the $k$-means objective, one can replace every cluster by its centroid weighted by the number of points in the cluster and then cluster these weighted centroids instead of using the nesting technique of Lin et al. [46] for general metrics. Instead of choosing the theoretically best approximation algorithm for $k$-means, we use $k$-means++ [8] to cluster the centroids as it is fast (running time $O(nkd)$) and usually produces solutions of high quality for the $k$-means problem in practice. Then the respective clustering on the original data set has separation at least $\Delta$ and at most $k$ clusters. One can show that this algorithm computes an $\alpha$-approximate Pareto set with $\alpha = (1, O(\log k))$.

**Prototypical desired behavior.** Consider Fig. 4. It shows the data set `2d-10c-no4` by Handl and Knowles [33], which is a synthetic data set with 10 clusters. The first two figures show the clusterings produced by $k$-means++ and Single Linkage for $k = 10$. We see that $k$-means++ finds most clusters but due to its limitation to spherical clusters, some clusters are split up and others are merged (Fig. 4a). Single Linkage on the other hand is too much affected by the outliers: It merges most points into two clusters, using the remaining 8 clusters for very few outlier points (see Fig. 4b). But when allowed many more clusters (here, 85 instead of 10), Single Linkage is very successful at finding the most important groups of points that belong together (see Fig. 8c in the suppl. material). Fig. 3a shows the Pareto curve. The 16th point with a separation of $0.68$ is the most successful combination with repect to all metrics. It is shown in Fig. 4c.

**Numerical evaluation.** We compare the clusterings in the approximate Pareto set to the ground truth. For this purpose we compute the Normalized Mutual Information (NMI), Rand Index (RI), and $F_\beta$-scores for $\beta = 0.5, 1, 2$. Each of these measures results in values in $[0, 1]$, where a value of $1$ is achieved when the computed clustering matches the ground truth. For every measure, we pick the clustering $C^*$ in the approximate Pareto set that has highest value with respect to this measure to demonstrate that there is a good solution on the Pareto front. The full experimental evaluation, plots of Pareto fronts, other plots and also the definition of all scores can be found in Sec. A.3 in the supplementary material. Table 5 shows the Normalized Mutual Information Score for all tested data sets. We see the best solution found in 20 runs of the respective algorithm (except for Single Linkage, which is deterministic). The algorithms were run with the $k$ associated with the ground truth of the data set. Single Linkage does in general not perform well in recovering the ground truth clustering. The clusterings produced by $k$-means++ are much better. For all data sets, the NMI of the best Pareto solution is even higher, with some notable positive examples like `iris` and `2d-10c-n4`. The other performance measures show a similar tendency, although sometimes $C^*$ is slightly worse, for example on data set `rice` (see Table 11 in Sec. A.3). This can happen because the solutions on

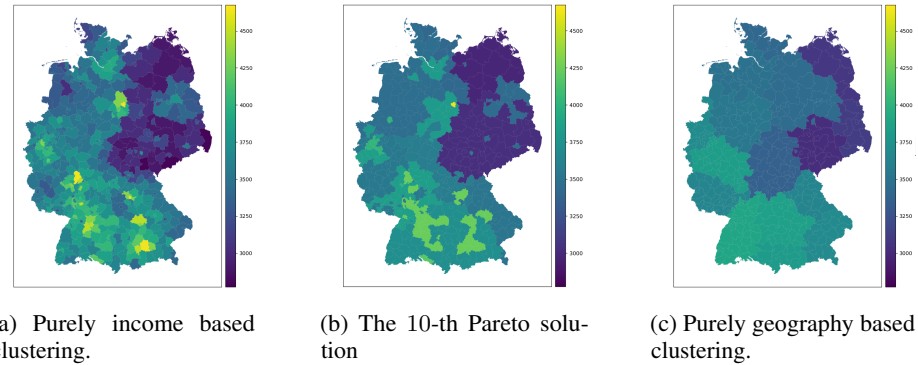

(a) Purely income based clustering.

(b) The 10-th Pareto solution

(c) Purely geography based clustering.

Figure 5: Comparison between the 10-th Pareto solution with the purely geographic and the purely income based clustering for $k = 16$.

Table 6: Comparison between the time series based clustering, the geography based clustering, and the 54-th Pareto clustering with regard to the biggest cluster radius for both metrics and the mean cluster radius over all clusters for both metrics.

|                              | max rad geo | mean rad geoc | max rad ts | mean rad ts |
|------------------------------|-------------|---------------|------------|-------------|
| **Geography based clustering** | 1566 km     | 619 km        | 20,9 cm    | 4,1 cm      |
| **54-th Pareto Solution**      | 2411 km     | 799 km        | 8,0 cm     | 3,1 cm      |
| **Time Series based clustering** | 18172 km  | 1570 km       | 6,7 cm     | 2,8 cm      |

the Pareto front do not necessarily include the pure $k$-means++ solution as special case because it is only an approximation to the optimal $k$-means clustering and therefore can be dominated by other solutions in the approximate Pareto set with smaller $k$-means cost.

On all data sets that are not synthetic, the best result in the approximate Pareto set uses a small separation value. Therefore we conjecture that it is sufficient to use a few steps of single linkage before starting $k$-means++ in order to improve the quality of the solution.

## 3.2 Applications: $k$-center with two different metrics

We use two $k$-center-objectives with different metrics to visualize two different geodetic data sets. The goal in both cases is to visualize data, but to also make it more compact and therefore easier to understand by trying to find geographically close areas that contain similar data. Details of this section are in Supplementary Sections C.2 and C.3. The first data set we use are the monthly median incomes of the 400 districts in Germany as obtained from [29]. We compute an approximate Pareto set for $k = 16$ (the number of states in Germany), which contains a total of 38 different clusterings. Figure 3b shows the approximate Pareto curve. In Figure 5 one can see the 10-th Pareto-optimal solution compared to the clusterings that consider only one of the objectives. One can see that the Pareto solution has a cleaner structure than the purely income based clustering, but at the same time shows how the median income behaves in different regions of Germany.

For our second application we use a data set that consists of (normalized) time series of the sea level height on different locations all over the world. We use time series that were created using the sea level simulation provided by ORA5 [18], at the location where the tide gauge stations of the PSMSL [50, 39] are located. As metric $d_1$ we use the mean difference between the time series. As geographic distance measure $d_2$ we use the Euclidean distance between two stations. The time-series-based distance and the Euclidean distance of the gauge stations can behave very differently. We cluster with $k = 150$. Then the approximate Pareto set contains 106 clusterings. Table 6 shows the maximum and mean radii of the clusters in the solutions computed purely based on geometry, in the solution purely based on the time-series-based distance, and in a carefully picked Pareto-optimal solution. The numbers show that the Pareto-optimal solution is a very reasonable trade-off.

## Acknowledgments and Disclosure of Funding

This work has been funded by the Deutsche Forschungsgemeinschaft (DFG, German Research Foundation) – 390685813 and 459420781 and by the Lamarr Institute for Machine Learning and Artificial Intelligence (lamarr-institute.org). We furthermore thank Johanna Hillebrand for providing code to visualize the sea level data.

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

# Supplementary Material

## A    Theoretical Part: Pareto Sets for Combinations Involving $k$-separation

Given a set of points $\mathcal{P}$, $k \in \mathbb{N}$ and metrics $d_1, d_2$ on $\mathcal{P}$, in the previous sections we define a $k$-clustering on $\mathcal{P}$ as a set $C$ of $k$ centers and an assignment $\sigma \colon \mathcal{P} \to C$. For combinations involving $k$-separation, it can be helpful to view clusterings as partitionings. We introduce the following equivalent notation that is used in Sections A.1, A.2.1 and Section A.2.2.

Observe that $\{\sigma^{-1}(c) \mid c \in C\}$ forms a partition of $\mathcal{P}$ of size $k$. The only change in definition is that we observe that this partition is already sufficient to compute the objective values of the solution. Let $\mathscr{C}$ be a partition of $\mathcal{P}$ of size $k$, for $i = 1, 2$ we define

$$\mathsf{sep}(\mathscr{C}, d_i) = \min_{C, C' \in \mathscr{C}, C \neq C'} \min_{p \in C, q \in C'} d_i(p, q)$$

$$\mathsf{diam}(\mathscr{C}, d_i) = \max_{C \in \mathscr{C}} \max_{p, q \in C} d_i(p, q)$$

$$\mathsf{rad}(\mathscr{C}, d_i) = \max_{C \in \mathscr{C}} \min_{c \in C} \max_{p \in C} d_i(p, c).$$

For convenience we view a $k$-clustering as a partition of $\mathcal{P}$ in $k$ clusters in Sections A.1, A.2.1 and Section A.2.2.

### A.1    Combining $k$-Separation and $k$-Separation

We consider the $k$-separation problem and observe that defining a threshold for the separation translates into a set of must-link constraints. We define the graph of points that have to be connected given a certain separation value $\tau$ and use this to define a clustering.

**Definition 2.** *Let $(\mathcal{P}, d)$ be a finite metric space and $\tau \geq 0$. The $\tau$-separation graph $G_{(\mathcal{P},d)}(\tau)$ of $\mathcal{P}$ consists of*

- *vertices $V_{(\mathcal{P},d)}(\tau) = \mathcal{P}$ and*

- *edges $E_{(\mathcal{P},d)}(\tau) = \{\{x, y\} \mid x, y \in \mathcal{P} \colon 0 < d(x, y) \leq \tau\}$.*

*We denote by $\mathscr{S}_{(\mathcal{P},d)}(\tau)$ the clustering consisting of the connected components of $G_{(\mathcal{P},d)}(\tau)$.*

The following shows that $\mathscr{S}_{(\mathcal{P},d)}$ is in some sense at the heart of $\tau$-separability. For brevity, we use the following in this lemma: We name a clustering $\mathscr{C}$ of $\mathcal{P}$ is $\tau$-*separated* if $\tau \leq \max_{C, C' \in \mathscr{C}, C \neq C'} \min_{(c, c') \in C \times C'} d(c, c')$.

**Lemma 3.** *Let $(\mathcal{P}, d)$ be a metric space and $\tau \geq 0$. Then*

1. *$\mathscr{S}_{(\mathcal{P},d)}(\tau)$ is $\tau$-separated.*

2. *If $\mathscr{C}$ is clustering of $(\mathcal{P}, d)$ then it is $\tau$-separated if and only if $\mathscr{S}_{(\mathcal{P},d)}(\tau)$ is a refinement of clustering $\mathscr{C}$. This means that for every $S \in \mathscr{S}_{(\mathcal{P},d)}(\tau)$ there exists some $C \in \mathscr{C}$ such that $S \subseteq C$.*

*In this sense, $\mathscr{S}_{(\mathcal{P},d)}(\tau)$ can be called the* minimally $\tau$-separated clustering.

*Proof.* This immediately follows from two easy observations:

1. The minimal distance between two different clusters $S, S' \in \mathscr{S}_{(\mathcal{P},d)}(\tau)$ is strictly more than $\tau$. Otherwise an edge would have been added between the respective points.

2. We cannot split any cluster $S \in \mathscr{S}_{(\mathcal{P},d)}(\tau)$ without lowering to separability to at most $\tau$: since $S$ is connected in $G_{(\mathcal{P},d)}(\tau)$ at least one edge would have to be cut.    $\square$

**Lemma 4.** *Let $(\mathcal{P}, d)$ be a metric space consisting of $n$ elements, and $\tau \geq 0$. We can compute $\mathscr{S}_{(\mathcal{P},d)}(\tau)$ in $O(n^2)$ time.*

*Proof.* The $\tau$-separation graph can be computed in $O(n^2)$ time. Connected components can be found in linear time. $\qquad\square$

Now we can conclude that we can compute the optimal Pareto set (in time $O(n^2)$) when we combine $f_1 = \mathsf{sep}$ and $f_2 = \mathsf{sep}$ on possibly different metrics $d_1$ and $d_2$. Given two thresholds $\tau_1$ and $\tau_2$, just compute $\mathscr{S} = \mathscr{S}_{(\mathcal{P},d_1)}(\tau_1)$ and merge all of its clusters $S, S'$ whose distance $\min_{(s,s')\in S\times S'} d_2(s,s')$ is at most $\tau_2$. In other words, we merge $S$ and $S'$, whenever there exists a cluster $S'' \in \mathscr{S}_{(\mathcal{P},d_2)}(\tau_2)$ with $S \cap S'' \neq \varnothing$ and $S' \cap S'' \neq \varnothing$. The resulting clustering $\mathscr{C}$ is minimally hierarchically compatible with both $\mathscr{S}_{(\mathcal{P},d_1)}(\tau_1)$ and $\mathscr{S}_{(\mathcal{P},d_2)}(\tau_2)$ in the sense that every other clustering hierarchically compatible with the latter has to be compatible with $\mathscr{C}$ as well. All merges we have conducted were necessarily forced to achieve the separation values. Had we left out any of them, one of the separation values would be violated. We can thus conclude with the following result.

**Theorem 5.** *Let $d_1, d_2$ be two metrics defined over a finite set $\mathcal{P}$. Then for $f_1 = \mathsf{sep}$ and $f_2 = \mathsf{sep}$ we can compute their Pareto set in polynomial time.*

### A.2 Combination of $k$-Diameter/$k$-Center with $k$-Separation

In this section, we combine $k$-diameter/$k$-center with $k$-separation. Although similar at first sight to the cases already discussed with $k$-diameter/$k$-center, the combinations with $k$-separations differ conceptually quite a bit. Whereas all prior combinations consisted of objective functions that penalize the same behavior, $k$-separation is concerned with a different clustering aspect. Not only do we want dissimilar points to belong to different clusters, but also for similar points to belong to the same cluster. Ideally, clusters should not only be small, but clear lines of demarcation should also run between them.

Between these two goals, there is an inherent tension that cannot readily be resolved. One favors clusterings comprising a small number of clusters (at the limit there are one, or rather, two clusters), while the other favors clusterings comprising a large number of clusters (at the limit every point belongs to a separate cluster). In fact, if we fix the number of clusters, then there's no guarantee that a solution that yields constant-factor approximation for both objectives at the same time even exists, not even when the underlying metric space derives from the Euclidean line $\mathbb{R}$.

Next we present an example where there exists no solution that is simultaneously a constant factor approximation for $k$-center/$k$-diameter and $k$-separation.

**Example 6.** *For any fixed $m \in \mathbb{N}_{\geq 1}$ consider the set*

$$\mathcal{P}_m = \{1, \ldots, k-1\} \cup \underbrace{\{-x/m \mid x \in \{0, \ldots, m(k-1)^2\}\}}_{A_m}$$

*and the Euclidean metric $d_1(x,y) = d_2(x,y) = |x-y|$ on $\mathbb{R}$.*

*We first show that any constant-factor approximation for either the $k$-diameter or the $k$-separation objective performs poorly for the other objective.*

*First, note that $\mathscr{C} = \{\{1\}, \ldots, \{k-1\}, A_m\}$ is the only clustering with a separation value of $\mathsf{sep}(\mathscr{C}, d_2) = 1$. Indeed, any other $k$-clustering $\mathscr{C}'$ has to break up $A_m$ and so has a separation of only $\mathsf{sep}(\mathscr{C}', d_2) = 1/m$ (the distance between any two consecutive points in $A_m$). In other words, $\mathscr{C}$ is the only solution that provides a constant-factor approximation for the $k$-separation objective. However, $\mathscr{C}$ does not provide a good approximation for the $k$-diameter objective. Since it does not break up $A_m$, it has a diameter of $(k-1)^2$. This is worse by a factor of $k-1$ than just indifferently dividing up $\mathcal{P}_m$ into $k$ sets of "equal size". Indeed, partitioning $[-(k-1)^2, k-1] \supset \mathcal{P}_m$ into $k$ intervals of equal length yields a clustering of diameter at most*

$$\frac{(k-1)^2 + (k-1)}{k} = \frac{k(k-1)}{k} = k-1.$$

*The same argument also shows that constant-factor approximations for the $k$-diameter objective perform poorly with respect to the $k$-separation objective. A solution that provides a constant-factor approximation for the $k$-diameter objective has to break up $A_m$ and thus has a separation value that is $m$ times worse than the optimal.*

*For $k$-center the argument works analogously with the only difference that the radius of $\mathscr{C}$ is lower bounded by $\frac{(k-1)^2}{2}$ while the optimal solution with respect to $k$-center has radius at most $\frac{k-1}{2} + 1$.*

Thus, instead of trying to optimize $k$-diameter/$k$-center and $k$-separation simultaneously, we again focus on Pareto sets.

### A.2.1 Combination of $k$-Diameter with $k$-Separation

In line with previous sections we will provide a subroutine that, given an estimation of the value of a Pareto-optimal solution, will compute a close approximation.

**Theorem 7.** *Let $S, D \in \mathbb{R}$ be two non-negative numbers. If there exists a clustering $\mathscr{C}^*$ with $\mathsf{diam}(\mathscr{C}^*, d_1) \leq D$ and $\mathsf{sep}(\mathscr{C}^*, d_2) \geq S$, then Algorithm 1 computes a solution $\mathscr{C}$ in polynomial time with $\mathsf{diam}(\mathscr{C}, d_1) \leq 2D$ and $\mathsf{sep}(\mathscr{C}, d_2) \geq S$. If no such clustering $\mathscr{C}^*$ exists, Algorithm 1 returns an error message.*

Algorithm 1 is a relatively simple subroutine that, for a pair of possible diameter and separation values, either constructs a $(1, 2)$-approximation in polynomial time, or signals that no such Pareto-optimal solution exists. Using this subroutine we can simply iterate over all diameter-separation pairs and collect the results to get an approximate Pareto set $\mathbb{P}_\alpha$ with $\alpha = (1, 2)$.

---

**Algorithm 1:** Approximate solution for $k$-diameter and $k$-separation

**input** : A finite set $\mathcal{P}$, two metrics $d_1, d_2$ on $\mathcal{P}$, a number $k \in \mathbb{N}$, two values $D, S \in \mathbb{R}_{\geq 0}$.

1   $\mathscr{C} \leftarrow \{\{x\} \mid x \in \mathcal{P}\}$
2   **while** $\exists C, C' \in \mathscr{C}, \ C \neq C' : \ d_2(C, C') < S$ **do**
3     $\mathscr{C} \leftarrow (\mathscr{C} \setminus \{C, C'\}) \cup \{C \cup C'\}$
4   **if** $D < \mathsf{diam}(\mathscr{C}, d_1)$ or $|\mathscr{C}| < k$ **then**
5     **return** *no such clustering exists*
6   $G_{\leq D} \leftarrow (\mathscr{C}, \{\{C, C'\} \mid \mathsf{diam}(C \cup C', d_1) \leq D\})$
7   $\mathscr{I} = \{I_1, \ldots, I_\ell\} \leftarrow$ maximal independent set in $G_{\leq D}$
8   **if** $\ell > k$ **then**
9     **return** *no such clustering exists*
10   partition $G_{\leq D}$ into star-graphs $S_1, \ldots S_\ell$ centered around $I_1, \ldots I_\ell$
11   **forall** $i \in \{1, \ldots, \ell\}$ **do**
12     $D = \emptyset$
13     **forall** $C \in S_i$ **do**
14       **if** $|\mathscr{C}| = k$ **then**
15         **return** $\mathscr{C}$
16       $\mathscr{C} \leftarrow (\mathscr{C} \setminus \{D\}) \cup \{D \cup C\}$
17       $D \leftarrow D \cup C$
18   **return** $\mathscr{C}$

---

Before proving Theorem 7 we will quickly outline Algorithm 1, splitting it into two phases. In the first phase (we could also call it the Single-Linkage phase), starting with $|\mathcal{P}|$ clusters of size one each, we successively merge those that have a distance of less than $S$ with regard to $d_2$. This is necessary to ensure a good separation value. In the second phase (we could call this the Hochbaum-Shmoys phase) we then reduce the number of clusters further, while ensuring that the diameter with regards to $d_1$ does not grow too large.

*Proof of Theorem 7.* Let us first consider what happens when Algorithm 1 fails and returns an error message. Clearly, this only happens when we reach line 5 or line 9. In any either case, we have to show that there cannot exist a clustering $\mathscr{C}^*$ with $\mathsf{diam}(\mathscr{C}^*, d_1) \leq D$ and $\mathsf{sep}(\mathscr{C}^*, d_2) \geq S$. We show that the assumption that such a clustering $\mathscr{C}^*$ exists yields a contradiction.

Consider the first case where Algorithm 1 reaches line 5. In lines 1 to 3 we have, starting with a separate cluster for each point, merged all clusters that are at a distance of at most $S$ from each other.

(Apart from the order in which the clusters get merged, this is the same as running the Single Linkage algorithm on $(\mathcal{P}, d_2)$.) Note, that a clustering in which any one such merge has not been carried out has to have a separation value strictly less than $S$, because there would exist two points $x, y \in \mathcal{P}$, contained in different clusters, with $d_2(x, y) < S$. In other words, to ensure a separation value of at least $S$, every merge performed so far has been obligatory and so, if we reach line 5, we know that $\mathscr{C}^*$ cannot exist.

Next, consider the other case in which Algorithm 1 reaches line 9. Within the second phase, we are trying to reduce the number of clusters, while maintaining a small diameter. To achieve this we have constructed a graph $G_{\leq D}$ in lines 6 to 7 whose edges indicate viable and cheap merges. That is, edges in this graph represent the only merges that could in fact be made without violating the requirements imposed on $\mathscr{C}^*$. They represent the only merges whose resulting clusters have a diameter of at most $D$. Now, once we reach line 9, we know that this graph contains an independent set $\mathscr{I}$ of size $\ell > k$. As outlined above, all merges so far have been obligatory and so it follows that for each $C \in \mathscr{I}$ there exists some $C^* \in \mathscr{C}^*$ with $C \subseteq C^*$. Therefore, by the pigeonhole principle there has to be a cluster $C^*$ that contains two different clusters from $\mathscr{I}$. But $\mathscr{I}$ is an independent set in $G_{\leq D}$ and so $\mathsf{diam}(\mathscr{C}^*, d_1) \geq \mathsf{diam}(C^*, d_1) > D$. Again $\mathscr{C}^*$ with the desired properties does not exist. This concludes the first part of the proof.

What's left to show is that, whenever Algorithm 1 returns a clustering, it in fact returns a good clustering. In the first phase we have constructed a clustering $\mathscr{C}^*$ with $\mathsf{sep}(\mathscr{C}^*, d_2) \geq S$ and $\mathsf{diam}(\mathscr{C}^*, d_1) \leq D$, but potentially $|\mathscr{C}^*| > k$. By subsequently merging clusters within star-graphs induced by an independent set within $G_{\leq D}$, we are able to resolve this problem. Of course, none of the merges can decrease the separation factor, so we only have to consider how the diameter changes. This, however, is just a straight-forward application of the triangle inequality: every star-graph clearly has a diameter of at most $2D$. □

**Corollary 8.** *Given a finite set $\mathcal{P}$ and metrics $d_1, d_2$ on $\mathcal{P}$. We can compute the approximate Pareto set $\mathbb{P}_\alpha$ for $\alpha = (1, 2)$ with respect to the objectives $f_1 = \mathsf{sep}$ and $f_2 = \mathsf{diam}$ in polynomial time.*

Running Algorithm 1 with a separation value of $S = 0$ (or $S < \min\{d_2(x, y) \mid x \neq y \in \mathcal{P}\}$) amounts to finding a 2-approximate solution for the standard $k$-diameter problem. Since it is known that there does not exist a polynomial-time $(2 - \varepsilon)$-approximation algorithm for this problem, unless $\mathsf{P} = \mathsf{NP}$ [25], we know that the $(1, 2)$-factor established in Theorem 7 cannot be improved.

### A.2.2  Combination of $k$-Center with $k$-Separation

We can also combine the $k$-separation objective function with the $k$-center objective function. Since the latter is closely related to the $k$-diameter objective function, most analyses here will be similar to those of the previous section. We focus mostly on the subroutine because that is the only part that qualitatively changes. Phase 2 of Algorithm 1 has to be adjusted to ensure that the following theorem, analogous to Theorem 7, holds.

**Theorem 9.** *Let $S, R \in \mathbb{R}$ be two non-negative numbers. If there exists a $k$-center solution $\mathscr{C}^*$ with $\mathsf{rad}(\mathscr{C}^*, d_1) \leq R$ and $\mathsf{sep}(\mathscr{C}^*, d_2) \geq S$, then Algorithm 2 computes a solution $\mathscr{C}$ in polynomial time with $\mathsf{rad}(\mathscr{C}, d_1) \leq 2R$ and $\mathsf{sep}(\mathscr{C}, d_2) \geq S$. If no such clustering $\mathscr{C}^*$ exists, Algorithm 2 returns an error message.*

Algorithm 2 shows the modified subroutine. The construction of our merging graph in the second phase now depends on the optimal centers of the previously computed clusters.

*Proof of Theorem 9.* Again, we start the proof by analyzing the points of failure and show that the existence of a solution $\mathscr{C}^*$ satisfying the bounds in the theorem leads to a contradiction. The first case, where Algorithm 2 return an error message at line 5 can be handled in exactly the same way as in the original proof and so we won't repeat it here. Instead, consider the case, where Algorithm 2 returns an error message in line 9. As before, all merges up to this point had been obligatory. Leaving out any such merge would yield a clustering that does not satisfy the lower bound for the separation. We thus know that the clustering $\mathscr{C}$ is in fact just a refinement of the clustering $\mathscr{C}^*$, which means that there have to exist two clusters $C, C' \in \mathscr{I}$ that are both subsets of the same cluster $C^* \in \mathscr{C}^*$. But then $C^*$ has to have a radius of $\mathsf{rad}(C^*, d_1) \geq \frac{1}{2}\mathsf{diam}(C \cup C') > R$, contradicting our assumptions.

**Algorithm 2:** Approximate solution for $k$-center and $k$-separation

**input** : A finite set $\mathcal{P}$, two metrics $d_1, d_2$ on $\mathcal{P}$, a number $k \in \mathbb{N}$, two values $R, S \in \mathbb{R}_{\geq 0}$.

**1** $\mathscr{C} \leftarrow \{\{x\} \mid x \in \mathcal{P}\}$
**2 while** $\exists C, C' \in \mathscr{C}, \ C \neq C' : \ d_2(C, C') < S$ **do**
**3** $\quad \lfloor \ \mathscr{C} \leftarrow (\mathscr{C} \setminus \{C, C'\}) \cup \{C \cup C'\}$
**4 if** $R < \mathsf{rad}(\mathscr{C}, d_1)$ or $|\mathscr{C}| < k$ **then**
**5** $\quad \lfloor$ **return** *no such clustering exists*
**6** $G_{\leq 2R} \leftarrow (\mathscr{C}, \{\{C, C'\} \mid \mathsf{diam}(C \cup C', d_1) \leq 2R\})$
**7** $\mathscr{I} = \{I_1, \ldots, I_\ell\} \leftarrow$ maximal independent set in $G_{\leq 2R}$
**8 if** $\ell > k$ **then**
**9** $\quad \lfloor$ **return** *no such clustering exists*
**10** partition $G_{\leq 2R}$ into star-graphs $S_1, \ldots S_\ell$ centered around $I_1, \ldots I_\ell$
**11 forall** $i \in \{1, \ldots, \ell\}$ **do**
**12** $\quad D = \emptyset$
**13** $\quad$ **forall** $C \in S_i$ **do**
**14** $\quad\quad$ **if** $|\mathscr{C}| = k$ **then**
**15** $\quad\quad\quad \lfloor$ **return** $\mathscr{C}$
**16** $\quad\quad \mathscr{C} \leftarrow (\mathscr{C} \setminus \{D\}) \cup \{D \cup C\}$
**17** $\quad\quad D \leftarrow D \cup C$
**18 return** $\mathscr{C}$

Finally, we prove that any clustering returned by the algorithm is in fact a good solution. That the lower bound for the separation value is satisfied is, again, obvious. Let us thus move on to the the upper bound on the radius. After the first phase all clusters necessarily have a radius of at most $R$ (otherwise the algorithm would fail in line 5). To see that the second phase does not increase the radius by much, fix $z_I \in I$ for every cluster $I \in \mathscr{I}$. For every $I \in \mathscr{I}$ all other clusters $C$ belonging to the star graph $S$ of $I$ are close to $I$ in the sense that $\mathsf{diam}(I \cup C) \leq 2R$. In particular $d_1(z_I, x) \leq 2R$ for all $x \in C$, implying that, indeed, $\mathsf{rad}(\bigcup_{C \in S} C) \leq 2R$. $\qquad \square$

**Corollary 10.** *Given a finite set $\mathcal{P}$ and metrics $d_1, d_2$ on $\mathcal{P}$. We can compute the approximate Pareto set $\mathbb{P}_\alpha$ for $\alpha = (1, 2)$ with respect to the objectives $f_1 = \mathsf{sep}$ and $f_2 = \mathsf{rad}$ in polynomial time.*

Again, the $(1, 2)$-approximation factor cannot be improved. For $S = 0$, Algorithm 2 computes 2-approximate solutions for $k$-center, which is the lowest approximation ratio achievable in polynomial time [38], unless $\mathsf{P} = \mathsf{NP}$.

### A.3 Combination of $k$-separation and $k$-median/$k$-means

The instance outlined in Example 6 can again be used to show that there is no hope of finding a clustering that performs well with regard to both the $k$-median (or $k$-means) and the $k$-separation objective.

**Observation 11.** *Consider a set of points $\{x/m \mid x \in \{0, \ldots, m\ell\}\}$ for any $\ell$ and even $m$. The 1-median cost of this set is given by*

$$\sum_{x=0}^{m\ell} \left| \frac{x}{m} - \frac{\ell}{2} \right| = 2 \sum_{x=0}^{\frac{m\ell}{2}-1} \left( \frac{\ell}{2} - \frac{x}{m} \right) = \frac{2}{m} \sum_{x=0}^{\frac{m\ell}{2}-1} \left( \frac{m\ell}{2} - x \right)$$

$$= \frac{2}{m} \sum_{x=1}^{\frac{m\ell}{2}} x = \frac{1}{4} \ell(m\ell + 2)$$

*since the points are evenly spaced and thus the optimal center is $\ell/2$. Now, if we consider the same set*

$$\mathcal{P}_{k,m} = \underbrace{\{-x/m \mid x \in \{0, \ldots, m(k-1)^2\}\}}_{A_{k,m}} \cup \{1, \ldots, k-1\}$$

*as in Example 6, then we run into the same issues. By the above computation, $A_{k,m}$ has a 1-median cost that is in $\Omega(k^4)$, whereas the subsets that we get from dividing $[-(k-1)^2, k-1]$ into $k$ intervals of equal length each have a 1-median cost that is in $O(k^2)$. The $k$-median cost of the only clustering $\mathscr{C} = \{\{1\}, \ldots, \{k-1\}, A_{k,m}\}$ that achieves a constant-factor approximation ratio with regard to* sep *is worse by a factor of $\Omega(k)$ than that of the optimum. Conversely, if we split up $A_{k,m}$ to achieve a good $k$-median solution, then the separation value will be bad.*

*Thus, as before, we have shown that all clusterings can have a good (i.e. constant-factor) approximation ratio either with regard to the $k$-median objective or with regard to the $k$-separation objective, but not both simultaneously.*

Similar to the case of the $k$-center objective the value of the $k$-separation function is determined by two of the points in $\mathcal{P}$, we know that for the same reasons the Pareto set for this combination consists of at most $\frac{n(n-1)}{2}$ different solutions.

**Theorem 12.** *Given a finite set $\mathcal{P}$, two metrics $d_1, d_2$ and objectives $f_1 = $ sep *and* $f_2 = $ med. *Then we can compute the $\alpha$-approximate Pareto set $\mathbb{P}_\alpha$ for $\alpha = (1, 2 + \delta)$ with respect to $f_1, f_2$ in polynomial time, where $\delta$ is the best approximation bound achievable for the $k$-median problem.*

The following corollary follows directly from the above theorem and the fact that Cohen-Addad et al. [16] presented a $(2.67059 + \epsilon)$-approximation algorithm for the $k$-median problem.

**Corollary 13.** *We can compute the approximate Pareto set $\mathbb{P}_\alpha$ for $\alpha = (1, 4.67059 + \epsilon)$ with respect to the objectives $f_1 = $ sep *and* $f_2 = $ med *in polynomial time.*

In order to prove the theorem, we use the nesting-technique introduced for hierarchical clustering by Lin et al. [46].

**Definition 14** (adapted from [46])**.** *We say that a clustering objective $f_1 \in \{$med, mean$\}$ admits the $(\gamma, \delta)$-nesting property if the following holds for every clustering instance. Let $\mathcal{P}$ be a finite set of points, $d$ a metric on $\mathcal{P}$ and $k \in \mathbb{N}$. For every two clusterings $(C, \sigma), (C', \sigma')$ with $|C| = k$ and $|C'| > k$ there exists a clustering $(\widetilde{C}, \widetilde{\sigma})$ with $|\widetilde{C}| = k$ which satisfies the following properties*

1. *for $p, q \in \mathcal{P}$ with $\sigma'(p) = \sigma'(q)$ we have $\widetilde{\sigma}(p) = \widetilde{\sigma}(q)$ and*

2. $\mathsf{med}(\widetilde{C}, \widetilde{\sigma}, d) \leq \gamma \cdot \mathsf{med}(C', \sigma', d) + \delta \cdot \mathsf{med}(C, \sigma, d)$.

*We call $(\widetilde{C}, \widetilde{\sigma})$ a $(\gamma, \delta)$-nesting with respect to solutions $(C, \sigma)$ and $(C', \sigma')$.*

The first property guarantees that points which are contained in the same cluster in $(C', \sigma')$ stay in the same cluster in $(\widetilde{C}, \widetilde{\sigma})$ and the second property guarantees that the cost of $(\widetilde{C}, \widetilde{\sigma})$ can be bounded in terms of the cost of the other two solutions. Lin et al. [46] show that $k$-median admits the $(2, 1)$-nesting property. For $k$-means in the Euclidean space Großwendt [7] shows that $k$-means admits the $(4, 2)$-nesting property.

For $k$-median we follow the approach for the $(2, 1)$-nesting by Lin et al. [46] to compute the approximate Pareto set $\mathbb{P}_{(1, 2+\delta)}$ with respect to $f_1 = $ sep and $f_2 = $ med.

**Lemma 15** ([46])**.** *The $k$-median objective admits the $(2, 1)$-nesting property.*

*Proof.* Let $\mathcal{P}$ be a finite set of points, $d$ a metric on $\mathcal{P}$ and $k \in \mathbb{N}$. Given two clusterings $(C, \sigma)$ and $(C', \sigma')$ with $|C| = k$ and $|C'| > k$ we want to construct a clustering $(\widetilde{C}, \widetilde{\sigma})$ which is a $(2, 1)$-nesting with respect to solutions $(C, \sigma)$ and $(C', \sigma')$.

We now construct our solution by setting $\widetilde{C} = C$. For $p \in \mathcal{P}$ we define the assignment as follows

$$\widetilde{\sigma}(p) = \underset{c \in \widetilde{C}}{\operatorname{argmin}} \, d(c, \sigma'(p)).$$

Now notice that $(\widetilde{C}, \widetilde{\sigma})$ satisfies $|\widetilde{C}| = k$. Furthermore we have for points $p, q \in \mathcal{P}$ with $\sigma'(p) = \sigma'(q)$ that $\widetilde{\sigma}(p) = \widetilde{\sigma}(q)$.

To bound the $k$-median cost, we first bound the distance of each point $p$ to its center:

$$d(p, \widetilde{\sigma}(p)) \leq d(p, \sigma'(p)) + d(\sigma'(p), \widetilde{\sigma}(p))$$
$$\leq d(p, \sigma'(p)) + d(\sigma'(p), p) + d(p, \sigma(p))$$
$$\leq 2 \cdot d(p, \sigma'(p)) + d(p, \sigma(p))$$

The first inequality holds due to the triangle inequality. The second inequality holds due to

$$d(\sigma'(p), \widetilde{\sigma}(p)) = \min_{c \in \widetilde{C}} d(\sigma'(p), c) \leq d(\sigma'(p), \sigma(p)) \leq d(\sigma'(p), p) + d(p, \sigma(p)).$$

Taking the sum over all points gives us:

$$\sum_{p \in \mathcal{P}} d(p, \widetilde{\sigma}(p)) \leq \sum_{p \in \mathcal{P}} (2 \cdot d(p, \sigma'(p)) + d(p, \sigma(p)))$$
$$= 2 \cdot \mathsf{med}(C', \sigma', d) + \mathsf{med}(C, \sigma, d).$$

This concludes the proof. $\qquad\square$

We now apply this lemma to compute the approximate Pareto set for $f_1 = \mathsf{sep}$ and $f_2 = \mathsf{med}$.

**Lemma 16.** *Given the minimal distance between two clusters $D > 0$. Let $(C^*, \sigma^*)$ be the optimal $k$-median solution with respect to $d_2$ that fulfills $\mathsf{sep}(C^*, \sigma^*, d_1) \geq D$. Then we can compute a solution $(C, \sigma)$ with $|C| = k$ such that all clusters have at least distance $D$ to each other with respect to $d_1$, and $\mathsf{med}(C, \sigma, d_2) \leq (2 + \delta) \cdot \mathsf{med}(C^*, \sigma^*, d_2)$ in polynomial time or report that there is no such clustering.*

*Proof.* Since we have given the minimal distance between clusters $D$, we know that all points that have distance smaller than $D$ to each other must lie in the same cluster of $(C^*, \sigma^*)$. Therefore we can construct a clustering $(C_S, \sigma_S)$ that greedily clusters all of these points together.

Note that if there exists a solution that fulfills our distance requirement then $|C_S| \geq k$ since all of these groups would have to be present in $(C^*, \sigma^*)$ which is a $k$-clustering. If not, then $D$ is chosen too large.

If $|C_S| = k$, then it must hold that $(C_S, \sigma_S) = (C^*, \sigma^*)$ and we are done. Therefore we assume for the rest of the proof that $|C_S| > k$.

It is clear that all points that are clustered together in $(C_S, \sigma_S)$ must be clustered together in our solution $(C, \sigma)$, because if there are points $p, q$ with $\sigma_S(p) = \sigma_S(q)$ but $\sigma(p) \neq \sigma(q)$ we have $d_1(p, q) < D$ and therefore $\mathsf{sep}(C, \sigma) < D$. Therefore the solution $(C, \sigma)$ must be compatible with $(C_S, \sigma_S)$ and at the same time ensure that the $k$-median cost with respect to $d_2$ is not too high. To ensure this, we denote by $(C_M, \sigma_M)$ a $\delta$-approximation to the best $k$-median solution (w.r.t. $d_2$). Therefore $(C_M, \sigma_M)$ is also a $\delta$-approximation to $(C^*, \sigma^*)$.

Since the separation cost function is independent of the set of centers $C_S$ we can without loss of generality choose the center set to be the set of points that minimizes $\mathsf{med}(C_S, \sigma_S, d_2)$. Since points which are in the same cluster in $(C_S, \sigma_S)$ have to be in the same cluster in $(C^*, \sigma^*)$ as well, we know

$$\mathsf{med}(C_S, \sigma_S, d_2) \leq \mathsf{med}(C^*, \sigma^*, d_2). \qquad (1)$$

Now let $(C, \sigma)$ be a clustering which is a $(2, 1)$-nesting with respect to $(C_M, \sigma_M)$ and $(C_S, \sigma_S)$. Such a clustering exists by Lemma 15.

For all $p, q$ with $d_1(p, q) < D$ it holds that $\sigma_S(p) = \sigma_S(q)$ and therefore also $\sigma(p) = \sigma(q)$. Thus we obtain $\mathsf{sep}(C, \sigma, d_1) \geq D$. For the $k$-median cost with respect to $d_2$ we obtain

$$\mathsf{med}(C, \sigma, d_2) \leq 2 \cdot \mathsf{med}(C_S, \sigma_S, d_2) + \mathsf{med}(C_M, \sigma_M, d_2)$$
$$\leq 2 \cdot \mathsf{med}(C^*, \sigma^*, d_2) + \delta \cdot \mathsf{med}(C^*, \sigma^*, d_2)$$
$$= (2 + \delta) \cdot \mathsf{med}(C^*, \sigma^*, d_2),$$

where the last inequality holds because of inequality (1). This concludes the proof. $\qquad\square$

The theorem now follows from the fact that there are only $\frac{n(n-1)}{2}$ many possible values for $D$ and we can test all of them in polynomial time. Note that in the last steps of the proof of Lemma 16 we say $\mathsf{med}(C_M, \sigma_M, d_2) \leq \delta \cdot \mathsf{med}(C^*, \sigma^*, d_2)$. The same guarantee holds also with the optimal $k$-median clustering instead of the Pareto-optimal solution $(C^*, \sigma^*)$, and in cases where the Pareto-optimal solution has bad values for the $k$-median target function, this might be a big advantage.

The combination of $k$-separation and $k$-means works exactly the same way, with the exception that we cannot simply use the triangle inequality in the analysis, leading to the following theorem.

**Theorem 17.** *Given a finite set $\mathcal{P}$ and two metrics $d_1, d_2$ and objectives $f_1 = \mathsf{sep}$ and $f_2 = \mathsf{mean}$. Then we can compute the $\alpha$-approximate Pareto set $\mathbb{P}_\alpha$ for $\alpha = (1, 4 + 4 \cdot \sqrt{\delta} + \delta)$ with respect to $f_1, f_2$ in polynomial time, where $\delta$ the best approximation bound achievable for the $k$-means problem with respect to $d_2$.*

The following corollary follows directly from the above theorem and the fact that Ahmadian et al. [4] presented a $(9 + \epsilon)$-approximation algorithm for the $k$-means problem in general metrics.

**Corollary 18.** *We can compute the approximate Pareto set $\mathbb{P}_\alpha$ for $\alpha = (1, 25 + \epsilon)$ with respect to the objectives $f_1 = \mathsf{sep}$ and $f_2 = \mathsf{mean}$ in polynomial time.*

We now follow the approach that Lin et al. [46] used to get a $(8, 2)$-nesting, but without setting $\tau = 2$ at the end of the proof, which leads to the following more general statement:

**Lemma 19** ([46]). *The $k$-means objective admits the $(4 + \tau^2, 1 + \frac{4}{\tau^2})$-nesting property for all $\tau > 0$.*

*Proof.* Analogously to the proof of Lemma 15 we get:

$$
\begin{aligned}
d(p, \widetilde{\sigma}(p))^2 &\leq (2 \cdot d(p, \sigma'(p)) + d(p, \sigma(p)))^2 \\
&= 4 \cdot d(p, \sigma'(p))^2 + 4 \cdot d(p, \sigma'(p))d(p, \sigma(p)) + d(p, \sigma(p))^2 \\
&\leq (4 + \tau^2) \cdot d(p, \sigma'(p))^2 + (1 + \frac{4}{\tau^2}) \cdot d(p, \sigma(p))^2
\end{aligned}
$$

for all $\tau > 0$. The last inequality follows from the fact that $4ab \leq \tau^2 \cdot a^2 + \frac{4}{\tau^2} \cdot b^2$ for all $a, b, \tau > 0$. Summing up over all points leads to:

$$
\begin{aligned}
\sum_{p \in \mathcal{P}} d(p, \widetilde{\sigma}(p))^2 &\leq \sum_{p \in \mathcal{P}} \left( (4 + \tau^2) \cdot d(p, \sigma'(p))^2 + (1 + \frac{4}{\tau^2}) \cdot d(p, \sigma(p))^2 \right) \\
&= (4 + \tau^2) \cdot \mathsf{mean}(C', \sigma', d) + (1 + \frac{4}{\tau^2}) \cdot \mathsf{mean}(C, \sigma, d).
\end{aligned}
$$

$\square$

In the following lemma, we can now choose the value of $\tau$ in relation to the available $k$-means approximation factor $\delta$ in order to improve our approximation factor.

**Lemma 20.** *Given the minimal distance between two clusters $D > 0$. Let $(C^*, \sigma^*)$ be the optimal $k$-means solution with respect to $d_2$ that fulfills $\mathsf{sep}(C^*, \sigma^*, d_1) \geq D$. Then we can compute a solution $(C, \sigma)$ with $|C| = k$ such that all clusters have at least distance $D$ to each other with respect to $d_1$, and $\mathsf{mean}(C, \sigma, d_2) \leq (4 + 4\sqrt{\delta} + \delta) \cdot \mathsf{mean}(C^*, \sigma^*, d_2)$ in polynomial time or report there is no such clustering.*

*Proof.* Both the algorithm and the analysis for $\mathsf{sep}(C^*, \sigma^*, d_1) \geq D$ works exactly as in the proof of Lemma 16. Let $(C_M, \sigma_M)$ and $(C_S, \sigma_S)$ be defined analogously to the proof of Lemma 16.

Let $(C, \sigma)$ be a clustering which is a $(4 + 2\sqrt{\delta}, 1 + \frac{2}{\sqrt{\delta}})$-nesting with respect to $(C_M, \sigma_M)$ and $(C_S, \sigma_S)$. Such a clustering exists by setting $\tau^2 = 2\sqrt{\delta}$ in Lemma 19.

For all $p, q$ with $d_1(p, q) < D$ it holds that $\sigma_S(p) = \sigma_S(q)$ and therefore also $\sigma(p) = \sigma(q)$. Thus we obtain $\mathsf{sep}(C, \sigma, d_1) \geq D$.

For the $k$-means cost with respect to $d_2$ we obtain

$$\mathsf{mean}(C, \sigma, d_2) \leq (4 + 2\sqrt{\delta}) \cdot \mathsf{mean}(C_S, \sigma_S, d_2) + (1 + \frac{2}{\sqrt{\delta}}) \cdot \mathsf{mean}(C_M, \sigma_M, d_2)$$

$$\leq (4 + 2\sqrt{\delta}) \cdot \mathsf{mean}(C^*, \sigma^*, d_2) + \delta(1 + \frac{2}{\sqrt{\delta}}) \cdot \mathsf{mean}(C^*, \sigma^*, d_2)$$

$$= (4 + 2\sqrt{\delta}) \cdot \mathsf{mean}(C^*, \sigma^*, d_2) + (\delta + 2\sqrt{\delta}) \cdot \mathsf{mean}(C^*, \sigma^*, d_2)$$

$$= (4 + 4\sqrt{\delta} + \delta) \cdot \mathsf{mean}(C^*, \sigma^*, d_2),$$

where the last inequality holds because $(C_M, \sigma_M)$ is a $\delta$-approximation to the optimal $k$-means solution. This concludes the proof.

$\square$

### A.4  Combining the $k$-MSR problem and $k$-Separation problem

In this section, we combine the $k$-separation and the $k$-msr objectives. Notice that we here allow that clusterings have less than $k$ clusters.

The $k$-msr objective differs from other center-based objectives on a fundamental level. Whereas it is always optimal with regard to the $k$-center, the $k$-median, or the $k$-means objectives to assign points to their closest center, the same does not hold for the $k$-msr objective. Even on a line graph and even when the centers are chosen optimally, assigning points to their closest centers can result in a clustering whose $k$-msr cost is three times the cost of an optimal solution [45], and for more complicated metric spaces this factor grows quickly. In particular, the black-box approach that we have used in the previous two sections is not applicable here because it relies heavily on this property. Instead, we will work through the explicit primal-dual algorithm of Buchem et al. [11] and show that it still works if we replace single points with preformed clusters. Before that, however, let us quickly ascertain, as before, that these two objectives cannot be approximated well simultaneously.

**Observation 21.** *The example that we now provide does not turn out to be as bad as the ones given in Example 6 or Example 11. This is because we can often merge clusters that are close to each other without increasing the $k$-msr cost by much. If we were to consider the other examples, then we would quickly find that the same arguments do not hold. For example, in the $k$-diameter case, we only had to compare the clustering*

$$\mathcal{C} = \{\{1\}, \dots, \{k-1\}, A_{k,m}\}$$

*with one that evenly partitions $\mathcal{P}_{k,m}$ as a subset of $[-(k-1)^2, k-1]$ into $k$ sets of equal length $k-1$, to deduce that $\mathcal{C}$, which is the only constant-factor approximation with regard to the $k$-separation objective, is only $\Omega(k)$-approximative with regard to the $k$-diameter objective. However, with regard to the $k$-msr objective we would have to sum up all the radii of the second partitioning and so end up with clustering whose is cost is also in the order of $\Omega(k^2)$. In other words, this example does not show that $\mathcal{C}$ is a bad clustering with regard to the $k$-msr objective. Instead we consider a graph metric consisting of two parts:*

1. *a line graph consisting of $k - 1$ points, where all edges have weight 1;*

2. *a clique consisting of $k - 1$ points, where all edges have weight $\sqrt{k}$.*

*Each distance $d(x, y)$ on the union of both graphs is given as the weight of a shortest path between $x$ and $y$. For there to always exist such a path we can just add a single edge somewhere between the first and the second part with arbitrarily large weight. Now let us consider the only constant-factor approximation with regard to the $k$-separation objective. As before, this clustering results from merging all points in the line graph and leaving all points in the clique separate. Its separation value is $\sqrt{k}$, whereas every other clustering has to have a separation value of 1 because it has to split up the line graph. However, the $k$-msr cost of this solution is $\lceil \frac{k-1}{2} \rceil$, which is worse by a factor of $\sqrt{k}$ than merging the clique and leaving each point in the line graph separate, as the latter has a $k$-msr cost of exactly $\sqrt{k}$.*

With this established, let us find a convenient formulation of the $k$-mim-sum-radii problem to be used in this section. We first recall the notion of balls around points.

**Definition 22.** *Let $(\mathcal{P}, d)$ be a finite metric space. For a pair $(c, r) \in \mathcal{P} \times \mathbb{R}$, let $B_{(\mathcal{P},d)}(c, r) = \{x \in \mathcal{P} \mid d(x, c) \leq r\}$ denote the set of points in $\mathcal{P}$ that are at distance at most $r$ from $c$.*

In the introduction, we said that the solution of an instance $(\mathcal{P}, d)$ of the $k$-msr problem is an assignment function $\sigma : \mathcal{P} \to C$ from $\mathcal{P}$ to a subset $C \subset \mathcal{P}$ that consists of $k$ centers. The stated goal was to find a $\sigma$ that minimizes $\sum_{c \in C} \max_{x \in \sigma^{-1}(c)} d(x, c)$. Since the cost of a single cluster is determined only by two points, we can conceptualize solutions slightly differently. Instead of an assignment we are looking for $k$ pairs $(c_1, r_1), \ldots, (c_k, r_k) \in \mathcal{P} \times \mathbb{R}$ that minimize $\sum_i r_i$ and that guarantee that $\mathcal{P} \subseteq \bigcup_i B_{(\mathcal{P},d)}(c_i, r_i)$. Each such pair induces a (not necessarily unique) assignment by mapping each point $x \in \mathcal{P}$ to a center $c_i$ with $d(x, c_i) \leq r_i$. We will work with this formulation in the remainder of this section. The following shows how we wish to combine the $k$-separation and $k$-msr objective functions.

**Definition 23** (The $\tau$-separated $k$-msr problem)**.** *Let $d_1$, $d_2$ be two metrics over a finite set $\mathcal{P}$, $k \in \mathbb{N}$, and $\tau \geq 0$. The goal in the $\tau$-separated $k$-msr problem is to find $k$ pairs $P = \{(c_1, r_1), \ldots, (c_k, r_k)\}$ that minimize $\mathrm{cost}(P) = \sum_{i=1}^{k} r_i$ and that guarantee that for all $S \in \mathscr{S}_{(\mathcal{P},d_1)}(\tau)$ there exists some $i_S \in \{1, \ldots, k\}$ with $S \subseteq B_{(\mathcal{P},d_2)}(c_{i_S}, r_{i_S})$.*

Instead of starting with individual points, we compute the minimally $\tau$-separated clustering in a preprocessing step and then only subsequently merge clusters in this clustering to reach $k$ clusters. By slightly adjusting the approach by Buchem et al. [11] we get the following result.

**Theorem 24.** *There exists a $(3 + \varepsilon)$-approximation algorithm for the $\tau$-separated $k$-msr problem that runs in $n^{O(1/\varepsilon)}$ time.*

The approximation factor is exactly the same as in the original paper. Before getting into the details of Buchem et al.'s primal-dual algorithm we have to note an essential observation established by Buchem et al.

**Definition 25.** *Let $(\mathcal{P}, d)$ be a finite metric space and $\mathcal{Q} \subset \mathcal{P} \times \mathbb{R}$ a finite set of pairs. The intersection-graph $G_{(\mathcal{P},d)}(\mathcal{Q})$ of $\mathcal{Q}$ consists of*

- *vertices $V_{(\mathcal{P},d)}(\mathcal{Q}) = \mathcal{Q}$ and*

- *edges $E_{(\mathcal{P},d)}(\mathcal{Q}) = \{\{(c, r), (c', r')\} \mid B_{(\mathcal{P},d)}(c, r) \cap B_{(\mathcal{P},d)}(c', r') \neq \emptyset\}$.*

One of the most interesting properties of the $k$-msr objective is that the cost of a solution can sometimes be reduced by merging clusters (which cannot happen with the other center-based objectives). For example, this is the case whenever $c_i \in B_{(\mathcal{P},d)}(c_j, r_j)$, i.e., when the center of one pair is contained within the ball of another pair. The following result of Buchem et al. shows that we can merge sets of intersecting clusters without introducing too much additional cost.

**Lemma 26** (Cf. Appendix C of [11])**.** *Let $(\mathcal{P}, d)$ be a finite metric space, $\mathcal{Q} \subseteq \mathcal{P} \times \mathbb{R}$ a finite set of pairs whose intersection-graph $G_{(\mathcal{P},d)}(\mathcal{Q})$ is connected, and $I \subset V_{(\mathcal{P},d)}(\mathcal{Q})$ an independent set that maximizes $\sum_{(c,r) \in I} r$. Then there exists a pair $(c^*, r^*) \in \mathcal{P} \times d(\mathcal{P} \times \mathcal{P})$ such that*

1. *$\bigcup_{(c,r) \in \mathcal{Q}} B_{(\mathcal{P},d)}(c, r) \subset B_{(\mathcal{P},d)}(c^*, r^*)$ and*

2. *$r^* \leq 3 \sum_{(c,r) \in I} r$.*

The proof of this result is dispersed over several technical results contained in Appendix C of [11]. Since it does not concern our new constraints, we won't repeat it here. However, with this taken care of, we are now in the position to work through the actual primal-dual algorithm. Fix an arbitrary instance (a finite set $\mathcal{P}$, two metrics $d_1$ and $d_2$ over $\mathcal{P}$, a number $k \in \mathbb{N}$, a threshold $\tau \geq 0$) and let $(c_1^*, r_1^*), \ldots, (c_k^*, r_k^*)$ denote the pairs of optimal solution $\mathcal{P}$ for the $\tau$-separated $k$-msr problem in non-increasing order of their radii.

The first step is common to all primal-dual approaches for the $k$-msr problem ([13, 28, 11]) and consists of guessing the $1/\varepsilon$ largest pairs $(c_1^*, r_1^*), \ldots, (c_{1/\varepsilon}^*, r_{1/\varepsilon}^*)$ of the optimal solution (where we assume w.l.o.g. that $1/\varepsilon \in \mathbb{N}$). This can be done in $n^{O(1/\varepsilon)}$ time and leaves us to only have to approximate the remaining $k' = k - 1/\varepsilon$ pairs $(c_{1/\varepsilon}^*, r_{1/\varepsilon}^*), \ldots, (c_k^*, r_k^*)$. The pairs comprising the optimal solution can be of drastically different size, but this segmentation ensures that all leftover

clusters
$$\mathscr{S}' = \{S \in \mathscr{S}_{(\mathcal{P},d_1)}(\tau) \mid \forall i \in \{1,\ldots,1/\varepsilon\}\colon S \not\subseteq \mathrm{B}_{(\mathcal{P},d_2)}(c_i, r_i)\}$$

can be covered by balls of radii at most $r_{1/\varepsilon} \leq \varepsilon \cdot \mathrm{cost}(\mathcal{P})$. This will give us some breathing room later on. From now on we can focus on the $\tau$-separated $k'$-msr problem for $\mathcal{P}' = \bigcup_{S \in \mathscr{S}'} S$.

For the second step of the algorithm, let us set up the primal and dual LP corresponding to the problem. While they look quite similar to those for the vanilla $k$-msr problem, conceptually they are slightly different. While Buchem et al. only have to cover individual points, we have to ensure that clusters from $\mathscr{S}' = \{S_1,\ldots,S_\ell\}$ are always covered as a whole.

**Primal LP:**

$$\text{minimize} \qquad \sum_{(c,r)} r \cdot x_{(c,r)}$$

$$\text{subject to} \quad \begin{array}{rcl} \sum_{(c,r)\colon S_j \subset \mathrm{B}_{(\mathcal{P},d_2)}(c,\,r)} x_{(c,r)} & \geq & 1 \quad \forall j \in \{1,\ldots,\ell\} \\ \sum_{(c,r)} x_{(c,r)} & \leq & k' \\ x_{(c,r)} & \geq & 0 \end{array}$$

where we have a variable $x_{(c,r)}$ for every pair $(c,r)$ consisting of a center $c \in \mathcal{P}$ and a radius $r \in R := \{d(x,y) \mid x,y \in \mathcal{P}\colon d(x,y) \leq r^*_{1/\varepsilon}\}$. The corresponding dual LP is as follows:

**Dual LP:**

$$\text{maximize} \qquad \sum_j \alpha_j - \lambda k'$$

$$\text{subject to} \quad \begin{array}{rcl} \sum_{j\colon\, S_j \subset \mathrm{B}_{(\mathcal{P},d_2)}(c,\,r)} \alpha_j & \leq & r + \lambda \quad \forall (c,r) \\ \alpha_j & \geq & 0 \qquad \forall j \in \{1,\ldots,\ell\} \\ \lambda & \geq & 0 \end{array}$$

As expected, the second step consists of computing a solution for the dual LP that can almost fully pay for a solution of the original $\tau$-separated $k$-msr problem. While this is the standard for primal-dual algorithms, Buchem et al.'s specific approach for the vanilla versions differs significantly from the earlier approaches of both Charikar and Panigrahy [13], as well as Friggstad and Jamshidian [28].

**Definition 27.** *Let $(\alpha,\lambda)$ be a solution for the dual LP and $\mu = \frac{r^*_{1/\varepsilon}}{|\mathcal{P}|^2}$. A pair $(c,r) \in \mathcal{P} \times R$ is* almost tight, *if the corresponding dual constraint is tight up to an additive factor of $\mu$, i.e., if*

$$\sum_{j\colon\, S_j \subset \mathrm{B}_{(\mathcal{P},d_2)}(c,r)} \alpha_j \geq r + \lambda - \mu.$$

Starting with the trivial solution $\alpha_1 = \ldots = \alpha_\ell = 0$ and $\lambda = 0$, Buchem et al. successively increase the values in such a way that the number of connected components of the intersection-graph of almost tight pairs decreases while still guaranteeing that the solution stays valid and that every point is contained in the ball of at least one almost tight pair. Roughly speaking, once they reach $k'$ connected components, they're able to derive a good solution for the $k'$-msr problem on $\mathcal{P}'$ with the help of Lemma 26.

The overall approach is the same for us, but we have to modify the invariant to ensure that the clusters $S \in \mathscr{S}'$ are *fully* contained within the ball of at least one tight pair. Now, contrary to the vanilla case, in which the trivial solution satisfies the invariant, because which every point $x \in \mathcal{P}'$ is contained within the respective ball $\mathrm{B}_{(\mathcal{P},d_2)}(x,\,0)$, the trivial solution for our LP does not necessarily satisfy our modified invariant. This is because the balls of radius 0 do not necessarily cover all clusters $S \in \mathscr{S}'$. But this can easily be rectified by successively increasing the $\alpha_j$'s for clusters $S_j$ that are not yet covered at a uniform speed and end when no such cluster remains. At the end of this phase, the invariant will be satisfied and we can continue in the same manner as Buchem et al. Let $\mathcal{T}$ denote the set of currently tight pairs. Since no $\alpha_j$ can be increased on its own, from now on we also have to increase $\lambda$ to offset this. Even then, if there is an (almost) tight pair, whose ball covers two different clusters from $\mathscr{S}'$, then we cannot increase both of them, as this would (quickly) lead to a violation of the respective constraint. Like Buchem et al. we greedily choose a maximal set of $\alpha_j$'s in such

a way that no two corresponding clusters are fully contained within the ball of some pair from $\mathcal{T}$. These variables as well as $\lambda$ are increased uniformly by some value until one additional pair that until then wasn't almost tight becomes tight. Due to the choice of the $\alpha_j$'s, the validity of the solution is guaranteed and the invariant satisfied. As long as the intersection-graph of the updated set $\mathcal{T}$ contains strictly more than $k'$ connected components, we will increase the $\alpha_j$ values of at least $k' + 1$ clusters, which means that the objective value of the solution increases as well. By relating this increase in cost during each iteration to the upper bound $k' r^*_{1/\varepsilon}$ of the primal LP, Buchem et al. prove that the number of possible iterations is upper bounded by $O\left(|\mathcal{P}|^4\right)$ (or rather $O\left(k|\mathcal{P}|^3\right)$). In particular, this shows that the process can be continued until at most $k'$ connected components are left.

From this point onward, the remaining analysis is independent of the newly imposed constraints and so we will not cover it in detail. Just note that the resulting set of almost tight clusters $\mathcal{T}$ satisfies all the requirements that are necessary for us to conclude the proof in the same manner as Buchem et al. have done (see the proof of Lemma 2.3 in [11]).

By considering all thresholds $\tau \in d_1(\mathcal{P} \times \mathcal{P})$ we get the following result on the Pareto set for $f_1 = \mathsf{sep}$ and $f_2 = \mathsf{msr}$.

**Theorem 28.** *Let $d_1, d_2$ be two metrics defined over a finite set $\mathcal{P}$ and $\varepsilon > 0$. Then for $f_1 = \mathsf{sep}$ and $f_2 = \mathsf{msr}$ we can compute an $\alpha$-approximate Pareto set $P_\alpha$ with $\alpha = (1, 3 + \varepsilon)$ in $O(n^{1/\varepsilon})$ time.*

# B  Theoretical Part: Pareto Sets for Combining Minimization Objectives

## B.1  Combination of two $k$-center/$k$-diameter objectives

We will calculate a $(2, 2)$-approximate Pareto set for the combination of two objectives $f_1, f_2 \in \{\mathsf{diam}, \mathsf{rad}\}$. To do this we adapt the algorithm by Hochbaum and Shmoys [37], which computes a 2-approximation for the objectives $\mathsf{rad}, \mathsf{diam}$, to the case where we want to optimize over two such objectives simultaneously.

Remember that we are given two metrics $d_1, d_2$ on $\mathcal{P}$ and want to minimize $f_i$ with respect to $d_i$. Let $\mathcal{R} = \mathcal{R}_1 \times \mathcal{R}_2$, where $\mathcal{R}_i = \{d_i(v, w) \mid v, w \in \mathcal{P}\}$. For any pair of distances $(R_1, R_2)$ let $G_{R_1, R_2}$ denote the graph containing an edge between exactly those pairs of nodes $v, w \in \mathcal{P}$ fulfilling that $d_1(v, w) \leq R_1$ and $d_2(v, w) \leq R_2$. We adjust the algorithm by Hochbaum and Shmoys [37] as follows. Let $\beta_i = 1$ if $f_i = \mathsf{diam}$ and $\beta_i = 2$ if $f_i = \mathsf{rad}$ throughout the remainder of this section. As in the original algorithm we can use the graph $G_{\beta_1 R_1, \beta_2 R_2}$ to test for a pair of distances $(R_1, R_2) \in \mathcal{R}$ whether there exists a solution with cost $(2R_1, 2R_2)$ or decide that there is no solution with cost $(R_1, R_2)$.

**Lemma 29.** *If the graph $G_{\beta_1 R_1, \beta_2 R_2}$ contains an independent set of size greater than $k$ then there does not exist a solution with cost at most $(R_1, R_2)$.*

*Proof.* Let us assume that there exists a solution $(C, \sigma)$ with cost $(R_1, R_2)$. Let $v, w \in \mathcal{P}$ with $\sigma(v) = \sigma(w)$. Observe that $d_i(v, w) \leq 2R_i$ if $f_i = \mathsf{rad}$ and $d_i(v, w) \leq R_i$ if $f_i = \mathsf{diam}$ and thus $v, w$ are connected by an edge in $G_{\beta_1 R_1, \beta_2 R_2}$. Let $I$ be an independent set in $G_{\beta_1 R_1, \beta_2 R_2}$. By the previous observation we know that $I$ cannot contain two points $v, w$ with $\sigma(v) = \sigma(w)$ and therefore $|I| \leq |C| = k$. $\square$

**Lemma 30.** *If $G_{\beta_1 R_1, \beta_2 R_2}$ contains a maximal independent set $I$ of size at most $k$, there exists a solution $(C, \sigma)$ with $C = I$ and cost at most $(2R_1, 2R_2)$.*

*Proof.* For every node $v$ we can simply set $\sigma(v) = v$ if $v \in I$ or $\sigma(v) = c$ for a neighbor $c$ of $v$ in $I$ (there always exists such a neighbor because otherwise $I$ would not be maximal). If $f_i = \mathsf{rad}$ it holds that $d_i(p, \sigma(p)) \leq 2R_i$ and thus $\mathsf{rad}(I, \sigma, d_i) \leq 2R_i$. If $f_i = \mathsf{diam}$ it holds for $p, q \in \mathcal{P}$ with $\sigma(p) = \sigma(q)$ that $d_i(p, q) \leq 2R_i$ and thus $\mathsf{diam}(I, \sigma, d_i) \leq 2R_i$. $\square$

Now let $R'_i$ be defined as the biggest distance with respect to $d_i$ smaller or equal than $\beta_i R_i$ between a pair of points in $\mathcal{P}$. Observe that actually $G_{\beta_1 R_1, \beta_2 R_2} = G_{R'_1, R'_2}$. Thus we obtain the following corollary of Lemma 29.

**Corollary 31.** *Let $R'_i = \max_{p,q \in \mathcal{P}: d_i(p,q) \leq \beta_i R_i} d_i(p, q)$ for $i = 1, 2$. If $G_{R'_1, R'_2}$ contains an independent set of size greater $k$ then there exists no solution with cost at most $(R_1, R_2)$.*

In conclusion it is sufficient to compute a maximal independent set in $G_{R_1,R_2}$ for all $(R_1, R_2) \in \mathcal{R} = \mathcal{R}_1 \times \mathcal{R}_2$ (remember that values in $\mathcal{R}$ directly correspond to potential cost values). If the independent set has size less or equal to $k$ we include the respective solution in the approximate Pareto set. Because of Lemma 30 and Corollary 31 one obtains a $(2, 2)$-approximate Pareto set. Given that $\mathcal{R}_i$ contains $O(n^2)$ distances between points in $\mathcal{P}$ and the calculation of a maximal independent set in $O(n^2)$ we can compute the approximate Pareto set in time $O(n^6)$. One can easily improve upon the running time by observing the following. If we find an independent set of size greater $k$ in the graph $G_{R_1,R_2}$ we do not have to consider any combination $(R_1', R_2)$ with $R_1' < R_1$. Similarly if the maximal independent set has size at most $k$ one does not need to consider any combination $(R_1, R_2')$ with $R_2' > R_2$ since it would be dominated by the currently found solution anyway. We combine both observations to reduce the number distances in $\mathcal{R}$ which we have to consider.

We start with the largest distance $R_1 \in \mathcal{R}_1$ and the smallest distance $R_2 \in \mathcal{R}_2$ and increase the $R_2$ if the maximal independent set computed in $G_{R_1,R_2}$ has size $> k$, we decrease $R_1$ otherwise. This way one would still obtain a $(2, 2)$-approximate Pareto set while reducing the number of considered pairs in $\mathcal{R}$ from $O(n^4)$ to $O(n^2)$ which yields a total running time of $O(n^4)$.

This can be further improved by observing the following. If we already have calculated the graph $G_{R_1,R_2}$ and a maximal independent set in this graph we probably do not need to recalculate the entire graph and independent set if we slightly decrease $R_1$ (and thus remove edges from the graph) or slightly increase $R_2$ (which adds edges). Instead one can simply modify the current graph while keeping track how this affects the independent set. This is far cheaper most of the time. To update the independent set properly we store for each node $p \in \mathcal{P}$ the set $B_p$ of its neighbors contained in the current independent set (intuitively these are the nodes that keep $p$ from joining the independent set as well). Following this idea one ends up with Algorithm 3. In the following we will prove its correctness and that its running time is $O(n^3)$.

---

**Algorithm 3:**

---

1   sort distances $d_1(p, q)$ for $p, q \in \mathcal{P}$: $d_1^{(1)} \leq \ldots \leq d_1^{(m)}$
2   sort distances $d_2(p, q)$ for $p, q \in \mathcal{P}$: $d_2^{(1)} \leq \ldots \leq d_2^{(m)}$
3   let $j = 0, \mathbb{P}_{(2,2)} = \emptyset, I = \mathcal{P}, E = \emptyset, R_1 = d_1^{(m)}, R_2 = 0$
4   let $B_p = \emptyset$ for all $p \in \mathcal{P}$
5   **for** $i = m, \ldots, 1$ **do**
6      **while** $|I| > k$ and $j \leq m$ **do**
7          $j{+}{+}$
8          $R_2 = d_2^{(j)}$
9          **foreach** *point pair* $\{p, q\}$ *with* $d_2(p, q) = R_2$ *and* $d_1(p, q) \leq R_1$ **do**
10             ADDEDGE($\{p, q\}$)
11      **if** $|I| \leq k$ **then**
12          let $C = I$, let $\sigma(p) = p$ for $p \in I$ and $\sigma(p) = q$ where $q$ is an arbitrary element in $B_p$ for $p \in \mathcal{P} \backslash I$. Add $(C, \sigma)$ to $\mathbb{P}_{(2,2)}$
13      **foreach** *point pair* $p, q$ *with* $d_1(p, q) = R_1$ *and* $d_2(p, q) \leq R_2$ **do**
14          REMOVEEDGE($\{p, q\}$)
15      $R_1 = d_1^{(i-1)}$
16   **return** $\mathbb{P}_{(2,2)}$

---

**Lemma 32.** *The following invariants are true for Algorithm 3 (at the beginning of line 6):*

1. $(\mathcal{P}, E) = G_{R_1, R_2}$

2. $I$ *is a maximal independent set of* $G_{R_1, R_2}$

3. *For all* $p \in \mathcal{P}$, *it holds that* $B_p$ *is the set of neighbors of* $p$ *contained in* $I$.

The invariant can be proven via a straightforward induction. The main idea is, that if we remove an edge between a point $p$ in the independent set $I$ and another point $q$ we need to check whether $q$ has

---

**Algorithm 4:** ADDEDGE($\{p, q\}$)

---

**1** W.l.o.g. $q \in I \Rightarrow p \in I$
**2** **if** $p \in I$ **then**
**3**      $B_q = B_q \cup \{p\}$
**4** **if** $p \in I, q \in I$ **then**
**5**      $I = I \setminus \{q\}$
**6**      **for** $\{q, r\} \in E$ **do**
**7**          $B_r = B_r \setminus \{q\}$
**8**          **if** $|B_r| = 0$ **then**
**9**              $I = I \cup \{r\}$
**10**              Add $r$ to $B_s$ for all neighbors $s$ of $r$

**11** $E = E \cup \{\{p, q\}\}$

---

---

**Algorithm 5:** REMOVEEDGE($\{p, q\}$)

---

**1** $E = E \setminus \{\{p, q\}\}$
**2** **if** $p \in I$ or $q \in I$ **then**
**3**      set $v = p, w = q$ if $q \in I$, set $v = q, w = p$ otherwise
**4**      $B_v = B_v \setminus \{w\}$
**5**      **if** $|B_v| = 0$ **then**
**6**          $I = I \cup \{v\}$
**7**          **foreach** neighbor $r$ of $v$ **do**
**8**              $B_r = B_r \cup \{v\}$

---

still neighbors in the independent set left (i.e. $B_q \neq \emptyset$). If this is not the case we need to add $q$ to the independent set to keep it maximal.

If we add an edge, the only critical case appears when it connects two nodes $p, q$ which are both in the independent set. If this is the case, we need to remove one of them from $I$ to keep it an independent set. However if we remove $q$ we need to check for every neighbor $r$ if it can now be added to $I$ (i.e., the respective set $B_r$ is empty) to ensure that $I$ stays maximal. In both operations the entries $B_x$ are updated for all $x \in \mathcal{P}$ accordingly.

**Theorem 33.** *Algorithm 3 calculates a $(2, 2)$-approximate Pareto set.*

*Proof.* Let us consider an arbitrary solution $S$ with cost $R_i^*$ with respect to $f_i$ for $i = 1, 2$. We choose $i', j'$ such that $d_1^{(i')} = \max_{p, q \in \mathcal{P}: d_1(p,q) \leq \beta_1 R_1^*} d_1(p, q)$ and $d_2^{(j')} = \max_{p, q \in \mathcal{P}: d_2(p,q) \leq \beta_2 R_2^*} d_2(p, q)$. Using Corollary 31 we may conclude that if Algorithm 3 reaches an iteration with $i = i'$ and $j = j'$ the maximal independent set has size at most $k$. By Lemma 30 this results in a solution with cost $(2R_1^*, 2R_2^*)$ which is added to $\mathbb{P}_{(2,2)}$. However there might be two cases which prevent this situation from happening:

- Case 1: At some point during the execution of the algorithm $i = i'$, $j \leq j'$ and $i$ gets decreased.

  One might note that $i$ only gets decreased if an independent set of size at most $k$ has been found in $G_{d_1^{(i')}, d_2^{(j)}}$. Since $d_1^{(i')} \leq \beta_1 R_1^*$ and $d_2^{(j)} \leq d_2^{(j')} \leq \beta_2 R_2^*$ we obtain by Lemma 30 that a solution with cost $(2R_1^*, 2R_2^*)$ has been found. Thus $S$ is $(2, 2)$-approximated.

- Case 2: At some point during the execution of the algorithm $i \geq i'$, $j = j'$ and $j$ gets increased.

  We observe that $j$ gets only increased if the current independent set $I$ of the algorithm has size greater $k$. Furthermore the edge set of $G_{d_1^{(i')}, d_2^{(j')}}$ is a subset of the one of $G_{d_1^{(i)}, d_2^{(j)}}$.

Thus $I$ is also an independent set of $G_{d_1^{(i')}, d_2^{(j')}}$. However by Corollary 31 this means that there cannot exist a solution with cost $(R_1^*, R_2^*)$ which is a contradiction.

Thus we may conclude that any solution is at least $(2,2)$-approximated by one of the solution contained in $\mathbb{P}_{(2,2)}$. $\qquad\square$

Now we want to bound the running time of the algorithm. To do this, we will introduce a potential function $\Phi$ whose value is determined by the current state of the edge set $E$ and the independent set $I$ of the algorithm. We define $\Phi((P, E), I) = \sum_{p \in I} \deg(p)$ where $\deg(p)$ is the degree of $p$ in $(P, E)$. Another way to think about this potential is that it is the sum of the size of $B_p$ for every node $p$, because each node in the independent set blocks every neighbor $q$ from becoming part of the independent set itself and thus appears in the respective set $B_q$. We will now bound both the cost of the operation REMOVEEDGE$(\{p, q\})$, as well as those of ADDEDGE$(\{p, q\})$ by a function in $O(n)$ plus the respective potential change $\Delta\Phi$ induced by this change (which might be negative).

Before doing this we need to clarify how the sets $(B_p)_{p \in \mathcal{P}}$ are actually represented by the algorithm. We want to be able to add a point to $B_p$, delete a point and extract an arbitrary element contained in $B_p$ in time $O(1)$ each. All these requirements are fulfilled by a doubly linked list if we maintain pointers for each edge $e = \{p, q\}$ telling us where $p$ is stored in the set $B_q$ and vice versa. Using this we can now bound the cost of the respective operations.

**Lemma 34.** *The cost of* REMOVEEDGE$(\{p, q\})$ *is bounded by* $O(2 + \Delta\Phi)$.

*Proof.* If neither $p$ nor $q$ are contained in $I$, the potential does not change while we only incur constant cost and the lemma holds. Otherwise let w.l.o.g. $p$ be contained in $I$ (note that not both nodes can be contained in $I$ since they were connected by an edge). We distinguish two cases:

If $q$ has another neighbor in $I$ besides $p$, i.e. $|B_q| \geq 2$, then the size of $B_q$ decreases by $1$ and the potential $\Phi$ also decreases by $1$. Thus $O(2 + \Delta\Phi) = O(1)$ while the operation takes only constant time.

If $p$ was the only neighbor of $q$ that was contained in the independent set, $q$ gets added to $I$ as well to ensure that $I$ is maximal. As a result for every neighbor $r$ of $q$ we need to add $q$ to $B_r$ which leads to a running time of $O(c)$, where $c$ is the number of neighbors of $q$, after the edge $\{p, q\}$ has been removed. At the same time every neighbor of $q$ also increases the potential $\Phi$ by one when $q$ gets added to $I$. Thus $\Delta\Phi = c - 1$. As a result the running time of the operation lies still within $O(2 + \Delta\Phi) = O(c + 1)$. $\qquad\square$

**Lemma 35.** *The cost of* ADDEDGE$(\{p, q\})$ *is bounded by* $O(2n + \Delta\Phi)$

*Proof.* The only critical case is that both $p$ and $q$ are contained in $I$ because otherwise we only incur constant cost while also the potential only changes by a constant. In this case the node $q$ gets removed from the independent set which means that for every neighbor $r$ we need to remove $q$ from $B_r$. Since $q$ has at most $n$ neighbors the cost of this procedure lies in $O(n)$ while the potential decreases by at most $n$. However if $q$ was the only neighbor of $r$ within $I$ we need to add $r$ to $I$ to ensure that $I$ is maximal. As a result we again need to update the set $B_s$ for all neighbors $s$ of $r$, which costs $O(\deg(r))$ while the potential gets increased by $\deg(r)$. In total we get that the cost of ADDEDGE is bounded by $O\left(n + \sum_{r \in I'} \deg(r)\right)$, where $I'$ denotes the set of nodes newly added to the independent set. At the same time:

$$O(2n + \Delta\Phi) = O\left(2n - n + \sum_{r \in I'} \deg(r)\right) = O\left(n + \sum_{r \in I'} \deg(r)\right)$$

Thus the lemma holds. $\qquad\square$

Using these two lemmas we can now bound the running time of Algorithm 3:

**Theorem 36.** *The running time of Algorithm 3 lies within* $O(n^3)$.

*Proof.* The running time of the algorithm mainly consists of three parts:

- The sorting of the distances in the beginning which needs time $O(n^2 \log(n))$

- Up to $\frac{n(n-1)}{2}$ calls of DELETEEDGE and ADDEDGE each (as each point pair gets inserted and deleted at most once). Using the previous two lemmas we can bound the respective costs by $O(2 + \Delta\Phi)$ and $O(2n + \Delta\Phi)$. One might note that both at the beginning as well at the end of the algorithm there is not a single edge in the edge set which means that $\Phi = 0$. Thus the total change of the potential is 0 and we can bound the total cost of the insertion and deletion operations by $O(n^3)$.

- Constructing a solution from the independent set costs at most $O(n)$ since we only have to extract an element of $B_p$ for every point $p \in \mathcal{P}$. Since at most $O(n^2)$ solutions get added to $\mathbb{P}_{(2,2)}$ (one for every $i$ in line 5 of Algorithm 3) the total time for constructing the solutions lies in $O(n^3)$.

Thus also the entire running time is bounded by $O(n^3)$. $\qquad\square$

The following corollary is an immediate consequence of Theorem 33 and Theorem 36.

**Corollary 37.** *Given a set $\mathcal{P}$ of $n$ points, two metrics $d_1, d_2$ on $\mathcal{P}$, $k \in \mathbb{N}$ and two objectives $f_1, f_2 \in \{\mathsf{rad}, \mathsf{diam}\}$. We can compute the approximate Pareto set $\mathbb{P}_{(2,2)}$ with respect to $f_1, f_2$ in time $O(n^3)$.*

## B.2 The size of the Pareto set

In this section we will prove that there can actually exist $\Omega(n^2)$ different Pareto optimal solutions if we combine two different $k$-center/$k$-diameter objectives with each other.

**Lemma 38.** *For every $n \in \mathbb{N}$ and $f_1, f_2 \in \{\mathsf{rad}, \mathsf{diam}\}$ there exist a set of $n$ points $\mathcal{P}$ and metrics $d_1, d_2$ on $\mathcal{P}$ such that the size of the Pareto set $\mathbb{P}$ with respect to $f_1, f_2$ is in $\Omega(n^2)$.*

*Proof.* Let $\mathcal{P}$ be a set of size $n$. We chose the metrics $d_1, d_2$ on $\mathcal{P}$ as follows. Let $p_1, ..., p_m$ be all pairs of distinct points in $\mathcal{P}$ (i.e. $m = \frac{n(n-1)}{2}$). For every $i$ we set $d_1(p_i)$ to $1 + \frac{i}{m}$ and $d_2(p_i)$ to $2 - \frac{i}{m}$. Furthermore we set $d_1(p,p) = d_2(p,p) = 0$ for all $p \in \mathcal{P}$. One may note that the distance between two distinct points lies always between 1 and 2. Thus the triangle inequality is trivially fulfilled and $d_1, d_2$ are indeed metrics on $\mathcal{P}$.

Now we consider the situation were we want to compute a $k$-clustering with $k = n - 1$. Observe that only a single point gets assigned to a center besides itself. Then every $k$-clustering corresponds to one pair $p_i$ of points that are contained in the same cluster and vice versa. The cost of this solution is $(d_1(p_i), d_2(p_i))$. It is easy to observe that each of theses solutions is in fact Pareto optimal. Let $p_i$ and $p_j$ be two pairs and assume w.l.o.g that $i < j$. Then it holds that $d_1(p_i) < d_1(p_j)$ while $d_2(p_i) > d_2(p_j)$ which means that neither solution dominates the other one. Thus we end up with $m \in \Omega(n^2)$ different Pareto optimal solutions. $\qquad\square$

However one might note, that in the clustering instance constructed to prove Lemma 38 many solutions differ only by a small factor. One could obtain a $(2, 2)$-approximate Pareto set of size 1 by just choosing an arbitrary solution. While this is of course a special case one can prove that in fact there always exists a $(2, 2)$-approximate Pareto set of size $n$. This result is a direct consequence of the fact that in a metric there can only be a limited number of distances between point pairs (which correspond to potential $k$-center/$k$-diameter solutions) that differ pairwisely by a factor of 2.

**Lemma 39.** *Let $\mathcal{P}$ be a set of $n$ points and let $d$ be a metric on $\mathcal{P}$. Then there do not exist $n$ pairs of points $p_1, ..., p_n$ in $\mathcal{P}$ such that for all $i \neq j$ it holds that $d(p_i) \leq \frac{1}{2}d(p_j)$ or $d(p_i) \geq 2d(p_j)$.*

*Proof.* Suppose the statement would be wrong. Then there exists a subset of indices $i_1, .., i_k$ such that $p_{i_1} = \{v_{i_1}, w_{i_1}\}, ..., p_{i_k} = \{v_{i_k}, w_{i_k}\}$ and $w_{i_j} = v_{i_{j+1}}$ for all $j = 1, \ldots, k-1$ and $w_{i_k} = v_{i_1}$. Let w.l.o.g. the distance of $p_{i_1}$ be maximum among distances of all $p_{i_j}$. Then it holds by the triangle inequality that

$$d(p_{i_1}) \leq \sum_{j=2}^{k} d(p_{i_j}).$$

We might observe that the second longest edge is at most half as long as $p_{i_1}$ and that the third longest edge is at most half as long as the second longest and so on. Thus we obtain

$$d(p_{i_1}) \leq d(p_{i_1}) \sum_{j=1}^{k} \frac{1}{2^j} < d(p_{i_1}).$$

which is a contradiction. $\qquad\square$

The following corollary is a consequence of Lemma 39.

**Corollary 40.** *Given a set $\mathcal{P}$ of $n$ points, two metrics $d_1, d_2$ on $\mathcal{P}$, $k \in \mathbb{N}$ and two objectives $f_1, f_2 \in \{\mathsf{rad}, \mathsf{diam}\}$. There exist a $(2, 2)$-approximate Pareto set with respect to $f_1, f_2$ of size $O(n)$.*

*Proof.* Let $\mathbb{P}$ be the Pareto set induced by $f_1, f_2$. We will now construct a (2,2)-approximate Pareto set $Q$ of $\mathbb{P}$. We will do this by iterating through $\mathbb{P}$. First let $Q = \emptyset$. We start by adding the solution that optimizes $f_1$ to $Q$. Then we move along the curve by improving $f_2$ while getting worse in $f_1$. Whenever we encounter a solution that is not (2,2)-approximated by the previous points in $Q$, we add it. We repeat this until we reach the end of the curve.

Let $x$ and $y$ be two neighboring solutions in Q. Without loss of generality we assume that $x$ was added to $Q$ before $y$. Note that since we only get worse in $f_1$ we know that $f_1(x) < f_1(y)$ and therefore it must hold that $f_2(y) < \frac{1}{2} f_2(x)$ since otherwise $x$ would (2,2)-approximate $y$.

By construction $Q$ is a (2,2)-approximate Pareto set to $\mathbb{P}$. Now assume for contradiction that $|\mathbb{P}| > n$. Then there would have been added more than $n$ solutions to $Q$, such that for all $x \neq y \in Q$ it holds that either $f_2(x) > 2f_2(y)$ or $f_2(y) > 2f_2(x)$. Since all solutions correspond directly to pairwise distances in $d_2$ of our point set $\mathcal{P}$, this is a contradiction to Lemma 39 and therefore the corollary follows. $\qquad\square$

### B.3 Combination of $k$-center/$k$-diameter with $k$-median/$k$-means

We consider the combination of $k$-center with $k$-median or $k$-means with respect to two distinct metrics $d_1$ and $d_2$. It is easy to see, that the size of the Pareto set is bounded by the number of possible radii for $k$-center. Since this is always the distance between two points, we obtain $|\mathbb{P}| \leq \frac{n(n-1)}{2}$. Alamdari and Shmoys [5] show that we can compute the approximate Pareto set $\mathbb{P}_\alpha$ for $\alpha = (4, 8)$ with respect to $f_1 = \mathsf{rad}$ and $f_2 = \mathsf{med}$ and the special case where $d_1 = d_2$. They modify the LP-based approach by Charikar et al. [12] for $k$-median to compute a solution which is also a good approximation to the radius.

In a similar way we show that the approach for $k$-median by Jain and Vazirani [41] can be modified to compute the approximate Pareto set $\mathbb{P}_\alpha$ for $\alpha = (9, 6 + \epsilon)$ in the more general case $d_1 \neq d_2$.

**Theorem 41.** *Given a finite set $\mathcal{P}$, two metrics $d_1, d_2$ on $\mathcal{P}$, $k \in \mathbb{N}$. If for $R > 0$ there exists a solution $(C, \sigma)$ for $\mathcal{P}$ with $\mathsf{rad}(C, \sigma, d_1) \leq R$ we can compute a solution $(C', \sigma')$ in polynomial time such that $\mathsf{rad}(C', \sigma', d_1) \leq 9 \cdot R$ and $\mathsf{med}(C', \sigma', d_2) \leq (6 + \epsilon) \cdot \mathsf{med}(C, \sigma, d_2)$.*

Since the algorithm in [41] also produces a $(54 + \epsilon)$-approximation for $k$-means we can also adapt it to compute the approximate Pareto set $\mathbb{P}_\alpha$ for $\alpha = (9, 54 + \epsilon)$ for the case $f_1 = \mathsf{rad}$ and $f_2 = \mathsf{mean}$.

**Theorem 42.** *Given a finite set $\mathcal{P}$, two metrics $d_1, d_2$ on $\mathcal{P}$, $k \in \mathbb{N}$. If for $R > 0$ there exists a solution $(C, \sigma)$ for $\mathcal{P}$ with $\mathsf{rad}(C, \sigma, d_1) \leq R$ we can compute a solution $(C', \sigma')$ in polynomial time such that $\mathsf{rad}(C', \sigma', d_1) \leq 9 \cdot R$ and $\mathsf{mean}(C', \sigma', d_2) \leq (54 + \epsilon) \cdot \mathsf{mean}(C, \sigma, d_2)$.*

We assume that the algorithm by Alamdari and Shmoys [5] for $k$-center and $k$-median can also be adapted to the combination of $k$-center and $k$-means with an adjusted approximation factor.

For the case $f_1 = \mathsf{rad}$ and $f_2 = \mathsf{med}$ we can compute the approximate Pareto set $\mathbb{P}_\alpha$ for $\alpha = (9, 6 + \epsilon)$ by checking all $\frac{n(n-1)}{2}$ possible values for the radius $R$. Suppose there exists a $k$-center solution for $\mathcal{P}$ with radius $\leq R$ and let $(C_R, \sigma_R)$ be such a solution that minimizes the $k$-median cost. Theorem 41 implies that we can compute a solution $(C'_R, \sigma'_R)$ with $f_1(C'_R, \sigma'_R, d_1) \leq 9 \cdot R$ and $\mathsf{med}(C'_R, \sigma'_R, d_2) \leq (6 + \epsilon) \cdot \mathsf{med}(C_R, \sigma_R, d_2)$ in polynomial time. Let $\mathbb{P}_\alpha$ be the set containing all these solutions, then $\mathbb{P}_\alpha$ is an $\alpha$-approximate Pareto set. Following the same approach we can

compute the approximate Pareto set $\mathbb{P}_\alpha$ for $\alpha = (9, 54 + \epsilon)$ in the case $f_1 = $ rad and $f_2 = $ mean. This yields the following corollary.

**Corollary 43.** *Given a finite set $\mathcal{P}$, two metrics $d_1, d_2$ on $\mathcal{P}$ and $k \in \mathbb{N}$. We can compute the approximate Pareto set $\mathbb{P}_\alpha$ for*

- $\alpha = (9, 6 + \epsilon)$ *with respect to the objectives $f_1 = $ rad and $f_2 = $ med*

- $\alpha = (9, 54 + \epsilon)$ *with respect to the objectives $f_1 = $ rad and $f_2 = $ mean*

*in polynomial time.*

To combine $k$-diameter with $k$-median/$k$-means we observe that any solution $(C, \sigma)$ with $\mathsf{diam}(C, \sigma, d_1) \leq R$ satisfies $\mathsf{rad}(C, \sigma, d_1) \leq R$. By Theorem 41 we can compute a solution $(C', \sigma')$ with $\mathsf{diam}(C', \sigma', d_1) \leq 2\mathsf{rad}(C'\sigma', d_1) \leq 18R$ and $\mathsf{med}(C'\sigma', d_2) \leq (6 + \epsilon)\mathsf{med}(C, \sigma, d_2)$ in polynomial time. A similar statement holds for the combination of $k$-diameter and $k$-means by Theorem 42.

**Corollary 44.** *Given a finite set $\mathcal{P}$, two metrics $d_1, d_2$ on $\mathcal{P}$ and $k \in \mathbb{N}$. We can compute the approximate Pareto set $\mathbb{P}_\alpha$ for*

- $\alpha = (18, 6 + \epsilon)$ *with respect to the objectives $f_1 = $ diam and $f_2 = $ med*

- $\alpha = (18, 54 + \epsilon)$ *with respect to the objectives $f_1 = $ diam and $f_2 = $ mean*

*in polynomial time.*

To prove Theorem 41 we repeat the $(6 + \epsilon)$-approximation algorithm for $k$-median proposed by Jain and Vazirani [41] and explain the relevant changes to obtain a solution which also approximates the radius. Following [41] we first develop an algorithm for the facility location problem.

In the facility location problem we deal with a set $F$ of possible facilities and a set $D$ of clients. Furthermore we are given a metric $d$ on $F \cup D$ and facility opening costs $f(i) \geq 0$ for every $i \in F$. The task is to compute a set $C \subset F$ which minimizes the sum of opening costs and distances between a client and its closest facility $\sum_{i \in C} f(i) + \sum_{j \in D} \min_{i \in C} d(i, j)$. Later, when we use the algorithm for facility location to solve the $k$-median problem on the set $\mathcal{P}$, we set $F = D = \mathcal{P}$.

We consider the combination of facility location with the $k$-center objective. Suppose we are given two metrics $d_1, d_2$ on $F \cup D$ and want to minimize the facility location cost (with respect to $d_2$) as well as the maximum distance between a client and its facility (with respect to $d_1$). Now a client may not be assigned to its nearest facility with respect to $d_2$, since this may result in a large radius with respect to $d_1$. Thus we say that a solution consists of a set $C \subset F$ and an assignment $\sigma \colon D \to C$ of clients to facilities. We seek to minimize the objectives $\mathsf{rad}(C, \sigma, d_1) = \max_{j \in D} d_1(j, \sigma(j))$ and $\mathsf{fac}(C, \sigma, d_2) = \sum_{i \in C} f(i) + \sum_{j \in D} d_2(j, \sigma(j))$.

**Theorem 45.** *Given a set of facilities $F$ with opening costs $f(i) \geq 0$ for $i \in F$, clients $D$, two metrics $d_1, d_2$ on $F \cup D$ and a number $R > 0$. If there exists a solution $(S, \tau)$ with $\mathsf{rad}(S, \tau, d_1) \leq R$, then we can compute a solution $(C, \sigma)$ in polynomial time such that $\mathsf{rad}(C, \sigma, d_1) \leq 3 \cdot R$ and $3 \sum_{i \in C} f(i) + \sum_{j \in D} d_2(j, \sigma(j)) \leq 3 \cdot \mathsf{fac}(S, \tau, d_2)$.*

For some radius $R > 0$ we want to compute an optimal facility location solution with respect to $d_2$ under the side constraint that the radius of every emerging cluster is at most $R$ with respect to $d_1$. We consider the standard formulation of the facility location problem as an integer program with the additional constraint that a client can only be assigned to a facility if it is at a distance at most $R$. Let $M = \{(i, j) \mid i \in F, j \in D, d_1(i, j) \leq R\}$ be the set of all such combinations. We consider the

following integer program.

$$\text{minimize} \sum_{i \in F} f(i)y_i + \sum_{(i,j) \in M} d_2(i,j)x_{ij}$$

subject to

$$\sum_{(i,j) \in M} x_{ij} \geq 1 \qquad \forall j \in D$$

$$x_{ij} \leq y_i \qquad \forall (i,j) \in M$$

$$x_{ij} \in \{0,1\} \qquad \forall (i,j) \in M$$

$$y_i \in \{0,1\} \qquad \forall i \in F$$

Given an optimal solution $(x,y)$ of this program we open a facility $i \in F$ iff $y_i = 1$ and assign a client $j$ to $i$ iff $x_{ij} = 1$. Let $(C, \sigma)$ be the respective solution for facility location. It is easy to see that we have $\text{rad}(C, \sigma, d_1) \leq R$ since we only allow a client to be assigned to a facility within distance of $R$ with respect to $d_1$.

To obtain the linear program we relax the last constraints to $x_{ij} \geq 0$ and $y_i \geq 0$. Its dual is then given by

$$\text{maximize} \sum_{j \in D} \alpha_j$$

subject to

$$\alpha_j - \beta_{ij} \leq d_2(i,j) \qquad \forall (i,j) \in M$$

$$\sum_{(i,j) \in M} \beta_{ij} \leq f_i \qquad \forall i \in F$$

$$\alpha_j \geq 0 \qquad \forall j \in D$$

$$\beta_{ij} \geq 0 \qquad \forall (i,j) \in M$$

We construct a solution for the dual LP by starting with $\alpha_j = 0$ for all $j \in D$ and growing them simultaneously at the same rate until some stopping condition is met. Later on the variable $\alpha_j$ will pay for the connection cost of client $j$ and it will pay partially for the opening cost of some facility.

As in the algorithm in [41] we introduce the notion of time. We start at time $0$, where all variables are set to $0$, all facilities are closed and all clients are unconnected. While time proceeds we increase all variables of unconnected clients uniformly such that if a client $j$ is unconnected at time $t$ we have $\alpha_j = t$ at this time.

We have to maintain the feasibility of the solution. There are two types of events at time $t$ which can violate the feasibility.

The first type of event happens when there is a unconnected client $j$ and $(i,j) \in M$ such that $\alpha_j = d_2(i,j)$. In this case we declare $(i,j)$ to be tight. If facility $i$ is open we declare $j$ to be connected to $i$. Otherwise from time $t$ on we have to maintain the constraint $\alpha_j - \beta_{ij} \leq d_2(i,j)$ by starting to increase $\beta_{ij}$ at the same rate as $\alpha_j$.

The second type of event happens when there is a facility $i$ such that $\sum_{(i,j) \in M} \beta_{ij} = f(i)$. In this case we declare facility $i$ to be open. Furthermore all unconnected clients $j$ with $(i,j)$ tight are declared to be connected to $i$.

We stop the process when all clients are connected. Let $C'$ be the set of open facilities at the end of this process. For a client $j$ let $\sigma'(j) = i$ if $j$ is declared connected to facility $i$. Then $(C', \sigma')$ is a feasible solution to the facility location problem with $\text{rad}(C', \sigma', d_1) \leq R$. We would like to bound the facility location cost $\text{fac}(C', \sigma', d_2)$ in terms of $\sum_{j \in C} \alpha_j$. However it is not possible to upper bound the facility opening costs for facilities in $C'$, since for a client $j$ there may be multiple facilities

$i \in C'$ for which $\beta_{ij} > 0$. Thus $\alpha_j$ pays for opening multiple facilities from $C'$. Therefore following the work of [41] we will close some of the facilities in $C'$ such that in the end every client pays for opening at most one facility.

To decide which facilities we close from $C'$ we construct a graph $G$ on $C'$ as follows. We connect two facilities $i, i' \in C'$ by an edge if there exists a client $j$ with $\beta_{ij} > 0$ and $\beta_{i'j} > 0$. Then let $C'' \subset C'$ be a maximal independent set in $G$. For a client $j$ we assign a facility from $C''$ as follows. If there is a facility $i \in C''$ with $\beta_{ij} > 0$ we set $\sigma''(j) = i$ and call $j$ directly connected. Since $C''$ is an independent set, this facility is even unique. If there is no such facility in $C''$, we consider the facility $i$ which $j$ is assigned to in $(C', \sigma')$, i.e., $i = \sigma'(j)$. If $i \in C''$ we set $\sigma''(j) = i$ and call $j$ directly connected. Otherwise we know by maximality of $C''$ that there exists a facility $i' \in C''$ which is connected to $i$ by an edge. In this case we set $\sigma''(j) = i'$ and call $j$ indirectly connected.

**Lemma 46.** *For the solution $(C'', \sigma'')$ computed by the algorithm we obtain $3 \sum_{i \in C''} f(i) + \sum_{j \in D} d_2(j, \sigma''(j)) \le 3 \sum_{j \in D} \alpha_j$ and $\mathsf{rad}(C'', \sigma'', d_1) \le 3R$.*

*Proof.* Let $D_1 \subset D$ be the set of directly connected clients and $D_2 \subset D$ be the set of indirectly connected clients.

For a facility $i \in C''$ we know that $f(i) = \sum_{(i,j) \in M} \beta_{ij} = \sum_{i \in D_1 : i = \sigma''(j)} \beta_{ij}$. Where the first equality holds, because $i$ was declared open by the algorithm and the second equality holds, because all clients $j \in D$ with $\beta_{ij} > 0$ have to be assigned to $i$ by $\sigma''$ and are directly connected.

Let $j \in D$ and $i = \sigma''(j)$. If $j$ is directly connected we know that $(i, j) \in M$ and $\alpha_j = \beta_{ij} + d_2(i, j)$. If $j$ is indirectly connected we claim that $d_2(i, j) \le 3\alpha_j$. Let $i' = \sigma'(j)$, since $j$ is indirectly connected, we know that $i$ and $i'$ are connected by an edge in $G$. Thus there must be a client $j' \in D$ with $\beta_{ij'} > 0$ and $\beta_{i'j'} > 0$. We know that $\alpha_{j'} = \beta_{ij'} + d_2(i, j')$ and $\alpha_{j'} = \beta_{i'j'} + d_2(i', j')$, so therefore $d_2(i, j') \le \alpha_{j'}$ and $d_2(i', j') \le \alpha_{j'}$. Let $t$ be the time when facility $i$ is declared open by the algorithm and $t'$ the time when $i'$ is declared open by the algorithm. Since $j'$ is declared to be connected as soon as it has a tight edge to an open facility, we know that $j'$ is connected at time $\min\{t, t'\}$ and therefore $\alpha'_j \le \min\{t, t'\}$. On the other hand we know that $t' = \alpha_j$, since $\sigma'(j) = i'$. In total we obtain by triangle inequality, that $d_2(i, j) \le d_2(i, j') + d_2(j', i') + d_2(i', j) \le 2\alpha_{j'} + \alpha_j \le 3\alpha_j$.

Altogether this implies

$$3 \sum_{i \in C''} f(i) + \sum_{j \in D} d_2(j, \sigma''(j)) = 3 \sum_{i \in C''} \sum_{j \in D_1 : i = \sigma''(j)} \beta_{ij} + \sum_{j \in D_1} \alpha_j - \beta_{\sigma''(j)j} + \sum_{j \in D_2} 3\alpha_j$$

$$\le 3 \sum_{j \in D_1} \beta_{\sigma''(j)j} + \sum_{j \in D_1} \alpha_j - \beta_{\sigma''(j)j} + \sum_{j \in D_2} 3\alpha_j$$

$$\le 3 \sum_{j \in D} \alpha_j.$$

This yields the first part of the lemma.

For the second part let $j$ be a client and $i = \sigma''(j)$. We have to prove that $d_1(i, j) \le 3R$. If $j$ is directly connected we have $(i, j) \in M$ which implies $d_1(i, j) \le R$. If $j$ is indirectly connected let $i' = \sigma'(j)$. Notice that this implies $(i', j) \in M$. Since $j$ is indirectly connected, we know that $i$ and $i'$ are connected by an edge in $G$. Thus there must be a client $j' \in D$ with $\beta_{ij'} > 0$ and $\beta_{i'j'} > 0$ and therefore $(i, j'), (i', j') \in M$. We obtain $d_1(i, j) \le d_1(i, j') + d_1(j', i') + d_1(i', j) \le 3R$. $\qquad \square$

We are now able to prove the theorem.

**Theorem 45.** *Given a set of facilities $F$ with opening costs $f(i) \ge 0$ for $i \in F$, clients $D$, two metrics $d_1, d_2$ on $F \cup D$ and a number $R > 0$. If there exists a solution $(S, \tau)$ with $\mathsf{rad}(S, \tau, d_1) \le R$, then we can compute a solution $(C, \sigma)$ in polynomial time such that $\mathsf{rad}(C, \sigma, d_1) \le 3 \cdot R$ and $3 \sum_{i \in C} f(i) + \sum_{j \in D} d_2(j, \sigma(j)) \le 3 \cdot \mathsf{fac}(S, \tau, d_2)$.*

*Proof.* Since $(S, \tau)$ is an integral solution to the LP with radius $R$, we know that the above algorithm computes a solution $\alpha$ for the dual LP with $\sum_{j \in D} \alpha_j \le \mathsf{fac}(S, \tau, d_1)$. Let $(C'', \sigma'')$ be the solution for facility location computed by the algorithm. We can apply Lemma 46 to see that $\mathsf{rad}(C'', \sigma'', d_1) \le 3 \cdot R$ and $3 \sum_{i \in C''} f(i) + \sum_{j \in D} d_2(j, \sigma''(j)) \le 3\mathsf{fac}(S, \tau, d_2)$. $\qquad \square$

Given some radius $R > 0$ and a set of points $\mathcal{P}$. We originally wanted to compute an optimal $k$-median solution of $\mathcal{P}$ with respect to $d_2$ under the side constraint that the radius of every emerging cluster is at most $R$ with respect to $d_1$. We consider the following formulation of the $k$-median problem as a linear program with the additional constraint that a point can only be assigned to a center if it is at distance at most $R$. Let $M = \{(i,j) \mid i,j \in \mathcal{P}, d_1(i,j) \leq R\}$ be again the set of all such combinations. We consider the following linear program.

$$\text{minimize} \sum_{(i,j)\in M} d_2(i,j)x_{ij}$$

subject to

$$\sum_{(i,j)\in M} x_{ij} \geq 1 \qquad\qquad \forall j \in \mathcal{P}$$

$$x_{ij} \leq y_i \qquad\qquad \forall (i,j) \in M$$

$$\sum_{i\in F} y_i \leq k$$

$$x_{ij} \geq 0 \qquad\qquad \forall (i,j) \in M$$

$$y_i \geq 0 \qquad\qquad \forall i \in \mathcal{P}$$

Its dual is then given by

$$\text{maximize} \sum_{j\in\mathcal{P}} \alpha_j - zk$$

subject to

$$\alpha_j - \beta_{ij} \leq d_2(i,j) \qquad\qquad \forall (i,j) \in M$$

$$\sum_{(i,j)\in M} \beta_{ij} \leq z \qquad\qquad \forall i \in \mathcal{P}$$

$$\alpha_j \geq 0 \qquad\qquad \forall j \in \mathcal{P}$$

$$\beta_{ij} \geq 0 \qquad\qquad \forall (i,j) \in M$$

$$z \geq 0$$

Let $(x^\star, y^\star)$ and $(\alpha^\star, \beta^\star, z^\star)$ be optimal solutions for the above linear program and its dual. We relax the constraint that restricts the number of centers by $k$. Instead as in [41] we introduce a cost $f$ for opening a center. This yields an instance for the facility location problem where the set of clients $D$ and facilities $F$ equals $\mathcal{P}$ and the cost for opening a facility equals $f$. We can use Theorem 45 to compute a facility location solution $(C, \sigma)$ for this instance. Let $(\alpha, \beta)$ be the respective solution for the dual program computed by the algorithm, then $\sum_{j\in\mathcal{P}} d_2(j,\sigma(j)) \leq 3(\sum_{j\in\mathcal{P}} \alpha_j - f|C|)$. Thus if $|C| = k$ we see that $(\alpha, \beta, f)$ is a feasible solution for the dual program with cost $\sum_{j\in\mathcal{P}} \alpha_j - f|C|$ and therefore $\mathsf{med}(C,\sigma,d_1) = \sum_{j\in\mathcal{P}} d_2(j,\sigma(j)) \leq 3(\sum_{j\in\mathcal{P}} \alpha_j - f|C|) \leq 3(\sum_{j\in\mathcal{P}} \alpha_j^\star - z^\star k) = 3\sum_{j\in\mathcal{P}} d_2(i,j)x_{ij}^\star$.

However we cannot rely on the existence of facility cost $f$ such that the algorithm for facility location outputs a solution with exactly $k$ centers. Instead for $\epsilon > 0$ we can find via binary search cost $f_1$ such that the respective solution $(C_1, \sigma_1)$ for facility location consists of $k_1 < k$ centers and the cost $f_2$ such that the respective solution $(C_2, \sigma_2)$ for facility location consists of $k_2 > k$ centers and $|f_2 - f_1| \leq \frac{\epsilon^2 d_{\min}}{3n}$ where $d_{\min} = \min_{i,j\in\mathcal{P}, i\neq j} d_2(i,j)$. We slightly modify the randomized rounding algorithm by [41] for $k$-median to select $k$ centers $C \subset C_1 \cup C_2$ such that in the end we obtain a solution with $\mathbb{E}(\mathsf{med}(C,\sigma,d_2)) \leq 6\sum_{j\in\mathcal{P}} d_2(i,j)x_{ij}^\star$ and $\mathsf{rad}(C,\sigma,d_1) \leq 9R$. We only explain the randomized rounding and do not go into detail about the binary search for $f_1$ and $f_2$ and the derandomization of selecting $C \subset C_1 \cup C_2$, since this can be done analogously to [41].

The randomized rounding works as follows: Let $a = \frac{k_2-k}{k_2-k_1}$. Furthermore for $i \in C_1$ consider the set $T_i = \{i' \in C_2 \mid \sigma_1^{-1}(i) \cap \sigma_2^{-1}(i') \neq \emptyset\} \subset C_2$. For every $i \in C_1$ let $c(i) = \mathrm{argmin}_{i'\in T_i} d_1(i',i)$ and

set $C_2' = \{c(i) \mid i \in C_1\} \cup T$ where $T$ is an arbitrary set of centers from $C_2 \setminus C_2'$ such that in the end $|C_2'| = k_1$. Let $S$ be a set of cardinality $k - k_1$ chosen uniformly at random from $C_2 \setminus C_2'$. With probability $a$ we set $C = C_1 \cup S$. For a point $j \in \mathcal{P}$ if $\sigma_2(j) \in S$ we set $\sigma(j) = \sigma_2(j)$ and otherwise $\sigma(j) = \sigma_1(j)$. With probability $1 - a$ we set $C = C_2' \cup S$. For a point $j \in \mathcal{P}$ if $\sigma_2(j) \in C$, we set $\sigma(j) = \sigma_2(j)$ and otherwise $\sigma(j) = c(\sigma_1(j))$.

The difference to the rounding procedure in [41] lies in the choice of $C_2'$. To ensure that the radius with respect to $d_1$ can be bounded, we are not allowed to simply set $C_2'$ as the set of points in $C_2$ closest to $C_1$. With a simple modification for $C_2'$ we can ensure that both the $k$-median cost and the radius can be bounded.

**Lemma 47.** *We have* $\mathbb{E}(\mathrm{med}(C, \sigma, d_2)) \leq (6 + 2\epsilon) \sum_{(i,j) \in M} x_{ij}^\star d_2(i, j)$ *and* $\mathrm{rad}(C, \sigma, d_1) \leq 9R$.

*Proof.* For a point $j \in \mathcal{P}$ we distinguish two cases. If $\sigma_2(j) \in C_2'$ then $\mathbb{E}(d_2(j, \sigma(j))) = a \cdot d_2(j, \sigma_1(j)) + (1 - a) \cdot d_2(j, \sigma_2(j))$. If $\sigma_2(j) \in C_2 \setminus C_2'$ then with probability $\frac{k - k_1}{k_2 - k_1} = 1 - a$ we have $\sigma_2(j) \in C$ and $d_2(j, \sigma(j)) = d_2(j, \sigma_2(j))$, with probability $a^2$ we have $\sigma_2(j) \notin C, \sigma_1(j) \in C$ and $d_2(j, \sigma(j)) = d_2(j, \sigma_1(j))$. With the remaining probability $a(1 - a)$ we have

$$d_2(j, \sigma(j)) = d_2(j, c(\sigma_1(j))) \leq d_2(j, \sigma_1(j)) + d_2(\sigma_1(j), c(\sigma_1(j)))$$
$$\leq d_2(j, \sigma_1(j)) + d_2(\sigma_1(j), \sigma_2(j))$$
$$\leq 2d_2(j, \sigma_1(j)) + d_2(j, \sigma_2(j)).$$

Where the second inequality holds because $\sigma_2(j) \in T_{\sigma_1(j)}$ and $c(\sigma_1(j)))$ is the closest center to $\sigma_1(j)$ among all centers in $T_{\sigma_1(j)}$. In total we obtain

$$\mathbb{E}(d_2(j, \sigma(j))) \leq (1 - a)d_2(j, \sigma_2(j)) + a^2 d_2(j, \sigma_1(j)) + a(1 - a)(2d_2(j, \sigma_1(j)) + d_2(j, \sigma_2(j)))$$
$$\leq 2(ad_2(j, \sigma_1(j)) + (1 - a)d_2(j, \sigma_2(j))).$$

Therefore

$$\mathbb{E}\left(\sum_{j \in \mathcal{P}} d_2(j, \sigma(j))\right) \leq 2\left(a \sum_{j \in \mathcal{P}} d_2(j, \sigma_1(j)) + (1 - a) \sum_{j \in \mathcal{P}} d_2(j, \sigma_2(j))\right).$$

It is left to prove

$$a \sum_{j \in \mathcal{P}} d_2(j, \sigma_1(j)) + (1 - a) \sum_{j \in \mathcal{P}} d_2(j, \sigma_2(j)) \leq (3 + \epsilon) \sum_{(i,j) \in M} x_{ij}^\star d_2(i, j).$$

Let $(\alpha^{(i)}, \beta^{(i)})$ be the solution for the dual facility location LP associated with the solution $(C_i, \sigma_i)$ for $i = 1, 2$. Observe that by Lemma 46 we have

$$\sum_{j \in \mathcal{P}} d_2(j, \sigma_i(j)) \leq 3\left(\sum_{j \in \mathcal{P}} \alpha_j^{(i)} - f_i k_i\right).$$

Moreover if we use $f_1 - f_2 \leq \frac{\epsilon^2 d_{\min}}{3n}$ we obtain

$$(1 - \epsilon^2) \sum_{j \in \mathcal{P}} d_2(j, \sigma_2(j)) \leq 3\left(\sum_{j \in \mathcal{P}} \alpha_j^{(2)} - f_1 k_2\right) + 3(f_1 - f_2)k_2 - \epsilon^2 \sum_{j \in \mathcal{P}} d_2(j, \sigma_2(j))$$
$$\leq 3\left(\sum_{j \in \mathcal{P}} \alpha_j^{(2)} - f_1 k_2\right) + \epsilon^2 d_{\min} - \epsilon^2 d_{\min}$$
$$= 3\left(\sum_{j \in \mathcal{P}} \alpha_j^{(2)} - f_1 k_2\right).$$

Finally we get

$$a \sum_{j \in \mathcal{P}} d_2(j, \sigma_1(j)) + (1 - a) \sum_{j \in \mathcal{P}} d_2(j, \sigma_2(j)) \leq (3 + \epsilon)\left(\sum_{j \in \mathcal{P}} a\alpha_j^{(1)} + (1 - a)\alpha_j^{(2)} - f_1 k\right)$$
$$\leq (3 + \epsilon) \sum_{(i,j) \in M} x_{ij}^\star d_2(i, j).$$

Where the last inequality follows from the fact that $(a\alpha^{(1)} + (1-a)\alpha^{(2)}, a\beta^{(1)} + (1-a)\beta^{(2)}, f_1)$ is a feasible solution to the dual of the $k$-median LP.

To bound the radius of $(C, \sigma)$ with respect to $d_1$ we observe that for every point $j \in \mathcal{P}$ we have $\sigma(j) \in \{\sigma_1(j), \sigma_2(j), c(\sigma_1(j))\}$. In the case $\sigma(j) = \sigma_i(j)$ we can use $d_1(j, \sigma_i(j)) \leq \mathsf{rad}(C_i, \sigma_i, d_1) \leq 3R$ and in the last case we use that there exists a point $j'$ with $\sigma_1(j) = \sigma_1(j)$ and $\sigma_2(j) = c(\sigma_1(j))$ and thus by triangle inequality $d_1(j, c(\sigma_1(j))) \leq d_1(j, \sigma_1(j))) + d_1(\sigma_1(j'), j') + d_1(j', \sigma_2(j)) \leq 2\mathsf{rad}(C_1, \sigma_1, d_1) + \mathsf{rad}(C_2, \sigma_2, d_1) \leq 9R$. $\qquad \square$

Theorem 41 follows immediately from the previous lemma.

The algorithm in [41] also extends to squared distances and yields a $(54 + \epsilon)$-approximation for $k$-means. Notice that the additional factor of 2 in the Euclidean space mentioned in [41] can be omitted since we restrict the set of possible centers to $\mathcal{P}$ from the beginning. Theorem 42 summarizes the results for $k$-means.

### B.4   Combinations of $k$-median and $k$-means

We first prove that the size of the Pareto set can be exponential for the combination $k$-median and $k$-means even if $d_1 = d_2$. For this purpose we present a small construction that consists of several points which should be covered by one center in the final solution. Both the optimal center and the value of the 1-median solution differ from the optimal center and the value of the 1-means solution in this construction.

Let $M \subset \mathbb{R}$ be such a construction and $d_1(x, y) = d_2(x, y) = \|x - y\|_2 = |x - y|$. In the following we denote by $c_1$, $c_2$ the optimal 1-median and the optimal 1-means center, respectively. Furthermore let $\alpha_i = \sum_{p \in M} |p - c_i|$ and $\beta_i = \sum_{p \in M} |p - c_i|^2$ for $i \in \{1, 2\}$ and assume that $\alpha_1 < \alpha_2$ and $\beta_1 > \beta_2$. An example of such a construction is given in Example 50.

We now construct an instance $\mathcal{P}$ with an exponentially large Pareto set by combining several scaled versions of $M$. Let $M_\lambda = \{2^\lambda \cdot x + \lambda \cdot \Delta \mid x \in M\}$ and $c_i^\lambda = 2^\lambda \cdot c_i + \lambda \cdot \Delta$ for $i \in \{1, 2\}$ and $\Delta \geq \delta 2^{k+1}$, where $\delta$ denotes the maximum diameter of $M$. Define $\mathcal{P}$ to be an instance consisting of $M_0, M_1, M_2, M_3, \ldots, M_{k-1}$. Thus $\mathcal{P}$ consists of $k$ substructures, each of which should be a cluster of the optimal $k$-clustering for both cost functions.

**Theorem 48.** *For every $k \in \mathbb{N}$ there exists a point set $\mathcal{P} \subset \mathbb{R}$ such that for $d_1 = d_2 = \|\cdot\|_2$ and $f_1 = \mathsf{med}, f_2 = \mathsf{mean}$ we have $|\mathbb{P}| \geq 2^k$.*

In order to prove this theorem, we look at the $2^k$ clusterings that are constructed by choosing either $c_1^i$ or $c_2^i$ as center of each of the $k$ different substructures $M_i$. More formally, define for $I \subseteq [k]$ the center set $C_I$ which contains $c_1^i$ iff $i \in I$ and $c_2^i$ otherwise. For all $p \in M_i$ we set $\sigma_I(p) = c_1^i$ if $c_1^i \in C_I$ and $\sigma_I(p) = c_2^i$ if $c_2^i \in C_I$.

The following lemma shows that every such center set induces a Pareto-optimal clustering, which directly proves the theorem.

**Lemma 49.** $(C_I, \sigma_I)$ *is Pareto-optimal for all $I \subseteq [k]$.*

*Proof.* We prove the lemma by contradiction. Assume there is an $I'$ such that $(C_{I'}, \sigma_{I'})$ dominates $(C_I, \sigma_I)$. We consider two cases.

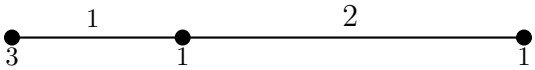

Figure 6: Example for the construction $M$. The numbers below indicate the numbers of points on the location.

**Case 1:** $\sum_{i \in I' \setminus I} 2^{i-1} > \sum_{i \in I \setminus I'} 2^{i-1}$. In this case we know

$$
\mathsf{mean}(C_I, \sigma_I, \|\cdot\|_2) = \sum_{i \in I} 2^{2i-2}\beta_1 + \sum_{i \notin I} 2^{2i-2}\beta_2
$$

$$
= \sum_{i \in I \setminus I'} 2^{2i-2}\beta_1 + \sum_{i \in I \cap I'} 2^{2i-2}\beta_1 + \sum_{i \in I' \setminus I} 2^{2i-2}\beta_2 + \sum_{i \notin I \cup I'} 2^{2i-2}\beta_2
$$

$$
< \sum_{i \in I' \setminus I} 2^{2i-2}\beta_1 + \sum_{i \in I \cap I'} 2^{2i-2}\beta_1 + \sum_{i \in I \setminus I'} 2^{2i-2}\beta_2 + \sum_{i \notin I \cup I'} 2^{2i-2}\beta_2
$$

$$
= \sum_{i \in I'} 2^{2i-2}\beta_1 + \sum_{i \notin I'} 2^{2i-2}\beta_2
$$

$$
= \mathsf{mean}(C_{I'}, \sigma_{I'}, \|\cdot\|_2)
$$

where the inequality holds because $\sum_{i \in I' \setminus I} 2^{i-1} > \sum_{i \in I \setminus I'} 2^{i-1}$ and the $\beta_1 > \beta_2$. This contradicts the assumption.

**Case 2:** $\sum_{i \in I' \setminus I} 2^{i-1} < \sum_{i \in I \setminus I'} 2^{i-1}$. Here we analogously get $\mathsf{med}(C_I, \sigma_I, \|\cdot\|_2) < \mathsf{med}(C_{I'}, \sigma_{I'}, \|\cdot\|_2)$, which also contradicts the assumption.

By the construction of $\mathcal{P}$ we know that $(C_I, \sigma_I)$ cannot be dominated by any other solution $(C, \sigma)$. It is easy to observe that if $M_i$ contains a center that it is always better to assign all other points within this center independently whether we consider the k-median or $k$-means objective. Thus for any Pareto-optimal solution $(C, \sigma)$ that dominates a solution $(C_I, \sigma_I)$ there exists a non-empty set $U$ of indices such that for all $i \in U$ there is no center chosen within $M_i$. At the same time the other $k - |U|$ scaled versions of $M$ contain $|U|$ additional centers. Thus we might remove arbitrary centers from these until every $M_j$ contains only a single center, choose an arbitrary center in $M_i$ for any $i \in U$ and for all $j \in \{0, ..., k-1\}$ reassign all points in $M_j$ to the respective center. For each index $i \in U$ the $k$-median assignment cost of any point in $M_i$ gets reduced from at least $\Delta - \delta \cdot 2^{k-1}$ to at most $\delta 2^{k-1}$. At the same time the assignment cost of any point contained in a cluster where a point gets removed increases by at most $\delta \cdot 2^{k-1}$ while the cost of the remaining nodes stays the same. Thus the $k$-median cost decreases by at least $|M||U|(\Delta - 3\delta 2^{k-1}) > 0$. A similar statement can be shown for the $k$-means objective. Thus the respective solution would not have been Pareto-optimal which is a contradiction which means that for any $I$ the solution $(C_I, \sigma_I)$ is Pareto-optimal and the lemma follows.

$\square$

In the following we see one possible example for the construction $M$.

**Example 50.** *Consider 5 points on a line, where the distance between the first three points and the second point is 1 and the distance between the second and the third point is 2 (see Figure 6). Then the optimal median center is one of the leftmost points, while the optimal mean center is the single point in the middle. Therefore $\alpha_1 = 4$, $\alpha_2 = 5$, $\beta_1 = 10$ and $\beta_2 = 7$, fulfilling the prerequisites on $M$.*

Even though the Pareto set of this combination may have exponential size, Papadimitrou and Yannakakis [49] showed that there always exists an $(1 + \epsilon, 1 + \epsilon)$-approximate Pareto set, whose size is polynomial in the input size and $\frac{1}{\epsilon}$. They furthermore showed that there is a polynomial algorithm to construct such an $(1 + \epsilon, 1 + \epsilon)$-approximate Pareto set if and only if one is able to solve the following problem in polynomial time: Given the instance and $b_1, b_2 \in \mathbb{R}$, either find a solution $s$ with $f_i(s) \leq b_i$ for $i \in \{1, 2\}$ or report that there is no solution $s$ with $(1 + \epsilon)f_i(s) \leq b_i$ for $i \in \{1, 2\}$. Unfortunately, for $f_1, f_2 \in \{\mathsf{med}, \mathsf{mean}\}$ it is unknown whether this problem can be solved efficiently.

From now on, we focus on finding an approximate convex Pareto set instead. Since in the following we only consider cases where $\alpha_1 = \alpha_2$, we write $\mathbb{CP}_\alpha$ instead of $\mathbb{CP}_{\alpha,\alpha}$ for convenience.

**Theorem 51.** *Given a finite set $\mathcal{P}$, metrics $d_1, d_2$ on $\mathcal{P}$ and two objectives $f_1, f_2 \in \{\mathsf{med}, \mathsf{mean}\}$. Let $\epsilon > 0$. Then we can compute an $(\delta + \epsilon)$-approximate convex Pareto set $\mathbb{CP}_{\delta+\epsilon}$ in time that is polynomial in the input size and $\frac{1}{\epsilon}$, where $\delta = 2.67059$ if $f_1 = \mathsf{med}$ and $f_2 = \mathsf{med}$, and $\delta = 9 + \epsilon$ if $f_1 = \mathsf{med}$ and $f_2 = \mathsf{mean}$ or $f_1 = \mathsf{mean}$ and $f_2 = \mathsf{mean}$. The size of this set $|\mathbb{CP}_{\delta+\epsilon}|$ is also polynomial in $\frac{1}{\epsilon}$ and $|P|$.*

To do this, we make use of a scheme presented by Diakonikolas [22]. The idea is essentially to reduce a given multi-objective optimization problem to an associated problem with a single, combined objective.

More formally, let $f_1$, $f_2$ be two minimization objectives, and $w = (w_1, w_2) \in \mathbb{R}^2_+$ a weight vector. Then we define the related Comb-problem to be the problem that seeks to minimize the weighted combination of both objectives $w \cdot f = w_1 \cdot f_1 + w_2 \cdot f_2$.

We denote by $\mathsf{Comb}_\delta(w)$ a routine that returns a $\delta$-approximate solution to the Comb-problem with weight vector $w$. We will show that a polynomial time $\mathsf{Comb}_\delta(w)$ routine is enough to compute a $(\delta \cdot (1 + \epsilon))$−approximate convex Pareto set to the bi-objective original problem $\Pi$ for all $\epsilon > 0$. To achieve this, we will describe a scheme which returns such a set where the number of calls to the $\mathsf{Comb}_\delta$ routine is polynomial in $\frac{1}{\epsilon}$ and the input size.

The scheme and its proof are basically identical to the one Diakonikolas describes in his thesis [22], where he showed that an (F)PTAS for the Comb-problem is sufficient to obtain an (F)PTAS to the convex Pareto set. In the following we show that the proof still works, if instead of an (F)PTAS for the Comb-routine we only have an approximate $\mathsf{Comb}_\delta$-routine.

A point $s \in \mathbb{R}^2_+$ is called approximately balanced if both its coordinates are within factor 2 of each other. We will first construct a set to $(\delta \cdot (1 + \epsilon))$-cover all approximately balanced solution points $\mathcal{S}_{\mathrm{bal}}$, and later use the fact that approximate convex Pareto sets are invariant under the scaling of their objectives to cover all solution points.

First we observe that for all $\epsilon > 0$ there is a set of size $\mathrm{O}(\frac{1}{\epsilon})$ that $(1 + \epsilon)$-covers all approximately balanced solution points. We want to compute a $\delta$-approximation to such a set using only $\mathrm{O}(\frac{1}{\epsilon})$ many calls to $\mathsf{Comb}_\delta$. Therefore we are looking for a set of weight vectors $\mathcal{W}_{\mathrm{bal}}$ with $|\mathcal{W}_{\mathrm{bal}}| \in \mathrm{O}(\frac{1}{\epsilon})$ whose corresponding solution set $\mathcal{Q}_{\mathrm{bal}} = \{\mathsf{Comb}_\delta(w) \mid w \in \mathcal{W}_{\mathrm{bal}}\}$ contains a $(\delta \cdot (1 + \epsilon))$-approximate convex Pareto set of $\mathcal{S}_{\mathrm{bal}}$.

We will now define the set $\mathcal{W}_{\mathrm{bal}}$ in a way, that is only dependent on $\epsilon$, but not on the given problem instance. For that purpose, we set $M = \lceil \frac{2}{\epsilon} \rceil$. Let $\mathcal{W}_{\mathrm{bal}} = \{(1, \frac{l}{M}), (\frac{l}{M}, 1) \mid l \in [M]\}$. It is easy to see that $|\mathcal{W}_{\mathrm{bal}}| \in \mathrm{O}(\frac{4}{\epsilon})$. Therefore we can compute $\mathcal{Q}_{\mathrm{bal}}$ with $\mathrm{O}(\frac{4}{\epsilon})$ many calls to $\mathsf{Comb}_\delta$. Let $\mathcal{Q}^*_{\mathrm{bal}}$ be the convex Pareto set of $\mathcal{Q}_{\mathrm{bal}}$.

The following lemma shows that the set $\mathcal{Q}^*_{\mathrm{bal}}$ is in fact a $(\delta \cdot (1 + \epsilon))$-approximate convex Pareto set for all approximately balanced solution points.

**Lemma 52.** *For every approximately balanced solution point $s \in \mathcal{S}_{bal}$, there is a convex combination of points in $\mathcal{Q}^*_{bal}$ that $(\delta \cdot (1 + \epsilon))$-covers it.*

*Proof.* We first prove the following claim: For any weight vector $w \in \mathbb{R}^2_+$ there is a solution $q^* \in \mathcal{Q}^*_{\mathrm{bal}}$ such that $w \cdot q^* \leq \delta(1 + \epsilon)w \cdot s$.
Let $w = (w_1, w_2)$ be an arbitrary weight vector. By symmetry and by scaling of the weights we can without loss of generality assume that $w_2 \leq w_1 = 1$. Let $w^*$ be $(w_1, w_2^*)$, where $w_2^*$ arises from $w_2$ by rounding up to the next higher multiple of $\frac{1}{M}$. It follows that $w^* \in W_{\mathrm{bal}}$ and therefore was considered by our scheme. Let $q^*$ be the solution point given by $\mathsf{Comb}_\delta(w^*)$.

By the definitions of $w^*$ and of $\mathsf{Comb}_\delta$ the following holds:

$$
\begin{aligned}
wq^* &\leq w^*q^* \\
&\leq \delta \cdot w^*s \\
&= \delta \cdot (w_1 \cdot s_1 + w_2^* \cdot s_2) \\
&\leq \delta \cdot \left(w_1 \cdot s_1 + \left(w_2 + \frac{1}{M}\right) \cdot s_2\right) \\
&= \delta \cdot (ws + \frac{1}{M}s_2) \\
&\leq \delta \cdot (ws + \frac{2}{M}s_1) \qquad\qquad (2) \\
&\leq \delta \cdot (ws + \frac{2}{M}ws) \qquad\qquad (3) \\
&\leq \delta \cdot (1 + \epsilon)ws,
\end{aligned}
$$

where inequality (2) follows from the fact that $s$ is approximately balanced and inequality (3) follows from the fact that $w_1 = 1$ and therefore $s_1$ contributes directly to $ws$. This proves the claim.

Let $q = (q_1, q_2)$ be the point that lies on the border of the convex hull of $\mathcal{Q}^*_{\mathrm{bal}}$ and fulfills $\frac{q_1}{q_2} = \frac{s_1}{s_2}$ (therefore $q$ intersects the line between $s$ and the origin).
Let $q^l, q^r$ be the solutions in $\mathcal{Q}^*_{\mathrm{bal}}$ such that $q_1^l \leq q_1 \leq q_1^r$ and there is no $p \in \mathcal{Q}^*_{\mathrm{bal}}$ with $q_1^l \leq p \leq q_1^r$.
Let $(1, w^l), (1, w^r)$ be the weights that generated $q^l$ and $q^r$, respectively.

Then we know that for all weights between $w^l$ and $w^r$ either $q^l$ or $q^r$ are approximately optimal (since $w \cdot p$ is linear in $p$ and $w$ for all $w, p$). Therefore there is a $w = (1, w^*)$ with $w^r \leq w^* \leq w^l$ such that $w \cdot q^l = w \cdot q^r$ and both points are approximately optimal for this weight. Since $q$ is a convex combination of $q^l$ and $q^r$ we know that $w \cdot q = w \cdot q^l \leq \delta(1 + \epsilon)w \cdot s$. Together with $\frac{q_1}{q_2} = \frac{s_1}{s_2}$ it follows that $s$ is approximately dominated by $q$ which is a convex combination of $q^l, q^r \in \mathcal{Q}^*_{\mathrm{bal}}$.

$\square$

Since normally not all solution points are approximately balanced, $\mathcal{Q}^*_{\mathrm{bal}}$ might not cover all solution points. Diakonikolas shows that this can be taken care of by looking at multiple different scalings of the objective functions such that for each point $s$ of the solution space there is one scaling such that $s$ is approximately balanced in the scaled version. He furthermore shows that a polynomial number of scalings is sufficient.

This proves that for all $\epsilon > 0$ one can construct a $((1 + \epsilon) \cdot \delta)$-convex Pareto set whose size is polynomial in $\epsilon$ if there is a polynomial $\mathsf{Comb}_\delta$-routine. The runtime of this construction is also polynomial in $\frac{1}{\epsilon}$ for two objectives.

All that is left to do in order to prove the theorem is to show that there exist $\mathsf{Comb}$ routines that yield the desired approximation factors to the different combinations of objective functions.

**Lemma 53.** *Let $d_1, d_2$ be two metrics. Then the following hold:*

1. *There is a $\mathsf{Comb}_{2.67059}$-routine for $f_1 = \mathrm{med}$ and $f_2 = \mathrm{med}$.*

2. *For all $\epsilon' > 0$ there is a $\mathsf{Comb}_{9+\epsilon'}$-routine for $f_1 = \mathrm{med}$ and $f_2 = \mathrm{mean}$.*

3. *For all $\epsilon' > 0$ there is a $\mathsf{Comb}_{9+\epsilon'}$-routine for $f_1 = \mathrm{mean}$ and $f_2 = \mathrm{mean}$.*

*Proof.*  1. Since for any weight vector $w \in \mathbb{R}^2_+$ the function $d' = w_1 \cdot d_1 + w_2 \cdot d_2$ is again a metric, the Comb-problem can be solved by any approximation algorithm for the $k$-median problem under general metrics. The currently best known algorithm is from Cohen-Addad et al. [16] and achieves an approximation factor of $2.67059$.

2. The algorithm for k-means in general metrics by Ahmadian et al. [4] extends to all sum-based cost functions of the form $\sum_{p \in \mathcal{P}} c(p, \sigma(p))$ whose distance function $c(i, j)$ fulfills the inequality $c(i, j) \leq 3(c(i, k) + c(k, l) + c(l, j))$. The combined objective for the $k$-means and $k$-median cost-functions for an arbitrary weight vector $w \in \mathbb{R}^2_+$ is given by $w_1 \cdot \mathrm{med}(C, \sigma, d_1) + w_2 \cdot \mathrm{mean}(C, \sigma, d_2) = w_1 \cdot \sum_{p \in \mathcal{P}} d_1(p, \sigma(p)) + w_2 \cdot \sum_{p \in \mathcal{P}} d_2^2(p, \sigma(p)) = \sum_{p \in \mathcal{P}} w_1 \cdot d_1(p, \sigma(p)) + w_2 \cdot d_2^2(p, \sigma(p))$. Thus all we have to do in order to apply the algorithm to our objective is to show that $c(i, j) = w_1 \cdot d_1(i, j) + w_2 \cdot d_2^2(i, j)$ fulfills the necessary property, which is done in the following:

$$\begin{aligned}
c(i, j) &= w_1 d_1(i, j) + w_2 d_2^2(i, j) \\
&= w_1(d_1(i, k) + d_1(k, l) + d_1(l, j)) + w_2(d_2(i, k) + d_2(k, l) + d_2(l, j))^2 \\
&\leq w_1(d_1(i, k) + d_1(k, l) + d_1(l, j)) + 3w_2(d_2^2(i, k) + d_2^2(k, l) + d_2^2(l, j)) \\
&\leq 3(w_1 d_1(i, k) + w_2 d_2^2(i, k) + w_1 d_1(k, l) + w_2 d_2^2(k, l) + w_1 d_1(l, j) + w_2 d_2^2(l, j)) \\
&= 3(c(i, k) + c(k, l) + c(l, j)).
\end{aligned}$$

Therefore we also achieve the factor $9 + \epsilon'$ for the combined cost function using the algorithm in [4].

3. This works analogously to 2., by setting $c(i,j) = w_1 \cdot d_1^2(i,j) + w_2 \cdot d_2^2(i,j)$ instead.

$\square$

The theorem now follows directly from Lemma 53 and the factor achieved by our scheme.

## C  More Details on Experimental Results

### C.1  Improving $k$-means++ by combination with separation

We test the approach to combine $k$-means with separation presented in Chapter A.3 on data sets with ground truth inputs [2, 26, 6, 1, 3, 34, 32, 33]. We hope to improve the quality of the solution found by $k$-means++ by a combination with Single Linkage. We evaluate the best solution (compared to the ground truth) found by our approach in comparison to solutions found by $k$-means++ [8] and Single Linkage [52]. Since all tested data sets are provided with a ground truth, we fix the number of clusters $k$ for the approximate Pareto set to be the desired number of clusters in the ground truth.

**Algorithm.**  For every data set, we compute the approximate Pareto set for the desired number of clusters as follows. Recall that for a data set of size $n$ there are only $O(n^2)$ possible values for the separation. We compute these values in time $O(n^2 d)$ and sort them in increasing order in time $O(n^2 \log(n))$. Here $d$ is the dimension of the data set. Starting with separation 0, we increase the separation in every step to the next largest value. Suppose the separation is $\Delta$ in the current step, then we merge all points whose distance is at most $\Delta$. This can be done efficiently via a Union Find data structure. Since the resulting clustering may have more than $k$ clusters, we have to reduce the number of clusters to $k$. For data sets in $\mathbb{R}^d$ and the $k$-means objective, one can replace every cluster by its centroid weighted by the number of points in the cluster and then cluster these weighted centroids instead of using the nesting technique of Lin et al. [46] for general metrics. Instead of choosing the theoretically best approximation algorithm for $k$-means, we use $k$-means++ [8] to cluster the centroids as it is known to be fast (running time $O(nkd)$) and to produce solutions of high quality for the $k$-means problem in practice. Then the respective clustering on the original data set has separation at least $\Delta$ and at most $k$ clusters. One can show that this algorithm computes an $\alpha$-approximate Pareto with $\alpha = (1, O(\log k))$.

**Evaluation.**  We compare the clusterings in the approximate Pareto set to the ground truth. For this purpose we compute the Normalized Mutual Information, Rand Index, and $F_\beta$-scores for $\beta = 0.5, 1, 2$. We briefly recall their definition below. For every measure we pick the clustering in the approximate Pareto set that has highest value with respect to this measure. We compare our results with two variants of $k$-means++ and Single Linkage. In the first variant we do not fix the number of clusters, i.e., $k$-means++ and Single Linkage are performed for every $k = 1, \ldots, n$ and we compute the highest values achieved by these clusterings. In the second variant we fix $k$ to the desired number of clusters in the ground truth and compute the respective values for all measures. Since $k$-means++ and our algorithm are randomized, the depicted values are the average of the maximum values achieved in 20 runs. The results are in Figure 11.

**Data Sets.**  We tested the algorithms on the following data sets.

- Iris [26]: 150 points, $d = 4, k = 3$
- Dry Bean [2]: 13611 points, $d = 16, k = 7$
- Rice [1]: 3810 points, $d = 7, k = 2$
- Wine [3]: 178 points, $d = 13, k = 3$
- Optdigits [6]: 5620 points, $d = 64, k = 10$
- 2d-4c-no3 [34] (Figure 7a): 1123 points, $d = 2, k = 4$
- 2d-10c-no3 [32] (Figure 7b): 3359 points, $d = 2, k = 10$
- 2d-10c-no4 [33] (Figure 7c): 3291 points, $d = 2, k = 10$

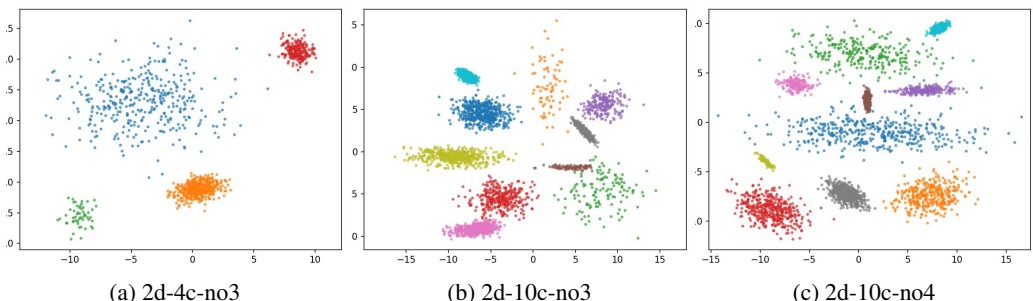

|     |     |     |
|-----|-----|-----|
| (a) 2d-4c-no3 | (b) 2d-10c-no3 | (c) 2d-10c-no4 |

Figure 7: Visualization of the ground truth for the three synthetic data sets.

**Results.** The results show that for all data sets except rice and all measures the best solution in the approximate Pareto set has a higher value than the solutions found by Single Linkage and $k$-means++ individually for the same number of clusters. In some cases $k$-means++ and Single Linkage provide better results when allowing different numbers of clusters. Especially Single Linkage does provide better results when we allow for more clusters, since it is sensitive to outliers. We see for example on the data set 2d-10c-no4 in Figure 8 that Single Linkage reconstructs all of the larger clusters when performed for an appropriate separation value and performs better than $k$-means++ with $k = 10$ (Figure 8a). However the number of clusters is large ($k = 85$) since there are some outliers which are in separate clusters. If we perform Single Linkage for $k = 10$ we see that the result is two large clusters containing most of the points and 8 clusters with at most 2 points each. The best solution in the approximate Pareto set (Figure 8d) mostly coincides with the solution found by Single Linkage in Figure 8c, however it has only 10 clusters. A similar behavior can also be observed on the data sets 2d-4c-no3 and 2d-10c-no3 in Figure 9 and Figure 10.

Based on our experiments the combination of $k$-means++ and Single Linkage can be useful to

- reduce the number of clusters found by Single Linkage or
- improve the quality of the $k$-means++ solution.

On all data sets that are not synthetic, the best result in the approximate Pareto set has a small separation, as we see in Figure 12. Therefore we assume that it is sufficient to execute the algorithm for small separation values instead of computing the whole approximate Pareto set to improve the quality of the solution. Notice that on the data sets rice, wine, and optdigits the best solutions on the approximate Pareto set for some measures are the ones with smallest separation. However it does not necessarily coincide with the solution found by $k$-means++ and in fact performs better with respect to NMI, RI, and $F_\beta$-scores. For a visualization of NMI on the approximate Pareto set we refer to Figure 13. In Figure 14 one can see the variance of the measures for $k$-means++ and the approximate Pareto set.

**Resources and running time.** The experiments were performed on a machine with a 2.7GHz AMD Ryzen 7 PRO 6850U processor and 32GB RAM. The source codes are in C++ and available at https://github.com/algo-hhu/paretoClustering in folder kMeans_SL. The first draft of the $k$-means++ algorithm and the Union Find data structure were produced using ChatGPT 4.0, but then adapted and extended for our needs. The code was compiled using g++ version 11.4.0 with optimization flag -O2. Since the experiments are performed for a proof of concept and are not optimized for running time, the running time was not tracked. The computation of the approximate Pareto set and the evaluation with respect to all measures took at most 15 minutes per data set.

**Measures.** The following measures are used to evaluate the similarity between the ground truth and the computed clustering. The measures take values in $[0, 1]$, where a value of 1 is achieved when the computed clustering matches the ground truth. For a detailed survey on such measures we refer to [54]. Let $(C, \sigma)$ be the computed clustering and $(C^*, \sigma^*)$ be the ground truth. For $i \in C, j \in C^*$ let

- $\mathcal{P}_{ij} = \{p \in \mathcal{P} \mid \sigma(p) = i, \sigma^*(p) = j\}$ be the set of points assigned to center $i$ by the computed clustering and $j$ by the ground truth,

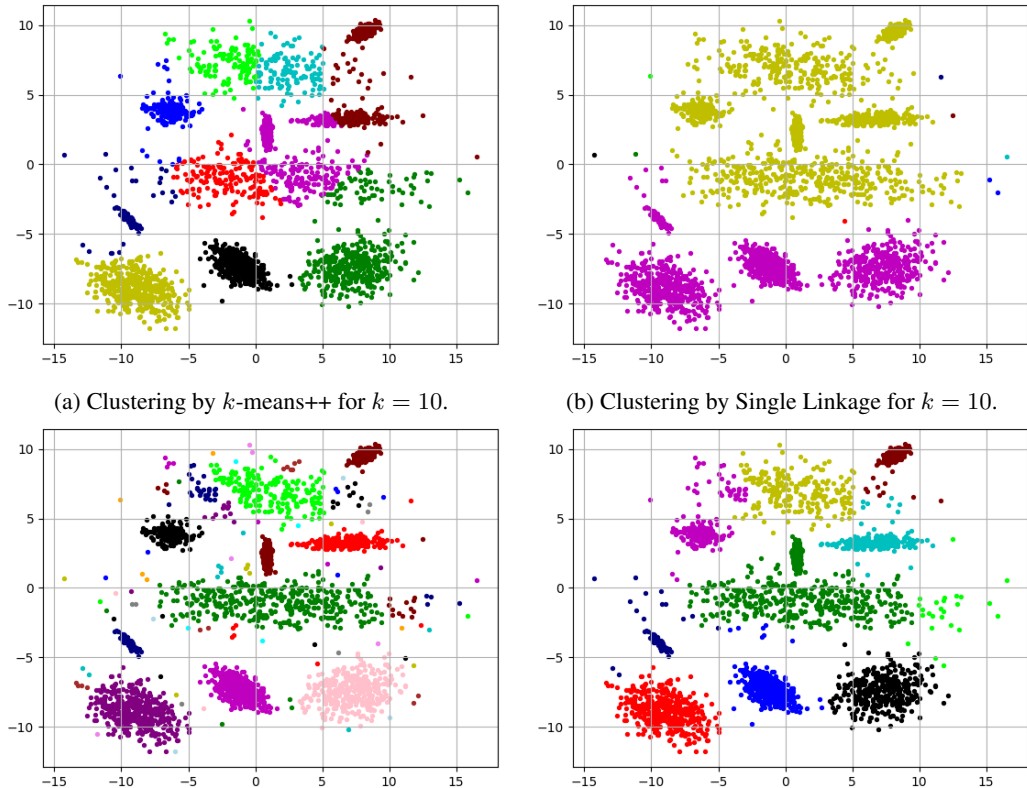

(a) Clustering by $k$-means++ for $k = 10$.

(b) Clustering by Single Linkage for $k = 10$.

(c) Clustering by Single Linkage for separation $\Delta = 0.67$ and $k = 85$.

(d) Clustering on the approximate Pareto curve for $k = 10$.

Figure 8: Clusterings computed on data set 2d-10c-no4.

- $\mathcal{P}_{*j} = \{p \in \mathcal{P} \mid \sigma^*(p) = j\}$ be the set of points assigned to $j$ by the ground truth,
- $\mathcal{P}_{i*} = \{p \in \mathcal{P} \mid \sigma(p) = i\}$ be the set of points assigned to $i$ by the computed clustering,
- $p_{ij} = \frac{|\mathcal{P}_{ij}|}{n}$, $p_{*j} = \frac{|\mathcal{P}_{*j}|}{n}$ and $p_{i*} = \frac{|\mathcal{P}_{i*}|}{n}$.

**Rand Index:** The Rand Index is defined by Rand [51]. Let the set of *true positives* be $TP = \{\{i, j\} \mid i, j \in \mathcal{P}, i \neq j, \sigma(i) = \sigma(j)$ and $\sigma^*(i) = \sigma^*(j)\}$, i.e., it contains pairs of points which are contained in the same cluster both in $(C, \sigma)$ and $(C^*, \sigma^*)$. Furthermore let the set of *true negatives* be $TN = \{\{i, j\} \mid i, j \in \mathcal{P}, i \neq j, \sigma(i) \neq \sigma(j)$ and $\sigma^*(i) \neq \sigma^*(j)\}$, i.e., it contains pairs of points which are contained in different clusters both in $(C, \sigma)$ and $(C^*, \sigma^*)$. The Rand Index is defined as

$$\mathsf{RI}(C, \sigma) = \frac{2(|TP| + |TN|)}{n(n-1)}.$$

$\mathsf{F}_\beta$**-score:** The F-score was introduced in the context of clustering by Larsen and Aone [43]. For centers $i \in C$ and $j \in C^*$, we define the *precision* as $\mathsf{Prec}(i, j) = \frac{|\mathcal{P}_{ij}|}{|\mathcal{P}_{i*}|}$ and the *recall* as $\mathsf{Rec}(i, j) = \frac{|\mathcal{P}_{ij}|}{|\mathcal{P}_{*j}|}$.

We combine their definition with the more general definition of an $\mathsf{F}_\beta$-score in binary classification by van Rijsbergen [53] and obtain the $\mathsf{F}_\beta$-score of a cluster $i$ and $j$ as

$$F_\beta(i, j) = (1 + \beta^2) \frac{\mathsf{Prec}(i, j) \cdot \mathsf{Rec}(i, j)}{\beta^2 \cdot \mathsf{Prec}(i, j) + \mathsf{Rec}(i, j)}.$$

The $\mathsf{F}_\beta$-score of $(C, \sigma)$ is then defined as

$$\mathsf{F}_\beta(C, \sigma) = \frac{1}{n} \sum_{j \in C^*} |\mathcal{P}_{*j}| \max_{i \in C} \mathsf{F}_\beta(i, j).$$

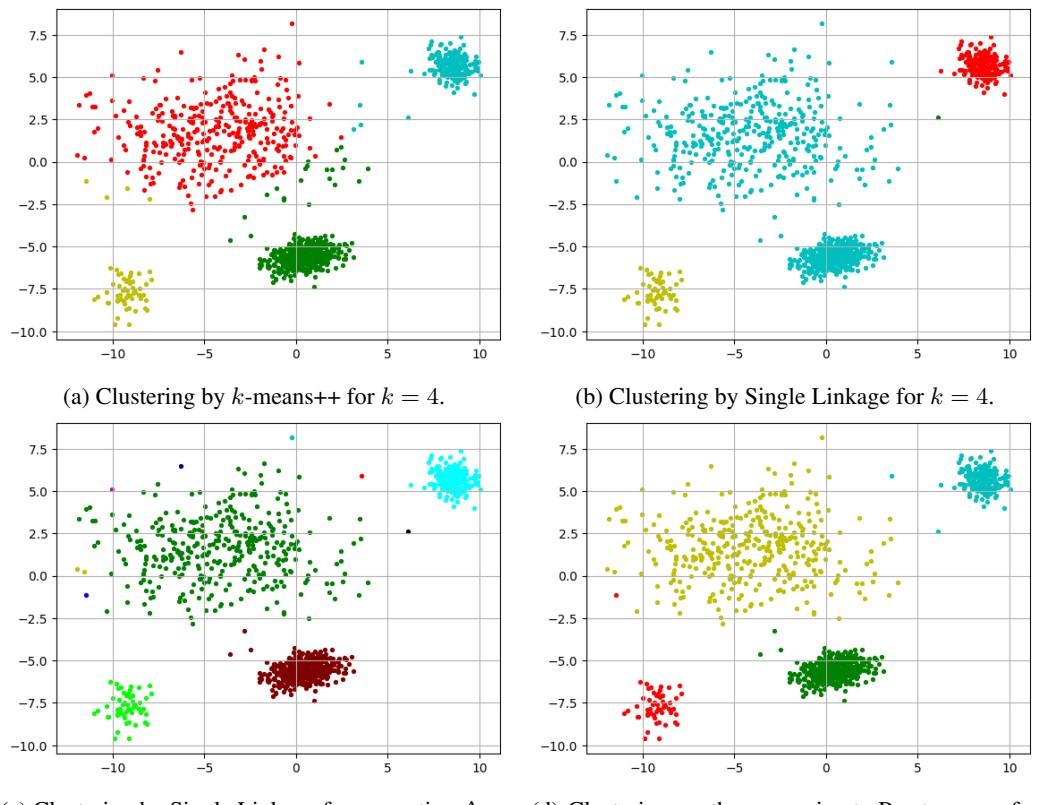

(a) Clustering by $k$-means++ for $k = 4$.

(b) Clustering by Single Linkage for $k = 4$.

(c) Clustering by Single Linkage for separation $\Delta = 1.37088$ and $k = 11$.

(d) Clustering on the approximate Pareto curve for $k = 4$.

Figure 9: Clusterings computed on data set 2d-4c-no3.

**Normalized Mutual Information:** We also consider the Normalized Mutual Information as defined by Fred and Jain [27].

$$\mathsf{NMI}(C, \sigma) = \frac{2}{\mathsf{H}(C, \sigma) + \mathsf{H}(C^*, \sigma^*)} \sum_{i \in C} \sum_{j \in C^*} p_{ij} \log\left(\frac{p_{ij}}{p_{*j} p_{i*}}\right),$$

where $\mathsf{H}(C, \sigma) = -\sum_{i \in C} p_{i*} \log(p_{i*})$ and $\mathsf{H}(C^*, \sigma^*) = -\sum_{i \in C^*} p_{*j} \log(p_{*j})$ are the entropies of the clusterings $(C, \sigma)$ and $(C^*, \sigma^*)$.

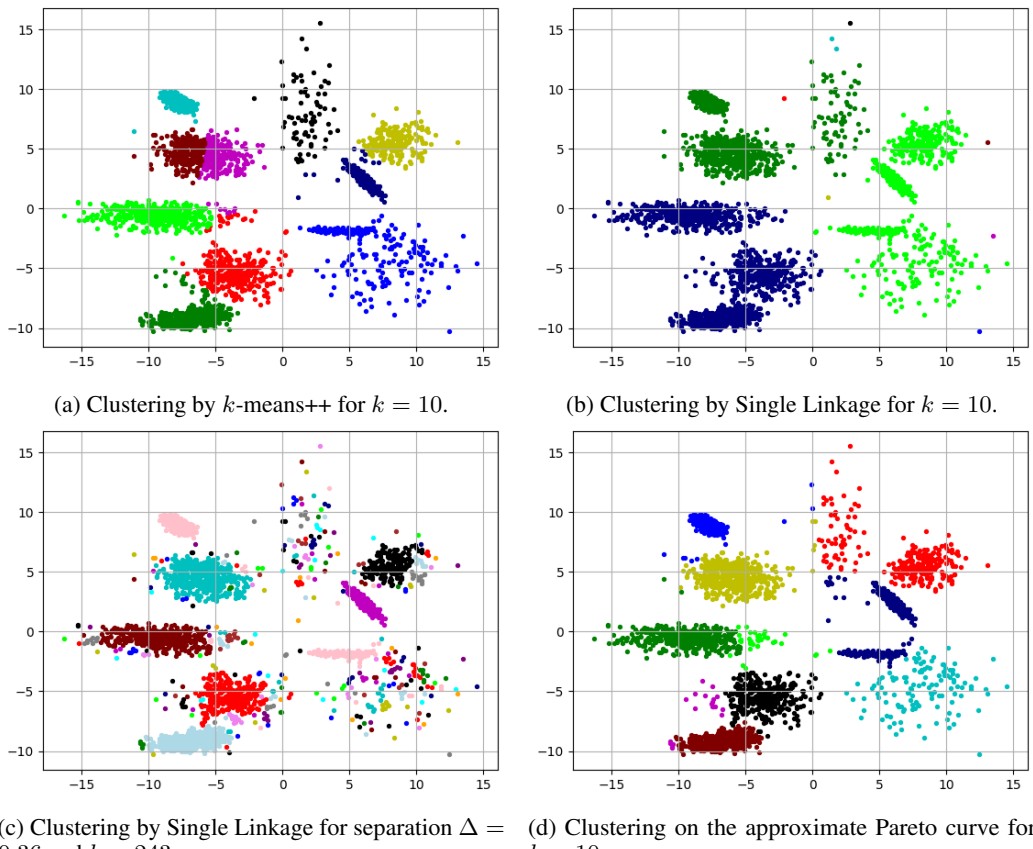

(a) Clustering by $k$-means++ for $k = 10$.

(b) Clustering by Single Linkage for $k = 10$.

(c) Clustering by Single Linkage for separation $\Delta = 0.36$ and $k = 243$.

(d) Clustering on the approximate Pareto curve for $k = 10$.

Figure 10: Clusterings computed on data set 2d-10c-no3.

| iris | | | | | |
|---|---|---|---|---|---|
| Measure | k-Means++ | kMeans++ with k=3 | Single Linkage | Single Linkage with k=3 | Pareto front with k=3 |
| RI | 0.8734 | 0.8683 | 0.8836 | 0.3289 | 0.9379 |
| F1 | 0.8829 | 0.8780 | 0.8838 | 0.5000 | 0.9481 |
| F2 | 0.8865 | 0.8807 | 0.8412 | 0.7143 | 0.9476 |
| F0.5 | 0.8964 | 0.8827 | 0.9349 | 0.3846 | 0.9512 |
| NMI | 0.7452 | 0.7405 | 0.7070 | 0.0000 | 0.8578 |

| dry-bean | | | | | |
|---|---|---|---|---|---|
| Measure | k-Means++ | kMeans++ with k=7 | Single Linkage | Single Linkage with k=7 | Pareto front with k=7 |
| RI | 0.8279 | 0.8026 | 0.8274 | 0.2470 | 0.8069 |
| F1 | 0.6016 | 0.5944 | 0.3975 | 0.3348 | 0.5968 |
| F2 | 0.7091 | 0.6339 | 0.5534 | 0.5369 | 0.6453 |
| F0.5 | 0.6130 | 0.5713 | 0.3647 | 0.2503 | 0.5772 |
| NMI | 0.5277 | 0.5164 | 0.3753 | 0.1626 | 0.5231 |

| rice | | | | | |
|---|---|---|---|---|---|
| Measure | k-Means++ | kMeans++ with k=2 | Single Linkage | Single Linkage with k=2 | Pareto front with k=2 |
| RI | 0.7892 | 0.7892 | 0.6325 | 0.5104 | 0.7888 |
| F1 | 0.8794 | 0.8794 | 0.6729 | 0.6728 | 0.8792 |
| F2 | 0.8796 | 0.8796 | 0.8353 | 0.8351 | 0.8794 |
| F0.5 | 0.8802 | 0.8802 | 0.7123 | 0.5647 | 0.8800 |
| NMI | 0.4693 | 0.4693 | 0.2334 | 0.0007 | 0.4728 |

| wine | | | | | |
|---|---|---|---|---|---|
| Measure | k-Means++ | kMeans++ with k=3 | Single Linkage | Single Linkage with k=3 | Pareto front with k=3 |
| RI | 0.7148 | 0.7105 | 0.6862 | 0.3628 | 0.7199 |
| F1 | 0.7145 | 0.6919 | 0.6804 | 0.5029 | 0.7241 |
| F2 | 0.8123 | 0.6982 | 0.7676 | 0.7036 | 0.7771 |
| F0.5 | 0.7224 | 0.7034 | 0.6452 | 0.3921 | 0.7440 |
| NMI | 0.4278 | 0.4265 | 0.4105 | 0.0615 | 0.4400 |

| optdigits | | | | | |
|---|---|---|---|---|---|
| Measure | k-Means++ | kMeans++ with k=10 | Single Linkage | Single Linkage with k=10 | Pareto front with k=10 |
| RI | 0.9528 | 0.9320 | 0.9024 | 0.0998 | 0.9390 |
| F1 | 0.8322 | 0.7854 | 0.1818 | 0.1818 | 0.8137 |
| F2 | 0.8216 | 0.7992 | 0.3572 | 0.3572 | 0.8246 |
| F0.5 | 0.8748 | 0.7864 | 0.2355 | 0.1220 | 0.8149 |
| NMI | 0.7846 | 0.7459 | 0.4627 | 0.0000 | 0.7627 |

| 2d-4c-no3 | | | | | |
|---|---|---|---|---|---|
| Measure | k-Means++ | kMeans++ with k=4 | Single Linkage | Single Linkage with k=4 | Pareto front with k=4 |
| RI | 0.9645 | 0.9645 | 0.9943 | 0.7243 | 0.9941 |
| F1 | 0.9726 | 0.9726 | 0.9960 | 0.7538 | 0.9955 |
| F2 | 0.9725 | 0.9725 | 0.9952 | 0.8759 | 0.9955 |
| F0.5 | 0.9732 | 0.9732 | 0.9968 | 0.6724 | 0.9956 |
| NMI | 0.9052 | 0.9052 | 0.9792 | 0.7260 | 0.9779 |

| 2d-10c-no3 | | | | | |
|---|---|---|---|---|---|
| Measure | k-Means++ | kMeans++ with k=10 | Single Linkage | Single Linkage with k=10 | Pareto front with k=10 |
| RI | 0.9795 | 0.9655 | 0.9850 | 0.7943 | 0.9894 |
| F1 | 0.9209 | 0.8803 | 0.9116 | 0.5431 | 0.9684 |
| F2 | 0.9294 | 0.8895 | 0.9248 | 0.7302 | 0.9698 |
| F0.5 | 0.9409 | 0.8867 | 0.9382 | 0.4374 | 0.9686 |
| NMI | 0.9260 | 0.9065 | 0.9118 | 0.6763 | 0.9491 |

| 2d-10c-no4 | | | | | |
|---|---|---|---|---|---|
| Measure | k-Means++ | kMeans++ with k=10 | Single Linkage | Single Linkage with k=10 | Pareto front with k=10 |
| RI | 0.9724 | 0.9597 | 0.9906 | 0.6056 | 0.9817 |
| F1 | 0.8874 | 0.8608 | 0.9715 | 0.3735 | 0.9410 |
| F2 | 0.8863 | 0.8707 | 0.9579 | 0.5825 | 0.9495 |
| F0.5 | 0.9157 | 0.8691 | 0.9871 | 0.2767 | 0.9447 |
| NMI | 0.8933 | 0.8715 | 0.9409 | 0.4631 | 0.9267 |

Figure 11: Highest values achieved by $k$-means++ for arbitrary $k$, $k$-means++ with fixed $k$, Single Linkage for arbitrary $k$, Single Linkage with fixed $k$ and approximate Pareto set with fixed $k$. Whenever $k$ is fix, we fix it to the cluster number in the ground truth.

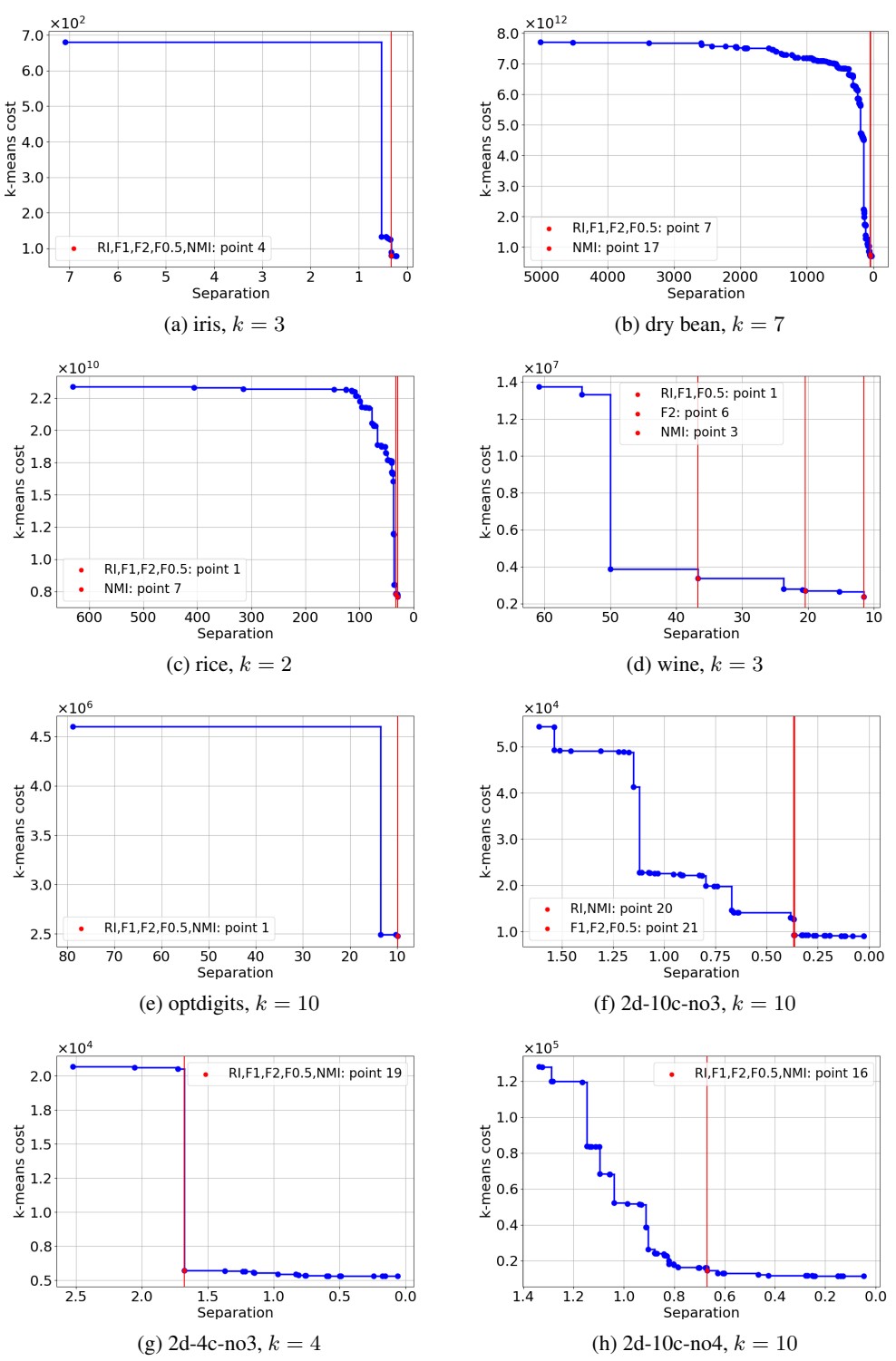

Figure 12: Approximate Pareto curves. The highlighted points show the solutions on the curve with highest NMI, RI, $F_\beta$-scores. The points on the curve are numbered from right to left.

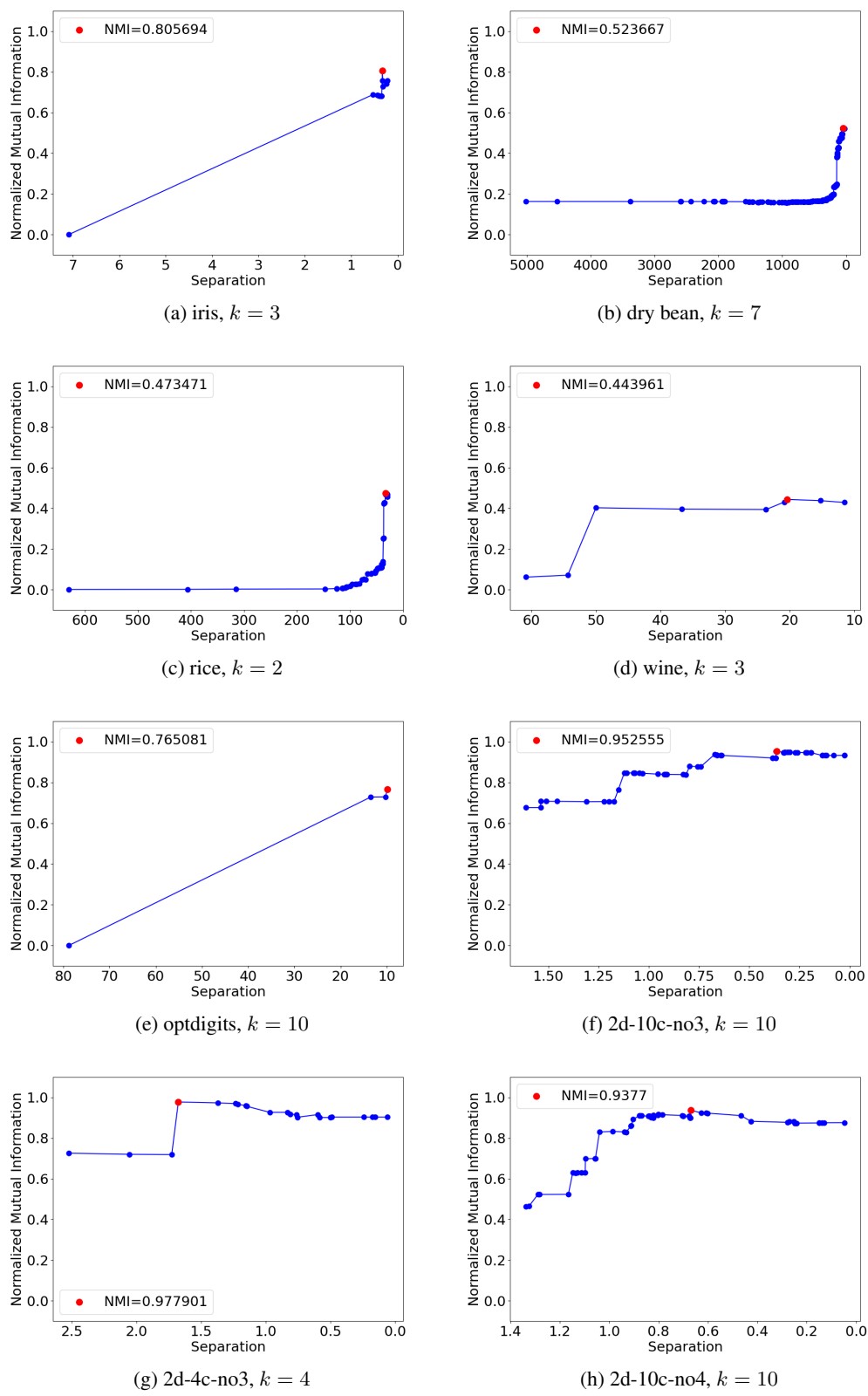

Figure 13: Normalized Mutual Information for approximate Pareto curves. The highlighted points show the solutions on the curve with highest values.

| Measures | Variance | | Measures | Variance |
|---|---|---|---|---|
| iris k-means++ | | | rice k-means++ | |
| RI | 1.6864e-04 | | RI | 1.4400e-08 |
| F1 | 5.0690e-04 | | F1 | 6.6097e-09 |
| F2 | 1.7680e-05 | | F2 | 6.6097e-09 |
| F0.5 | 1.1936e-04 | | F0.5 | 6.0840e-09 |
| NMI | 3.4781e-04 | | NMI | 4.5369e-08 |
| iris k-means++ k=7 | | | rice k-means++ k=2 | |
| RI | 1.2151e-03 | | RI | 1.4400e-08 |
| F1 | 1.9033e-03 | | F1 | 6.6097e-09 |
| F2 | 8.0220e-04 | | F2 | 6.6097e-09 |
| F0.5 | 2.5604e-03 | | F0.5 | 6.0840e-09 |
| NMI | 1.3560e-03 | | NMI | 4.5369e-08 |
| iris Pareto set | | | rice Pareto set | |
| RI | 8.9365e-04 | | RI | 0.0000e+00 |
| F1 | 8.0365e-04 | | F1 | 0.0000e+00 |
| F2 | 8.4472e-04 | | F2 | 0.0000e+00 |
| F0.5 | 5.6302e-04 | | F0.5 | 1.2326e-32 |
| NMI | 1.1630e-03 | | NMI | 1.1937e-06 |
| dry-bean k-means++ | | | wine k-means++ | |
| RI | 1.5876e-07 | | RI | 4.9972e-05 |
| F1 | 2.2125e-05 | | F1 | 6.3148e-07 |
| F2 | 5.2971e-05 | | F2 | 3.6481e-06 |
| F0.5 | 6.8142e-06 | | F0.5 | 1.6959e-04 |
| NMI | 7.2917e-06 | | NMI | 2.5335e-06 |
| dry-bean k-means++ k=2 | | | wine k-means++ k=3 | |
| RI | 1.4560e-05 | | RI | 1.5471e-04 |
| F1 | 2.5828e-05 | | F1 | 1.2263e-03 |
| F2 | 3.4703e-04 | | F2 | 1.3653e-04 |
| F0.5 | 2.0610e-04 | | F0.5 | 1.4320e-03 |
| NMI | 1.7967e-05 | | NMI | 1.9449e-05 |
| dry-bean Pareto set | | | wine Pareto set | |
| RI | 1.9592e-05 | | RI | 6.6312e-07 |
| F1 | 1.7173e-05 | | F1 | 2.5850e-05 |
| F2 | 1.0253e-04 | | F2 | 9.2865e-05 |
| F0.5 | 3.5594e-05 | | F0.5 | 8.4455e-05 |
| NMI | 5.7523e-07 | | NMI | 4.8492e-05 |
| optdigits k-means++ | | | 2d-10c-no3 k-means++ | |
| RI | 1.3865e-05 | | RI | 4.4768e-05 |
| F1 | 8.2189e-04 | | F1 | 8.5650e-04 |
| F2 | 6.0707e-04 | | F2 | 7.6900e-04 |
| F0.5 | 2.8317e-04 | | F0.5 | 3.4128e-04 |
| NMI | 9.0615e-05 | | NMI | 1.3592e-04 |
| optdigits k-means++ k=3 | | | 2d-10c-no3 k-means++ k=7 | |
| RI | 1.2933e-04 | | RI | 2.0670e-04 |
| F1 | 1.8262e-03 | | F1 | 1.7448e-03 |
| F2 | 1.3216e-03 | | F2 | 1.3333e-03 |
| F0.5 | 1.9015e-03 | | F0.5 | 2.0250e-03 |
| NMI | 5.6838e-04 | | NMI | 4.3568e-04 |
| optdigits Pareto set | | | 2d-10c-no3 Pareto set | |
| RI | 2.8145e-05 | | RI | 2.2066e-06 |
| F1 | 3.1205e-04 | | F1 | 4.8728e-05 |
| F2 | 2.5979e-04 | | F2 | 2.4905e-05 |
| F0.5 | 1.6697e-04 | | F0.5 | 6.8946e-05 |
| NMI | 1.1354e-04 | | NMI | 1.4273e-05 |
| 2d-4c-no3 k-means++ | | | 2d-10c-no4 k-means++ | |
| RI | 3.8151e-07 | | RI | 2.1759e-05 |
| F1 | 6.1481e-07 | | F1 | 3.4336e-04 |
| F2 | 6.2162e-07 | | F2 | 1.4351e-04 |
| F0.5 | 5.5008e-07 | | F0.5 | 1.8235e-04 |
| NMI | 5.9403e-06 | | NMI | 8.8342e-05 |
| 2d-4c-no3 k-means++ k=4 | | | 2d-10c-no4 k-means++ k=4 | |
| RI | 3.8151e-07 | | RI | 5.9045e-05 |
| F1 | 6.1481e-07 | | F1 | 8.7161e-04 |
| F2 | 6.2162e-07 | | F2 | 5.6323e-04 |
| F0.5 | 5.5008e-07 | | F0.5 | 1.1198e-03 |
| NMI | 5.9403e-06 | | NMI | 2.1481e-04 |
| 2d-4c-no3 Pareto set | | | 2d-10c-no4 Pareto set | |
| RI | 4.9304e-32 | | RI | 4.4955e-05 |
| F1 | 1.2326e-32 | | F1 | 5.5125e-04 |
| F2 | 4.9304e-32 | | F2 | 1.3260e-04 |
| F0.5 | 1.2326e-32 | | F0.5 | 4.5889e-04 |
| NMI | 1.2326e-32 | | NMI | 7.0230e-05 |

Figure 14: Variance of values for 20 runs of $k$-means++ and our algorithm.

## C.2 Visualizing median incomes in Germany

In this section we use our algorithm for combining two different $k$-center objectives to cluster the 400 German districts by their median income. Our goal is to compute clusters of districts of similar median income that are also geographically compact. Hence, we use as second objective function the geographic distance between the districts. This way it can be ensured that only districts close to each other get clustered together. This results in clusters that actually correspond to meaningful regions which could help identifying and analyzing regional inequality within a country.

To calculate the difference between the monthly median incomes of two states we used the information of the German "Bundesagentur für Arbeit" [29] from December 31, 2022. To calculate the distance between the districts we obtained a geojson file containing their border [21], calculated the centers of mass where we interpreted the shapes as regular polygons (as the curvature of the earth should not have too much influence on that scale), and then calculated the distance between those centers while accounting for the curvature of the earth.

Figure 15 provides an overview on the income structure in Germany. We will mainly discuss the details of the results for the case that $k = 16$ (the number of states in Germany). The respective Pareto curve is shown in Figure 16. However, the behavior for other choices of $k$ is similar, as can be seen in Figure 17.

In total the Pareto curve contains 38 different solutions, where the first optimizes the geographic radius of the clusters as much as possible and the 38-th minimizes the income difference. We will have a closer look at the 10-th and 15-th solution. We consider solutions that appear relatively early in the Pareto front because the geographic radius increases to 300 or even 350 kilometers relatively early on the Pareto front. Clusterings with such a large radius are not interesting because the air-line distance between Kiel (which lies in the north of Germany) and Munich (which lies in the south) is less than 700 kilometers, meaning that the solutions appearing later in the Pareto front cluster together districts that are far apart. In Subfigure 19b the 10-th Pareto solution is depicted. On the left the clusters are colored by the average income of the respective districts and on the right in colors that are easier to distinguish but have no intrinsic meaning. One can see that multiple clusters are next to each other that barely differ in their income which is not surprising because if the algorithm would cluster them together the geographic distance between the different parts of the cluster would get too large. At the same time we have some regions where two or more clusters basically cover the same geographic region and split up the districts depending on the respective median income in these districts. Despite all of this, the left image gives us a very good idea how the incomes differ in different regions of Germany. This is also interesting because the geographic radius of this clustering is roughly 196 kilometers and the maximum income difference between a district and its respective center is 463€ per month. Even though the latter value allows the algorithm to cluster districts together that have a large income difference (as a comparison the maximum income difference is roughly 1900€) the clusters appear to be reasonable and meaningful.

To get a better idea about the quality of the chosen Pareto solution in Figure 19a, the clustering produced by a $k$-center 2-approximation that only considers the income difference as a metric as well as a similar clustering for the geographic distance metric are depicted in Figure 19c. As expected for the income-based clustering the clusters are split up over the entirety of Germany. If one depicts them in the colors corresponding to the median income of the districts in the cluster, one basically ends up with a modified version of Figure 15 where the income levels have been discretized to 16 different levels. On the other hand, in the geography-based clustering districts with very different median income have been clustered together. As a result the median incomes are very similar for many pairs of clusters (it is almost the average over the entirety of Germany) and the information where a wealthy region starts and a not so wealthy region ends gets obscured.

In Figure 20 the 15-th Pareto solution is depicted. The geographic radius of this solution is 245 km and the maximum income difference between a district and its center is 321€. Indeed one can observe that in the map which colors the clusters according to their average median income a lot more details are visible. However at the same time, the clusters are a lot less compact often being split up into multiple regions or having holes in which the contained districts are part of another cluster. The resulting images also give an idea why we did not look at Pareto solutions prioritizing the income metric even more since already in this solution the clusters do not really resemble something that a person would intuitively describe as a region.

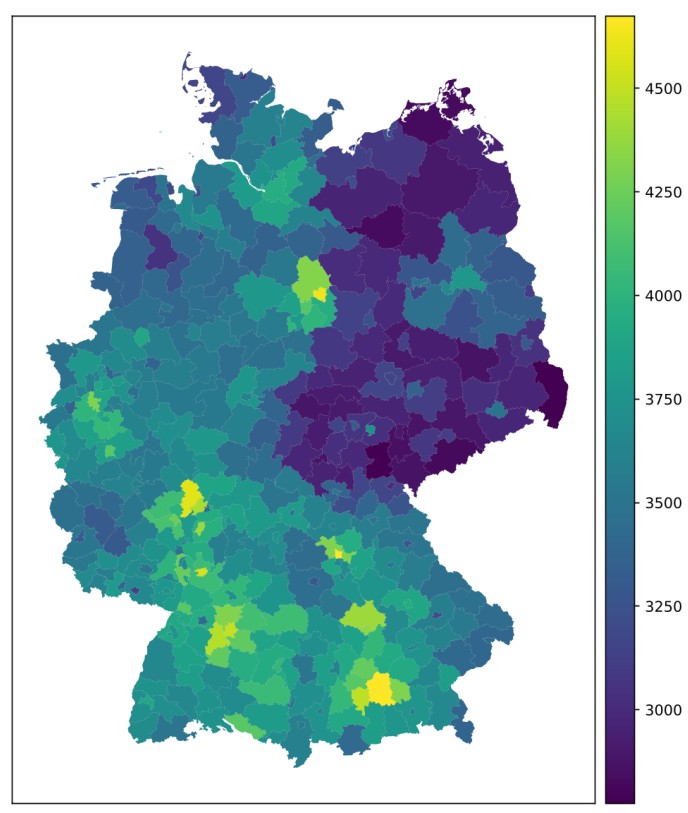

Figure 15: Visualization of the median monthly incomes in Germany (in Euro).

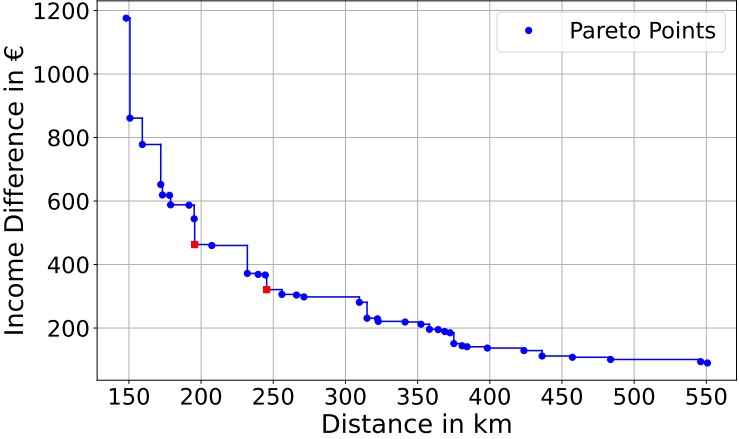

Figure 16: The resulting Pareto curve for $k = 16$. The 10-th and 15-th solution are marked with red squares.

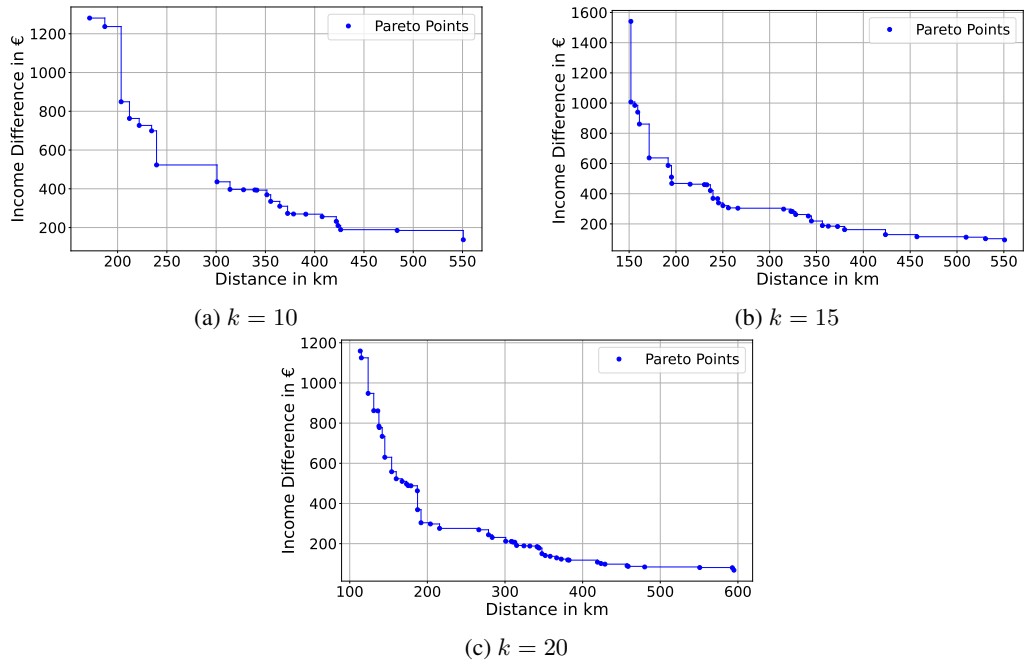

(a) $k = 10$

(b) $k = 15$

(c) $k = 20$

Figure 17: The resulting Pareto curves for different $k$

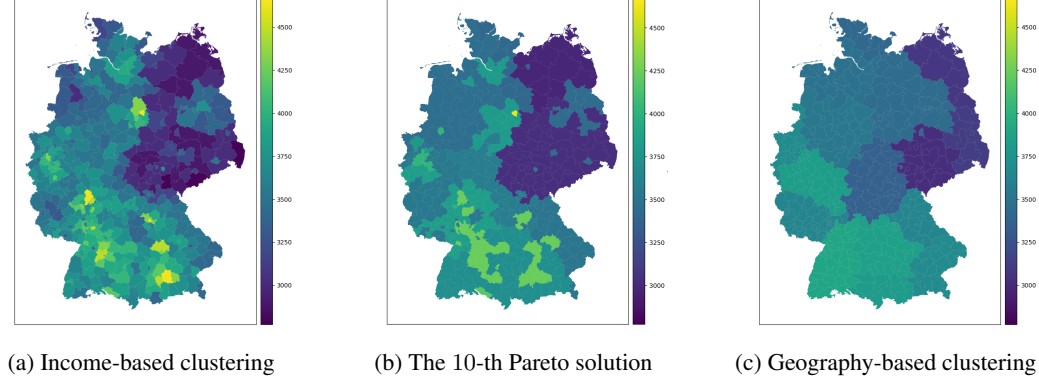

(a) Income-based clustering     (b) The 10-th Pareto solution     (c) Geography-based clustering

Figure 18: Comparison between the 10-th Pareto solution with the purely geographic and the purely income based clustering for $k = 16$.

**Resources and running time.** The experiments were performed on a machine with a 1.8 GHz Intel Core i7-8565U processor and 16 GB RAM. The source codes are in Java and were compiled using oracle OpenJDK version 21.02. The computations of the Pareto sets took at most a couple of seconds.

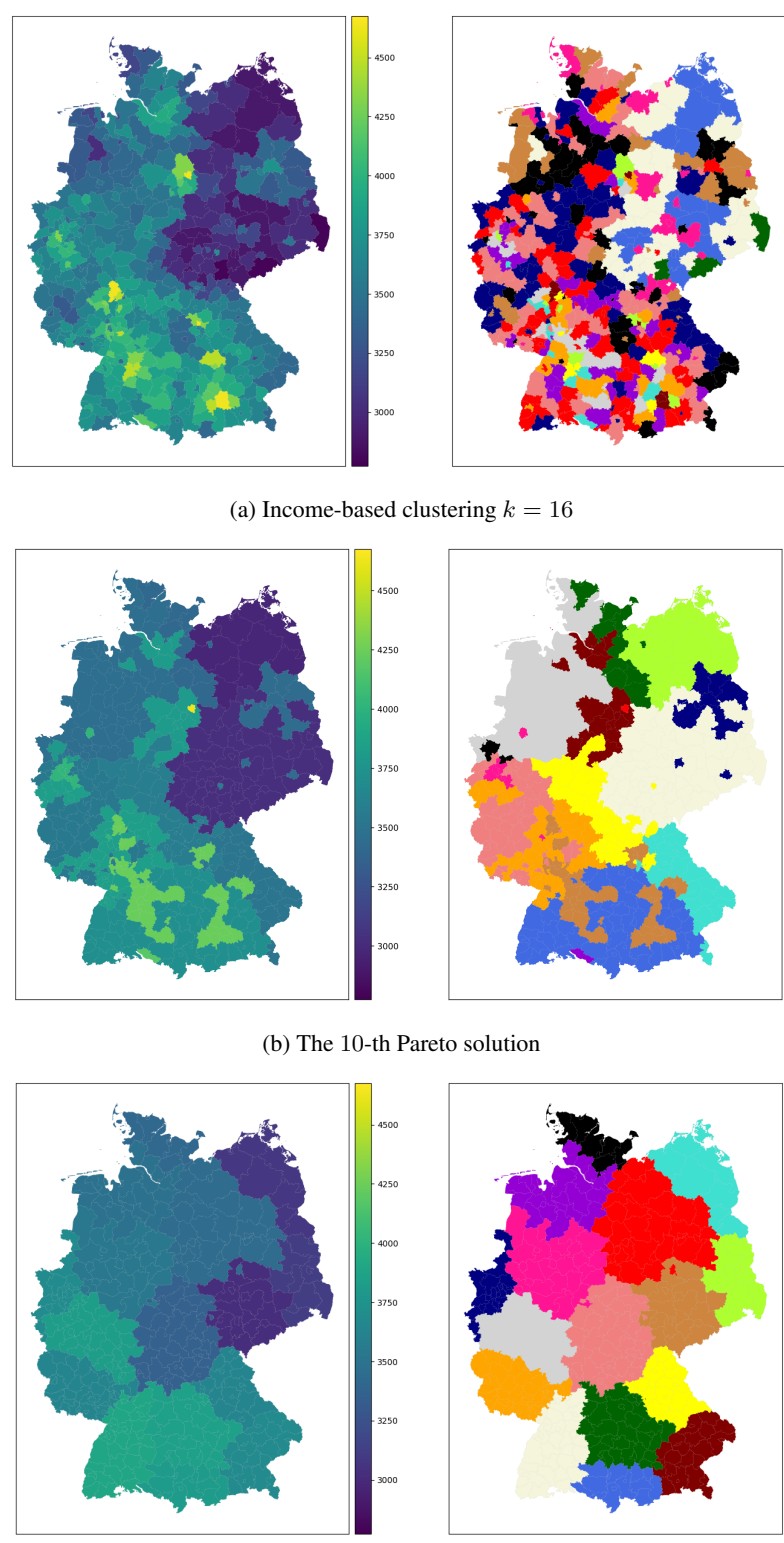

(a) Income-based clustering $k = 16$

(b) The 10-th Pareto solution

(c) Geography-based clustering

Figure 19: Comparison between the 10-th Pareto solution with the purely geographic and the purely income-based clustering for $k = 16$. The clusters are once colored according to the average income over all districts in that cluster (left) and once with well distinguishable colors (right).

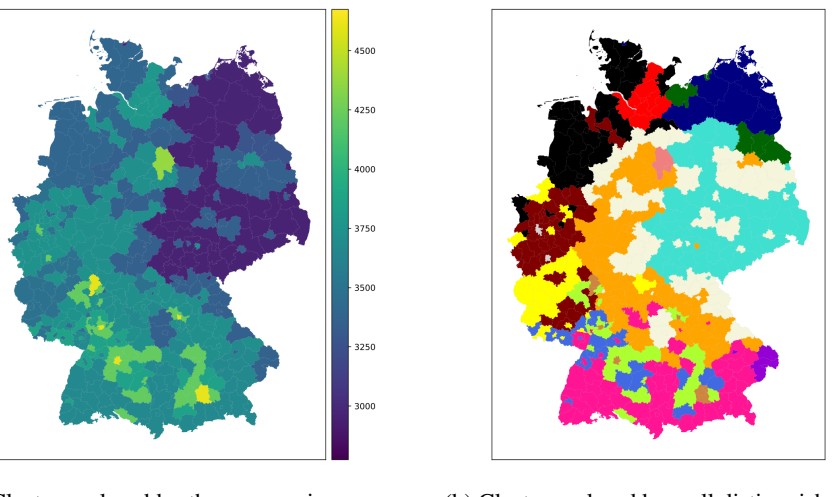

(a) Clusters colored by the average income of the districts

(b) Clusters colored by well distinguishable colors

Figure 20: The $15$-th Pareto solution for $k = 16$.

### C.3 Clustering of sea level data

Another example for an application of combining two k-center objectives with different metrics is to cluster the tide gauge stations of the Permanent Service for mean Sea Level (PSMSL) [50, 39] dataset. This dataset consists of 1581 tide gauge stations all over the world, most of them located on coastlines. Figure 21 shows their positions. Each of those stations measures the sea level at its respective location on a monthly basis, resulting in 1581 time series, some of which date back into the 19-th century.

Our goal is to cluster stations together that show similar behavior of the sea level, but at the same time we also want clusters to be geographically close to each other. In this way, we hope to be able to identify regions where the sea level behaves similarly and find a representative set of stations, that uses fewer stations, but does not loose a lot of information. It might also be helpful in order to achieve a more evenly spread out set of stations since the northern hemisphere is over-represented in terms of concentration of the stations in comparison to the southern hemisphere.

We clustered the stations according to their geographic distance and the distance of the corresponding time series. For the geographic distances, we used the Euclidean distance between the stations, while accounting for the curvature of the earth. For the distance between the time series we used the mean

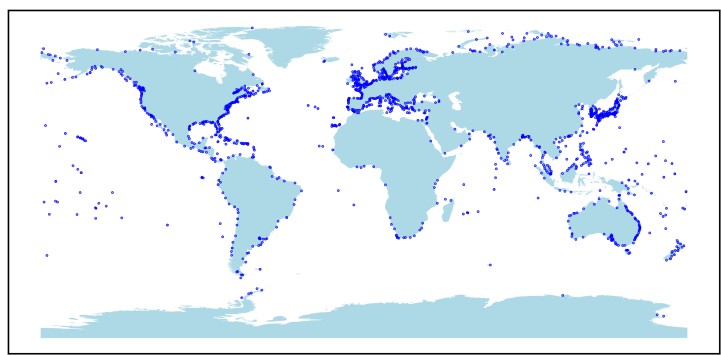

Figure 21: Positions of the PSMSL tide gauge stations.

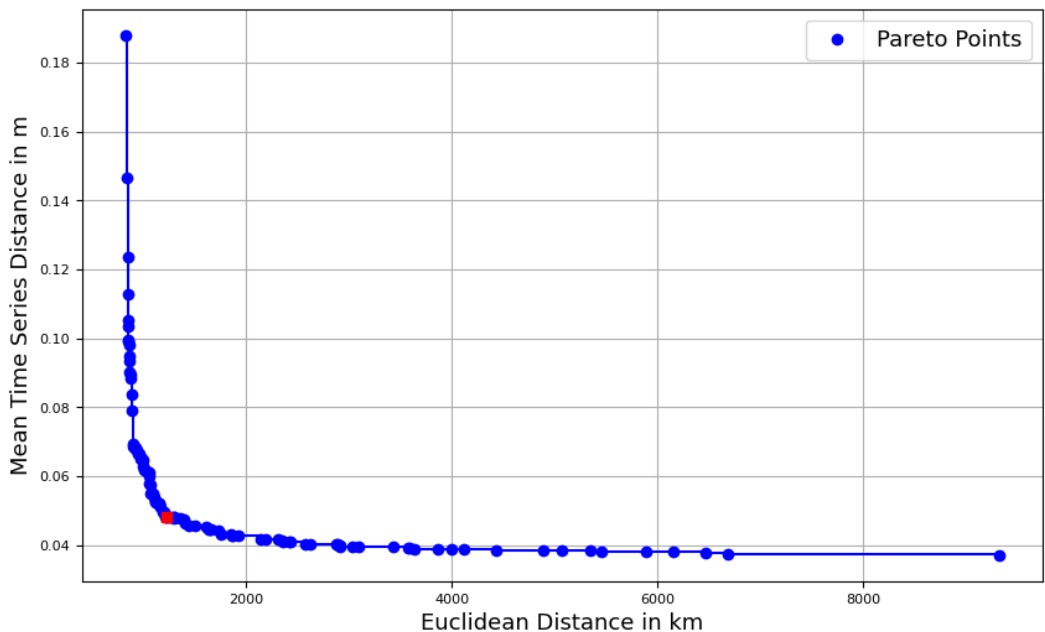

Figure 22: Approximate Pareto curve for the sea level data for k=150. The 54-th approximate Pareto solution is highlighted in red.

over the pairwise difference between two time series, namely

$$d_{mean} = \frac{1}{l} \sum_i t_i^{(1)} - t_i^{(2)},$$

where $t^{(1)}, t^{(2)}$ are two time series with $|t^{(1)}| = |t^{(2)}| = l$. For both distance measures, we used the $k$-center objective.

Unfortunately, the time series do not all cover the same time span and many of them have missing periods of data points due to a lot of different reasons (e.g., damage of the respective station, stations were not all built at the same time). Since our distance measure is not metric under these circumstances, which is a crucial part of our analysis, we conducted our experiments on a synthetic dataset using the ORAS5 [18] global ocean reanalysis, which simulates the sea level beginning in 1958 until today on grid coordinates all around the world. We created time series similar to the PSMSL dataset by mapping the coordinates of every station to the geographically closest grid point of the simulation that lies in the sea. With this method, we were able to produce 1581 complete time series, dating from January 1958 until December 2022. Since we are only interested in the change in sea level height, we normalized all time series by subtracting their mean value.

We considered the clustering for $k = 150$. In total, the approximate Pareto set contains 106 solutions. Figure 22 shows the approximate Pareto curve. The first Pareto solution optimizes for the time series metric, while the 106-th solution minimizes for the geographic distance. The 54-th Pareto solution is highlighted in red, since we are using it in later figures as example for a Pareto solution that balances both objectives quite well. One can see that by only moving by one or two solutions in the Pareto curve, there can be major improvements on one metric while the loss on the other metric is relatively small in comparison, especially on its border. Therefore the Pareto solutions might even be interesting in the case, where one is mainly interested in one of the metrics.

In the following, we will compare the 54-th Pareto solution to the clusterings, that either only consider the time series based clustering, or only consider the geography based clustering. The clusterings we compare to have been computed without any knowledge of the second objective. They do not necessarily coincide with the two extreme solutions in the approximate Pareto set. We compare those two solutions to the 54-th solution of our Pareto curve, which is located on the middle of the Pareto curve and therefore should be a good trade-off between the different objectives.

Table 7: Comparison between the time series based clustering, geography based clustering and the 54-th Pareto clustering with regard to the biggest cluster radius for both metrics, the mean cluster radius over all clusters for both metrics, and the max. cluster size (cs). The values for the time series based metrics (ts) are in m and rounded to three decimals, the geographic distances (geo) are in km and rounded to two decimals.

|  | max geo | mean geo | max ts | mean ts | max cs |
|---|---|---|---|---|---|
| **Geography based clustering** | 1,566.48 | 619.40 | 0.209 | 0.041 | 95 |
| **54-th Pareto Solution** | 2,411.21 | 799.05 | 0.080 | 0.031 | 128 |
| **Time Series based clustering** | 18,172.36 | 1,570.64 | 0.067 | 0.028 | 157 |

One can see a visualization of the three clusterings in Figure 23. For more details, Figure 24 shows the region of Europe (since there is a high concentration of stations there). The Pareto solution and the geographic solution both show quite compact clusters, while the clusters of the time series based clustering span bigger distances and are also overlapping each other. One can also see, that for the time series based clustering, the regions that have a very high concentration of stations usually also contain more cluster centers than the regions with a low concentration of cluster centers, while in the Pareto solution and especially in the geography based clustering, the centers are more evenly distributed over the globe.

Figure 26 shows the clusters with the largest geographic distance for the geography based solution, the time series based solution, and the 54-th approximate Pareto solution. It is easy to see that both the geographic and the Pareto clustering are quite compact, and all points of the cluster are located in the same geographic region, while the points of the time series based clustering span over three different oceans.

Analogously, Figure 25 shows the worst cluster with respect to the time series for the same three clusterings. One can see that in the Pareto clustering and the time series based clustering, the red center curves seem to be relatively good representatives of the other curves, while in the geography based clustering there are curves that obviously do not match the center curve at all.

Table 7 confirms these findings, and also shows that these effects are not only restricted to the worst cluster in a solution, but also visible in the mean of the radii over all clusters. Like before, for both objectives the mean radius of the Pareto solution is much closer to the mean radius of the clustering that is optimized for the respective objective than to the other one.

**Resources and running time.** The experiments were performed on a machine with an 2,4 GHz Apple M2 processor and 24 GB RAM. The source codes are in Java and were compiled using oracle OpenJDK version 21.02. The computations of the Pareto sets took less than a minute.

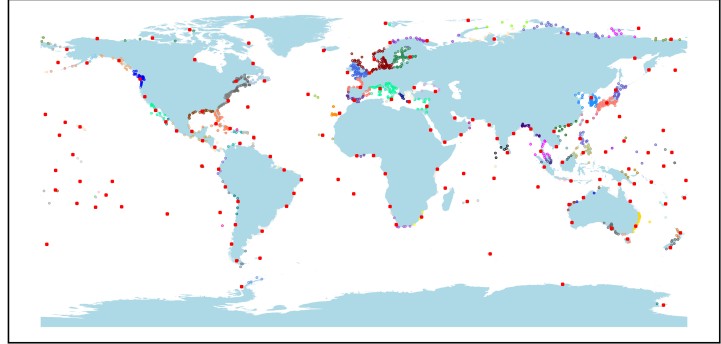

(a) Geography based clustering.

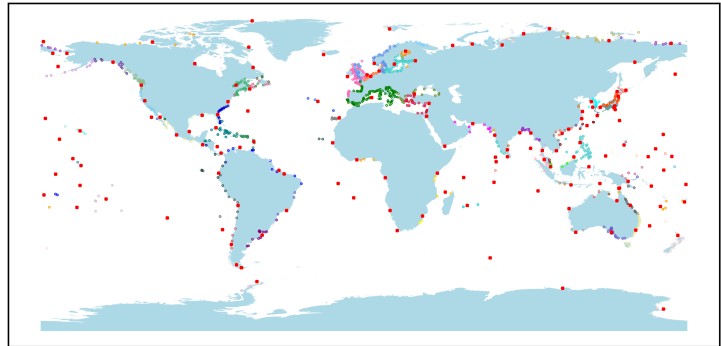

(b) The 54-th Pareto solution.

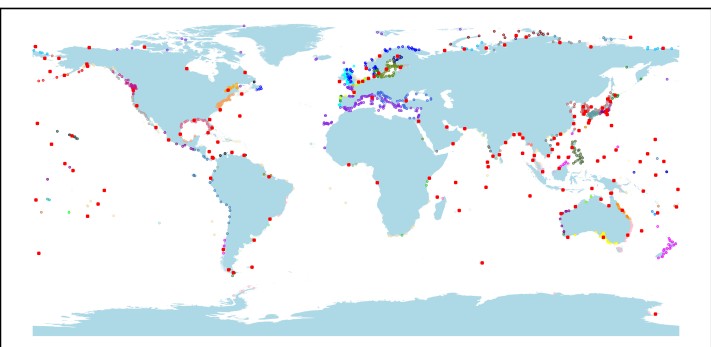

(c) Time series based clustering.

Figure 23: The geographically biggest clusters of the 54-th Pareto solution, the clustering that only considers the geographic distance, and the clustering that only considers the time series for k = 150. Different clusters are depicted in different colors. The centers of the clusters are marked by a red square.

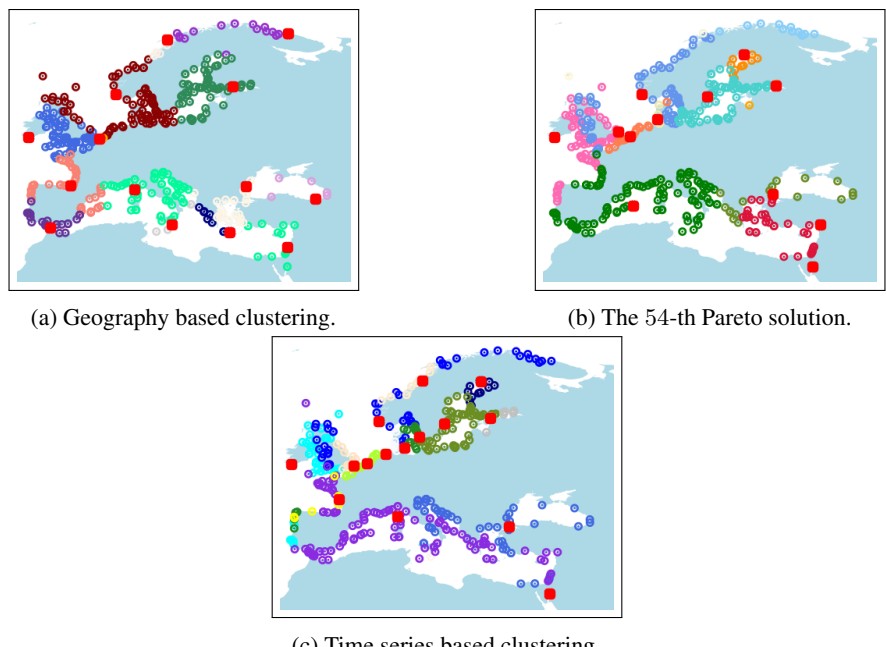

(a) Geography based clustering.

(b) The 54-th Pareto solution.

(c) Time series based clustering.

Figure 24: The geography based clustering, the 54-th Pareto solution, and the time series based clustering in the area of Europe. Centers are drawn in red.

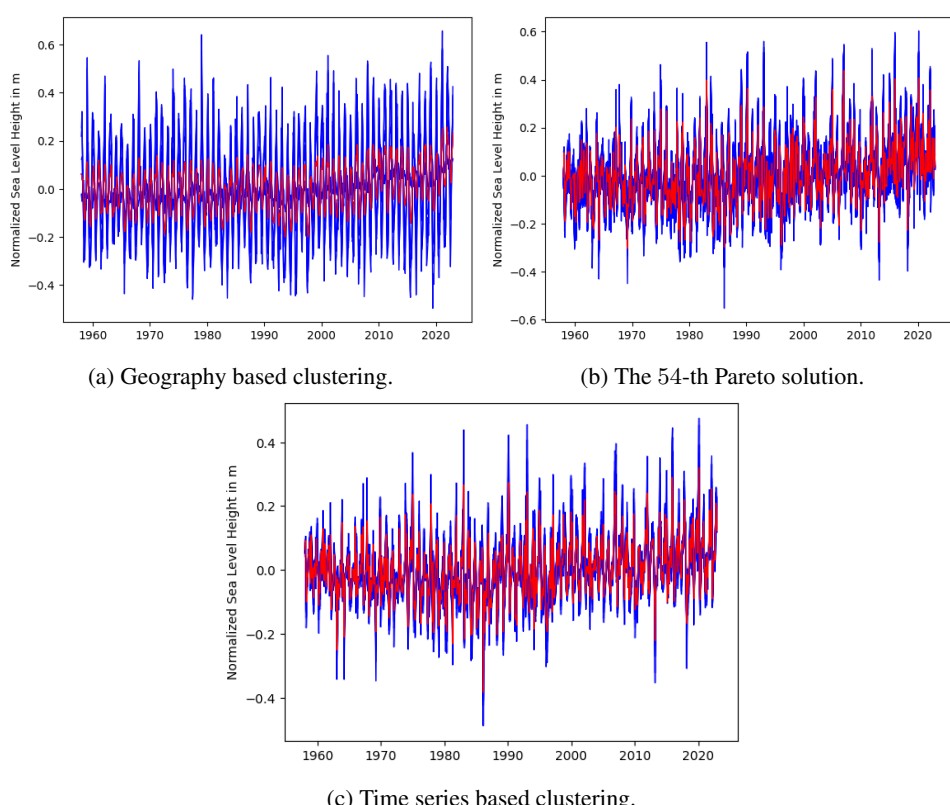

(a) Geography based clustering.

(b) The 54-th Pareto solution.

(c) Time series based clustering.

Figure 25: Comparison between the worst clusters (for the time series based distance function) of the 54-th Pareto solution, the clustering that only considers the geographic distance, and the clustering that only considers the time series for k = 150. The centers of the clusters are marked red.

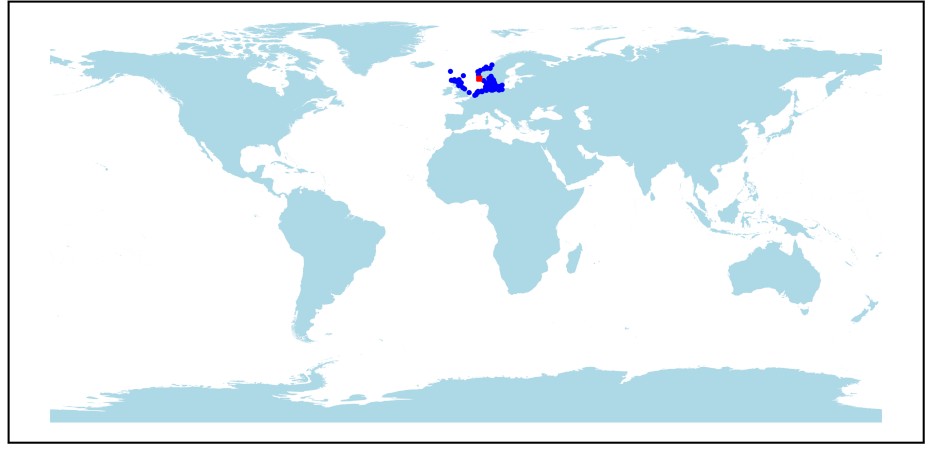

(a) Geography based clustering.

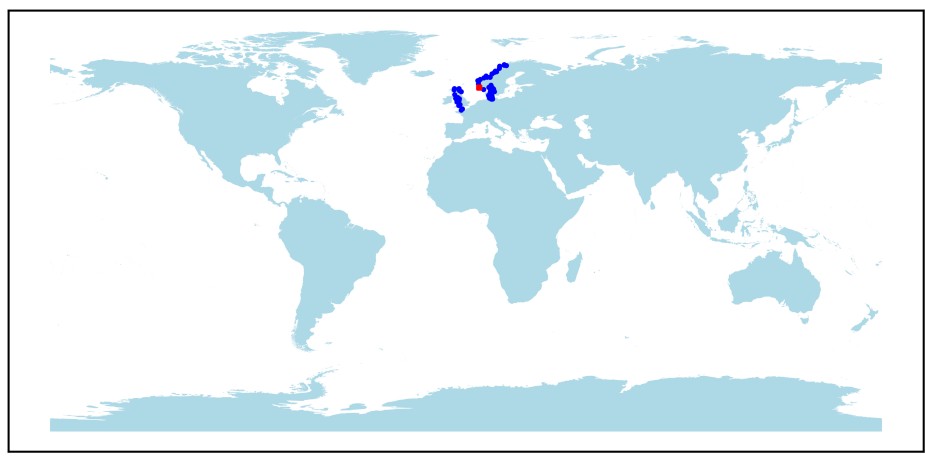

(b) The 54-th Pareto solution.

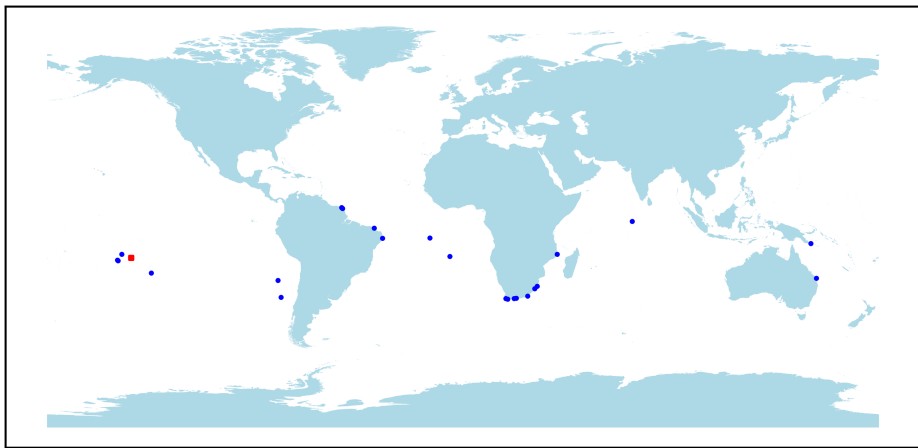

(c) Time series based clustering.

Figure 26: Comparison between the geographically worst cluster of the 54-th Pareto solution with the clustering that only considers the geographic distance and the clustering that only considers the time series for k = 150. The centers of the clusters are marked by a red square.

