# OpenReview forum: "Approximately Pareto-optimal Solutions for Bi-Objective k-Clustering"
_NeurIPS.cc/2024/Conference — NeurIPS 2024 poster_

### Official Review · Reviewer_2DdM · 2024-07-12

**Soundness:** 4
**Presentation:** 3
**Contribution:** 3
**Rating:** 7
**Confidence:** 4

**Summary:**

This work develops efficient algorithms to approximate the Pareto-optimal set for different bi-objective clustering problems with solid theoretical guarantees. The problem can have different clustering objectives (e.g., k-separation, k-center, k-diameter, k-median, k-means, k-MSR) and/or different metrics (e.g., different distance measures).

Two different types of algorithms are considered in this work to (1) approximate the whole Pareto set for problems that combine k-separation with various k-clustering minimization objectives or combine k-center/k-diameter with a k-clustering minimization problem, and (2) approximate the convex Pareto set (e.g., the convex hull of the ground truth Pareto set) for problems that combine k-median and k-means, where the Pareto set can be exponentially large.

Thorough and comprehensive theoretical analyses are provided to support and demonstrate the efficiency of the proposed algorithms. Experimental results show that the proposed algorithms can achieve promising performance on different bi-objective clustering problems.

**Strengths:**

+ This paper is well written and easy to follow.

+ The multi-objective clustering problem is valuable and important for many real-world applications. This work would be impactful and inspire many follow-up work on this research direction.

+ The proposed algorithms and their corresponding theoretical analysis are comprehensive and cover a wide range of different problem settings in a systematic way, which is a solid contribution.

+ The experiments are well-designed to nicely show the benefits of multi-objective clustering.

**Weaknesses:**

I enjoy reading this paper and do not see any obvious weakness of this work. Below are some minor concerns and questions.

**1. Centers Chosen from the Point Set**

One limitation of the proposed algorithms is that the clustering centers can only be chosen from the point set. I can understand the necessity of this requirement when we hope to approximate all Pareto solutions (up to $O(n^2)$ with this requirement) in subsection 2.1 and subsection 2.2.

However, it is not clear whether this restriction is still necessary when we want to approximate the convex Pareto set in section 2.3. What is the challenge for this case if we are allowed to choose the center from an ambient metric space?

**2. Extension of the LP Rounding Algorithm**

As mentioned in the related work section, Alamdari and Shmoys (2017) leverage the LP rounding algorithm by Charikar et al. to handle the bi-objective clustering problem with k-center and k-median, while this work uses the primal-dual algorithm by Jain and Vazirani (2001). I am curious whether the LP rounding algorithm can be extended to tackle (some of) the other settings considered in this work.

In addition to the incomparable approximation factor, what are the pros and cons between the LP rounding algorithm and the primal-dual algorithm used in this work for multi-objective clustering?

**3. Generalization to more than Two Objectives**

This work focuses on bi-objective clustering, but many real-world applications might involve more than two objectives. What makes it hard to generalize the current methods to handle more objectives for approximating the Pareto set or convex Pareto set?

**4. Minimum Cardinality Set**

It seems that the proposed methods in this work do not consider minimize the cardinality of the approximate solution set. Can these methods be further extended to find the smallest (minimum cardinality) approximate Pareto set as discussed in Diakonikolas (2011)?

**5. Finding the Most Suitable Solution**

The number of solutions found by the proposed methods for a given method can still be large. How can we efficiently find the best solution for each application (e.g., the best clustering in Figure 4)? Since the ground truth clustering is unknown, we cannot calculate the quality metric (e.g., Normalized Mutual Information and Rand Index) for each solution. Will the users have to check all the solutions themselves and then choose the most suitable one?

In addition, how can the clustering number $k$ be properly chosen for real-world multi-objective clustering problems?

**6. Order of the Problem Settings**

There is a mismatch between the order of problem settings in the main paper and the appendix. In the main paper, the order or problems is 1) k-separation + k-clustering minimization, 2) k-center/k-diameter + k-clustering minimization, and then 3) combination of k-median and k-means. However, in the appendix, the order is A) combination of k-clustering minimization , and then B) combinations involving k-separation. What is the reason for this choice?

**7. Publication Venue**

*(This part will not affect my decision.)*

This work conducts a comprehensive study on multi-objective clustering, which is not possible to be compressed into a 9 (or 10 in camera-ready) page conference paper. Many important materials (e.g., algorithms, all theorems, and theoretical analyses), which are actually the crucial contributions of this work,  have to be put in the supplemental materials. There is also no room left for a conclusion section.

I think a journal like JMLR should be a more suitable choice to publish this work.

**Questions:**

See weaknesses.

**Limitations:**

This work has discussed its limitations in the introduction (page 2, objectives subsection), but I think an explicit limitation subsection could be much more helpful.

---

> ### Author Rebuttal · Authors · 2024-08-04
>
> *1. Centers Chosen from the Point Set*
>
> We decided to restrict to this case in the theory part. In 2.1 we could switch to choosing centers from a larger space, e.g., the infinite metric space $R^d$ because this algorithm is still a 2-approximation even if this case. It would make even more sense to switch for k-means (in the practical algorithm in C.1 we actually compute centers from $R^d$.). But e.g. in 2.2, if we would allow $R^d$ for one objective we can not use standard R^d techniques like epsilon-nets unless both objectives use $R^d$. On the other hand, all our results imply results for the infinite case with an additional factor of 2, so we opted against including more case distinctions. We agree that this is not sufficiently discussed in the introduction yet.
>
> *2. Extension of the LP Rounding Algorithm*
>
> We did not try to adapt the algorithm by Charikar et al. But in the single objective case, it is actually the JV algorithm that is often adapted, while the rounding algorithm (which is actually older than the JV algorithm) involves many intricate steps and we are not aware of many adaptations. For example, the (3+eps)-k-msr-approximation is based on the JV algorithm. We are not even sure if the Alamdari/Shmoys algorithm can be adapted to the case of different metrics which they did not consider (it works for two objectives on the same metric). Which other setting would you be interested to know about?
>
> Pro/Cons: The incomparable approximation factor may be even in favor of Alamdari/Smoys. But we expect that our algorithm is better suited for practical purposes. We had positive experiences with implementations of other JV adaptations in the past. We did not write anything about this because we did not practically test it here. Finally, the algorithm by Charikar et al. requires actually solving an LP, while the primal dual algorithm is combinatorial (only the proof needs the LP), which could also be a pro of our algorithm.
>
> *3. Generalization to more than Two Objectives*
>
> For the sake of simplicity we focus on the combination of two objectives. We believe that the cases where we combine at most one k-median or k-means objective with other (possibly multiple) objectives as k-center/k-diameter/k-separation can be carried out with techniques similar to the ones presented in this work. The combination of two k-median/k-means objectives differs substantially from the other combinations since we are only able to compute an approximation to the convex pareto set. As long as one only uses sum based objectives (k-median or k-means), this approach should also extend to more than two objectives.
> One may adapt the primal dual algorithm by Jain and Vazirani where we change the objective to a convex combination of multiple k-median/k-means objectives and combine it with (multiple) k-center objectives to compute an approximation to the convex pareto set.
>
> From a practical point of view the main downside of computing the pareto set for multiple objectives lies in the large size of the pareto set and therefore also in the high running time. This only seems to make sense when combined with autmatic methods to identify an interesting subset of the approximate Pareto set that we want to compute.
>
> *4. Minimum Cardinality Set*
>
> This is an interesting question that we did not consider yet. Diakonikolas (2011) and Vassilvitskii, Yannakakis (2005) both show that the smallest approximate Pareto set can be approximated for large groups of problems, namely if you have a polynomial time gap routine (VY2005) or if you can solve a restricted version of the problem in polynomial time (D2011). Unfortunately, those frameworks do not directly carry over to our setting, since we generally only have constant factor approximations for the gap problem. It seems possibly that these frameworks could be adapted to work in this context as well.
> It would be also interesting to check if our algorithms can be adapted directly to compute smaller appr. Pareto sets for example for the combination of two k-center objectives if one really has to consider all combinations or can cleverly skip certain values that are already 2-approximated.
>
> *5. Finding the Most Suitable Solution*
>
> Indeed we originally thought of this as an ensemble which is then checked manually. However, it clearly makes sense to combine the algorithms with known techniques to choose k (elbow method, more complicated phase transistioning detections), or to extend these methods to compute an "interesting" subset of the Pareto curve, or to apply other techniques to reduce the size of the approximate Pareto set (also related to question 4). In case of applications C.1, C.2 and C.3 the number of solutions was managable manually.
>
> In C.1 we observed a general trend that small separation values improve the solution while enforcing a large separation did not help. It seems that the reason is that k-means++ sometimes splits large clusters in the middle and this is not possible when close points are forced to be in the same cluster. So for practical applications as a rule of thumb one could compute the first few clusterings (which have small separation and small k-means cost) on the Pareto curve and select one of them manually. Only computing this part of the curve of course also reduces the running time.
>
> *6. Order of the Problem Settings*
>
> This is just an oversight and has no hidden meaning. Thank you for pointing it out!
>
> *7. Publication Venue*
>
> Thank you very much for pointing out the comprehensive nature of our paper. It is true that it does not really fit into the page limit and we struggled to present all material to the extend that we would have liked to. We chose to sent the paper to NeurIPS  since we introduce a new angle to the study of approximation algorithms for clustering. We believe that it would be an interesting contribution to NeurIPS since there are many possible follow-up questions to obtain faster and better approximations of the Pareto curves.

---

> > ### Comment · Reviewer_2DdM · 2024-08-09
> >
> > Thank you for the thorough response. All my concerns have been properly addressed, so I keep a positive score (7) at this stage.

---

### Official Review · Reviewer_Xxjr · 2024-07-13

**Soundness:** 2
**Presentation:** 2
**Contribution:** 2
**Rating:** 4
**Confidence:** 3

**Summary:**

This paper presents a novel framework for clustering pareto optimization. This paper studies the approximately pareto-optimal solutions for bi-objective k-clustering problems. The authors focus on the computationally very efficient single linkage / must-link constraints and the computation of pareto-optimal solutions. They first establish that it is not possible to simultaneously approximate k-separation and any of the minimization problems to constant factors. By iterating through all pairs of possible separation and possible radius/diameter, they obtain the approximate Pareto set. The authors also give the results combining k-center or k-diameter with a k-clustering minimization problem, and give the results combinations of k-median and k-means.

**Strengths:**

1. The authors give a novel approximately pareto-optimal approach for bi-objective k-clustering problems.The authors gives the results for combining k-separation with various objectives with (1,\alpha) approximate Pareto set with respect to sep with metric d_1 and f_2 with metric d_2.

2. The paper produces an informative ensemble. These algorithms achieve provable approximation guarantee. The theoretical evidence are solid and the experiments are well-established.

3. The authors verify that the approximate pareto front contains good clustering which cannot be found by considering a single objective.

**Weaknesses:**

1. This paper is not very motivated. The authors seem not explain the motivation for studying the pareto-optimal algorithm for the clustering problem. Moreover, the authors did not give the motivation why the combination of these clustering objectives is studied.

2. The authors seem not conduct experiments on large-scale datasets, such as 100 million points. Moreover, there are also faster version of k-means++ method, such as the rejection sampling method proposed in [1] and random projection based k-means++ method proposed in [2]. The author did not consider comparative experiments with these algorithms.

3. The algorithms have polynomial running time. As far as we know, the pareto optimization of other problems can be done in linear time.

4. Some sections, particularly those involving heavy mathematical notation and proofs, could be made clearer with additional explanations or visual aids. This would make the paper more clear.

[1] Cohen-Addad V, Lattanzi S, Norouzi-Fard A, et al. Fast and accurate k-means++ via rejection sampling[C]//Proceedings of the 34th International Conference on Neural Information Processing Systems. 2020: 16235-16245.

[2] Charikar M, Henzinger M, Hu L, et al. Simple, scalable and effective clustering via one-dimensional projections[C]//Proceedings of the 37th International Conference on Neural Information Processing Systems. 2023: 64618-64649.

**Questions:**

Q1: Why do the authors give this definition of pareto-optimal solutions for bi-objective clustering? The authors should explain the motivation of proposing this definition.

Q2: The authors focus on the single linkage/ must-link constraints. The authors should explain the significance of these constraints and give the comparison with a single objective.

Q3: Can the authors provide more detailed real-world applications where these bi-objective clustering algorithms would be particularly beneficial?

Q4: Do these experiments have scalability on large datasets? The authors should give some experiments on large-scale datasets. Moreover, the authors should add comparative experiments with other basic clustering algorithms.

**Limitations:**

This paper is mainly a theoretical result, and there is no negative societal impact.

---

> ### Author Rebuttal · Authors · 2024-08-04
>
> *W1: This paper is not very motivated. The authors seem not explain the motivation for studying the pareto-optimal algorithm for the clustering problem. Moreover, the authors did not give the motivation why the combination of these clustering objectives is studied.*
>
> We chose these objectives since they are the most known classical partitional clustering objectives and it seems useful to know about their joined optimization. Indeed, we originally were motivated by hoping for algorithms that could simultaneaously approximate several objectives until we proved that this was not possible.
> From a practical point, our original motivation stemed from applications C.2 and C.3. In collaborations we observed the need to find clusterings while optimizing over two very different metrics, e.g., one based on Euclidean distances and one based on time series similarity. We believe that such trade-offs are quite natural in practical applications.
>
> *W2: The authors seem not conduct experiments on large-scale datasets, such as 100 million points. Moreover, there are also faster version of k-means++ method, such as the rejection sampling method proposed in [1] and random projection based k-means++ method proposed in [2]. The author did not consider comparative experiments with these algorithms.*
>
> We did indeed not implement a super fast k-means++ variant. There are several options to do so, but they require additional work to satisfy theoretical guarantees. Notice that it is not sufficient to simply pick a fast k-means algorithm where it is not clear how to bound the second objective. In C.1 we go a middle ground by using k-means++ instead of the primal dual algorithm used in A.3 (this change decreases the approximation ratio but the algorithm still satisfies weaker guarantees). The k-means++ algorithm is state-of-the art for medium sized data. We have speculatd about the case of large data quite a bit and believe that super fast combinations are also possible, but the aim of this paper was an initial (and already pretty page-heavy) collection of first results on the topic.
>
> *W3: The algorithms have polynomial running time. As far as we know, the pareto optimization of other problems can be done in linear time.*
>
> For clustering problems as k-center and k-means, there are no algorithms which obtain constant factor approximations and have linear running time even in the case where we optimize over a single objective. Therefore we conjecture that it is not possible to compute a pareto set or an approximate pareto set in linear time since it requires solving multiple such problems. Our experiments can be seen as proof of concept that the approaches can be practical and work on medium size data sets.
>
> *W4 Some sections, particularly those involving heavy mathematical notation and proofs, could be made clearer with additional explanations or visual aids. This would make the paper more clear.*
>
> Answer: We apologize for the more technical nature of the appendix. We agree that the exposition could be better and plan to improve it for a long version of the paper. We hope that this comment did not apply to the main part of the paper. We welcome any additional pointers to sections that were unclear in order to improve them.
>
> *Q1: Why do the authors give this definition of pareto-optimal solutions for bi-objective clustering? The authors should explain the motivation of proposing this definition.*
>
> We apply the standard definitions for Pareto sets to the setting where the aim is to optimize two clustering objectives. The fact that we also include the possibility to optimize over two different metrics is due to the application in C.3 where this is necessary.
>
> *Q2: The authors focus on the single linkage/ must-link constraints. The authors should explain the significance of these constraints and give the comparison with a single objective.*
>
> SL may be over-highlighted in the introduction. We actually consider two angles: i) Combining single linkage with other objectives and ii) combining various well-known partitional clustering objectives with each other. For i), our motivation was of a more conceptual nature, to see if we can improve clustering for finding ground truths by including separation constraints into the consideration. ii) is directly motivated by applications where two different metrics are present.
>
> "Q3: Can the authors provide more detailed real-world applications where these bi-objective clustering algorithms would be particularly beneficial?"
>
> We point out that the data in C.2 and C.3 are from real-world applications. In particular C.3 is a question that is actually studied in geodesy and where two metrics are present over which one needs to optimize. Could you clarify in which aspect more details would be required? It may be that we described the applications a bit too short in the main body of the paper, and we apologize for that.
>
> "Q4: Do these experiments have scalability on large datasets? The authors should give some experiments on large-scale datasets. Moreover, the authors should add comparative experiments with other basic clustering algorithms."
>
> In C.1 we focus on the question whether the combination of two clustering objectives can improve the quality of a solution. Thus our experiments provide a comparison between the quality of a clustering obtained by k-means++,  a Single Linkage clustering, and the best clustering on the pareto curve. In C.2 and C.3 our aim was to highlight the usefulness of bi-objective clustering. We did not optimize the implementation for speed. Improving our implementation speed-wise or extending our guarantees to faster algorithms is an interesting open question. We also think that the computation of the full Pareto set can in most cases be avoided by identifying a smaller set of interesting solutions automatically, but we did not yet test this approach.

---

> > ### Comment · Reviewer_Xxjr · 2024-08-12
> >
> > Thank you for your response. However, after carefully considering your points, I prefer to maintain the original score.

---

> > > ### Author Response · Authors · 2024-08-12
> > >
> > > Thank you for your comment. Could you maybe give us a little more detailed response on why you prefer to maintain your negative score? We would be very interested to know what the remaining criticism is.

---

### Official Review · Reviewer_CrwW · 2024-07-15

**Soundness:** 3
**Presentation:** 3
**Contribution:** 3
**Rating:** 6
**Confidence:** 4

**Summary:**

The paper introduces novel algorithms to approximate the Pareto-optimal solutions for bi-objective k-clustering problems. The authors focus on combinations of clustering objectives such as k-center and k-means, or k-center with two different metrics. Usually, these objectives are conflicting and cannot be optimized simultaneously, making it necessary to find trade-offs. The algorithms provide provable approximation guarantees and are demonstrated through experiments to yield good clustering that single-objective approaches fail to capture.

**Strengths:**

1. The paper addresses the complex issue of multi-objective clustering. The motivation with k-diameter and k-separation problems accurately captures the difficulty of optimizing multi-objective optimization settings. The work provides a novel solution to approximate the Pareto front for bi-objective k-clustering problems.
2. The authors validate their approach through extensive experiments, showing that the approximate Pareto front includes superior clustering compared to those obtained by single-objective methods.
3. The paper has been written very well with detailed approaches.

**Weaknesses:**

1. It is not entirely clear the main novelty aspect of the work. In the setting of sec 2.1 where k-sep is combined with other objectives, the main takeaway seems to be that the authors were able to integrate existing state-of-the-art guarantees into their framework. Similarly for other sections for Pareto-optimal solutions were discussed, the authors leverage relies heavily on already existing approaches.
2. Most results are either incomparable with respect to the related work, because of bicriteria guarantees or translate to the existing guarantees. The results section should succinctly describe the main technical contributions (even if a few) for a better understanding of technical innovations.

**Questions:**

I would request the authors to clarify the main technical innovations in this work -- e.g., in 229: "The input to the algorithm now consists of clusters instead of single points". Does it require substantial change in techniques?

Pareto-optimality makes sense for multi-objective guarantees. But falling back on bi-criteria guarantees also might defeat the purpose. Is there a way to reconcile both and argue about them?

The approximate pareto-optimal sets-based approaches rely heavily on existing works. Can you please highlight the challenges or technical innovations in adopting it to the set of problems considered.

---

> ### Author Rebuttal · Authors · 2024-08-04
>
> *Weakness 1: It is not entirely clear the main novelty aspect of the work. In the setting of sec 2.1 where k-sep is combined with other objectives, the main takeaway seems to be that the authors were able to integrate existing state-of-the-art guarantees into their framework. Similarly for other sections for Pareto-optimal solutions were discussed, the authors leverage relies heavily on already existing approaches.*
>
> The main novelty is that algorithms with theoretical guarantees for the computation of Pareto sets for clustering have not been studied before (the only known related result was developed in a different framing). We give many results matching the best single-objective approximation ratios, and (nearly) none of this was known beforehand.
>
> Given that better bi-objective guarantees would also correspond to better single-objective approximations and most of these clustering problems are well studied, we find it unlikely that we could get any further improvements in these cases. We are aware that at some points our bi-objective algorithms are pretty straightforward adaptions of existing algorithms but we felt that it still makes sense to include these results for completeness. Rather than calling one single algorithm the main result we would consider the sum of all techniques presented in this paper to deal with all the different combinations of objective functions, as well as the given worst case instances and the experiments, our contribution to a natural research question that has only been studied scarcely before.
>
> Indeed Sec. 2.1 has less to offer in terms of technical contribution in comparison to the other sections. We put it in front because we like the conceptual contribution as it offers a new angle to the very basic question of how to obtain a meaningful clustering that reconciles two natural ideas (separation and compactness of clusters).
>
> *Weakness 2 / Question 1+3 (Technical contribution) I would request the authors to clarify the main technical innovations in this work -- e.g., in 229: "The input to the algorithm now consists of clusters instead of single points". Does it require substantial change in techniques?*, also *The approximate pareto-optimal sets-based approaches rely heavily on existing works. Can you please highlight the challenges or technical innovations in adopting it to the set of problems considered.*
>
> The refered line 229 did actually not require much change in techniques. Here is a list of novel contributions of technical nature:
> - the runtime reduction technique in A.1. The main idea to achieve the improvement was to not calculate an entirely new graph and independent set in every iteration of Hochbaum and Shmoys but rather to modify the already existing datastructures. Via the careful usage of a potential function we were able to prove the reduced runtime of $O(n^3)$ (pages 14-16)
> - the lower bound for the size of the Pareto set for k-center/k-center in A.2 (page 17)
> - using nesting for the k-sep/k-median case (Lemma 41/45)
> - lower bound for the size of the Pareto set for k-median and k-means (Theorem 21)
> - lower bound examples showing that SL cannot be reasonably approximated together with the other objectives (Ex. 1, Ex. 31, Obs. 36, Obs. 46, )
> - the algorithm in C.1 is also novel (but relatively straightforward)
>
> Aside from this we believe that collecting many techniques and finding those that can be applied in this setting is a contribution in its own right, like using nesting from hierarchical clustering (Lemma 41/45) or observing that Diakonikolas' framework works well with approximation algorithms.
>
> Finally, Sec. C.2 and C.3 offer novel modeling for two real world problems to highlight the usage of bi-objective optimization algorithms.
>
> *Q2-Pareto-optimality makes sense for multi-objective guarantees. But falling back on bi-criteria guarantees also might defeat the purpose. Is there a way to reconcile both and argue about them?*
>
> As we have shown in Ex. 1, Ex. 31, Obs. 36 and Obs. 46, at least for the combination of the k-separation objective with any other clustering objective we cannot hope for any bi-criteria approximation guarantee even if both objectives are defined on the same metric. Similarly it can be shown that it is not possible to optimize k-means or k-median and k-center or k-diameter at the same time since the k-center/diameter objective could force the algorithm to pick outliers which might decrease the quality of the means/median solution by an arbitrary amount (e.g., combining k-center/diameter with k-means/median, one would construct two point sets of high cardinality close to each other and one single point far away. For $k=2$, kmeans/median has to place one point in each large point set while k-center would spend one center for the outlier.) Thus, bi-criteria guarantees are in some sense the best we can hope for in this setting (at least for polynomial-time algoirthms).
>
> We are not sure if we understood the question correctly. If it is about the fact that we have two objectives rather then many, then we instead remark that we are definitely interested in the extension to multiple objectives, and in many cases, it should be possible. But the paper is already very long, so we left including additional technical work for multiple objectives to future work.

---

> ### Comment · Area_Chair_QTPu · 2024-08-12
> **Please respond to the rebuttal**
>
> Dear reviewer CrwW,
>
> Can you please read and comment on the rebuttal from the authors? Also, since we have a somewhat mixed rating, please take other reviews and responses into consideration.
>
> Regards,
> Area Chair

---

> > ### Comment · Reviewer_CrwW · 2024-08-12
> > **Rebuttal**
> >
> > Thanks for answering my questions -- they clarified some of the questions I had with regards to novelty.
> >
> > However, I do want to point out to the authors, that the presentation of the paper requires significant reworking to bring out the main technical contributions, for others in the research community to appreciate (or even understand) the technical contributions/results.
> >
> > After reading the rebuttal responses, I'm increasing my score.

---

### Author Rebuttal · Authors · 2024-08-04

We thank all reviewers for considering our work and providing detailed reviews. Replies to remarks and questions can be found in the individual rebuttals for each review.

---

### Decision · Program_Chairs · 2024-09-25

**Decision:**

Accept (poster)

**Comment:**

This paper gives various bounds for the Pareto approximation of k-clustering problems.

Summarizing the discussion, the notion of Pareto approximation was proposed before, and the main contribution of this paper greatly expands the range of problems that admit such approximation. The motivation in practical applications are justified by concrete examples. However, the ratios/results are not an improvement/incomparable to previous results, and the techniques are not particularly novel. The writing should be improved to better position the contribution of the paper.

Overall, I think this paper presents a set of interesting results, and the general direction may be of interest for future research. I thus recommend for accept.